# Molecular mechanisms of inorganic-phosphate release from the core and barbed end of actin filaments

Wout Oosterheert [1,5], Florian E. C. Blanc [2,5], Ankit Roy [3], Alexander Belyy [1], Micaela Boiero Sanders [1], Oliver Hofnagel[1], Gerhard Hummer [2,4] ✉, Peter Bieling [3] ✉ & Stefan Raunser [1] ✉

The release of inorganic phosphate ($P_i$) from actin filaments constitutes a key step in their regulated turnover, which is fundamental to many cellular functions. The mechanisms underlying $P_i$ release from the core and barbed end of actin filaments remain unclear. Here, using human and bovine actin isoforms, we combine cryo-EM with molecular-dynamics simulations and in vitro reconstitution to demonstrate how actin releases $P_i$ through a 'molecular backdoor'. While constantly open at the barbed end, the backdoor is predominantly closed in filament-core subunits and opens only transiently through concerted amino acid rearrangements. This explains why $P_i$ escapes rapidly from the filament end but slowly from internal subunits. In a nemaline-myopathy-associated actin variant, the backdoor is predominantly open in filament-core subunits, resulting in accelerated $P_i$ release and filaments with drastically shortened ADP-$P_i$ caps. Our results provide the molecular basis for $P_i$ release from actin and exemplify how a disease-linked mutation distorts the nucleotide-state distribution and atomic structure of the filament.

The dynamic turnover of actin filaments (F-actin) controls the shape and movement of eukaryotic cells and is driven by changes in the molecular identity of the adenine nucleotide that is bound to actin[1–3]. Monomeric actin (G-actin) displays only weak ATPase activity[4], but actin polymerization results in the flattening of the protein and a rearrangement of amino acids and water molecules near the nucleotide-binding site[5–8]. Accordingly, the ATPase activity of actin increases by about 42,000-fold, and hydrolysis takes place within seconds of filament formation (at a rate of 0.3 s⁻¹) (ref. 9). Because the release of cleaved $P_i$ occurs at much slower rates than does ATP hydrolysis[10], the cap of a growing filament is generally rich in ADP-$P_i$-bound subunits[11], whereas 'aged' F-actin primarily adopts the ADP-bound state. In vivo, these changes in the nucleotide state are sensed by several actin-binding proteins

(ABPs)[12–14]. For instance, ADF/cofilin family proteins efficiently bind and sever the ADP-bound state of the filament, but bind only weakly to ADP-$P_i$–F-actin[15–17]. Severing by ADF/cofilin promotes actin turnover following network assembly[18], making $P_i$ release from the F-actin interior an essential mechanism for polarized, directed cell migration.

It was long debated whether the $P_i$ release rate of a given F-actin subunit depends on the nucleotide state of its neighboring subunits[10,19,20]. Finally, single-filament experiments showed that $P_i$ release from the F-actin core is stochastic[21,22]; each ADP-$P_i$-bound subunit in the filament releases its bound phosphate with equal probability at a rate of 0.002 to 0.007 s⁻¹, which corresponds to a half-time ($\tau$) of several minutes[10,21–25]. During filament growth, $P_i$ release occurs solely from filament core subunits because the barbed-end growth velocity

[1]Department of Structural Biochemistry, Max Planck Institute of Molecular Physiology, Dortmund, Germany. [2]Department of Theoretical Biophysics, Max Planck Institute of Biophysics, Frankfurt am Main, Germany. [3]Department of Systemic Cell Biology, Max Planck Institute of Molecular Physiology, Dortmund, Germany. [4]Institute for Biophysics, Goethe University, Frankfurt am Main, Germany. [5]These authors contributed equally: Wout Oosterheert, Florian E. C. Blanc. ✉e-mail: gerhard.hummer@biophys.mpg.de; peter.bieling@mpi-dortmund.mpg.de; stefan.raunser@mpi-dortmund.mpg.de

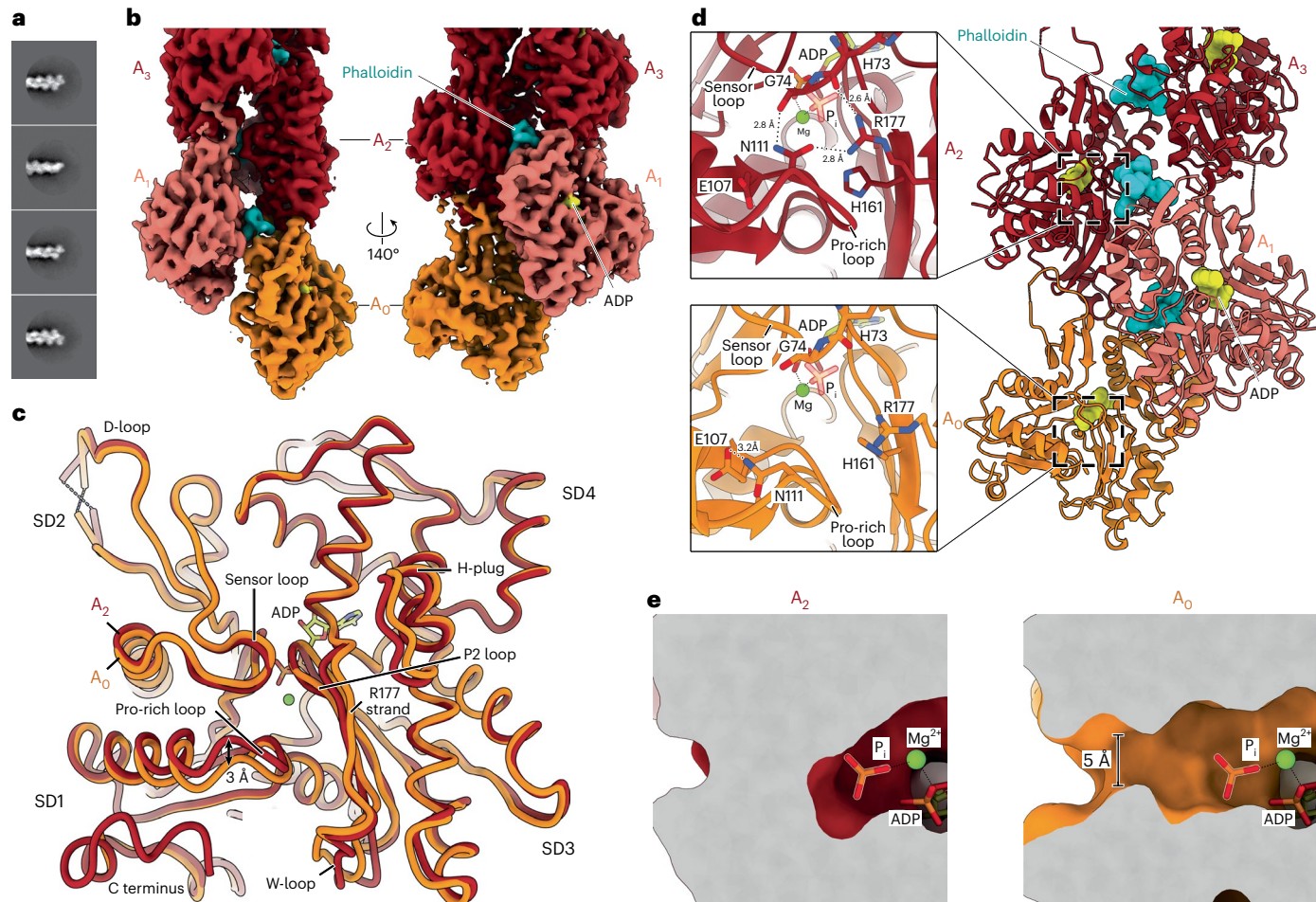

**Fig. 1 | Cryo-EM structure of the phalloidin-bound barbed end of F-actin.**
**a**, Exemplary two-dimensional (2D) class averages of the barbed-end particles. The particle boxes are 465 × 465 Å². Additional 2D class averages are depicted in Extended Data Figure 1b. **b**, Cryo-EM density map of the barbed end at 3.6-Å resolution, shown in two orientations. Actin subunits with complete inter-subunit contacts (A₂ and above) are colored dark red, the penultimate subunit (A₁) is colored salmon, and the ultimate subunit (A₀) is depicted in orange. Phalloidin (cyan) and ADP (yellow) are annotated. **c**, Superimposition of the A₂ and A₀ subunits at the barbed end. Actin subdomains (SD1–SD4) and the regions

that are different between the two subunits are annotated. P2 loop, residues 154–161. R177 strand, residues 176–178. **d**, Right, molecular model of the F-actin barbed end. The images on the left depict a zoom-in on the R177–N111 backdoor as seen from the F-actin exterior. Amino acids that form the backdoor are annotated. Although the subunits adopt the ADP state, $P_i$ from PDB 8A2S (F-actin in the Mg²⁺-ADP-$P_i$ state) is fitted and shown semi-transparently to emphasize the $P_i$-binding site. **e**, A slice through the structures of the A₂ and A₀ subunits near the nucleotide-binding site, shown in surface representation. Similar to **d**, $P_i$ from PDB 8A2S is shown to emphasize the $P_i$-binding site.

(~10–500 monomers s⁻¹) (ref. 26) is much faster than the ATP hydrolysis rate of actin (0.3 s⁻¹), indicating that barbed-end subunits effectively adopt the ATP state only before becoming internal subunits. However, after the transition from filament growth to depolymerization, actin subunits that have hydrolyzed ATP and adopt the ADP-$P_i$ state can be exposed at the shortening barbed end. Interestingly, these barbed-end subunits release $P_i$ more than 300-fold faster (~2 s⁻¹) than do those within the filament core, even though the affinities for $P_i$ binding to the barbed end and the filament core are essentially the same[21,27].

At the atomic level, $P_i$ release from actin was investigated in pioneering molecular dynamics (MD) simulation studies in the late 1990s (refs. 28,29), which suggested that the disruption of the ionic bond between $P_i$ and the nucleotide-associated divalent cation (Mg²⁺ or Ca²⁺) could represent the rate-limiting step for $P_i$ release, because actin rearrangements were not required for $P_i$ to exit. These studies furthermore predicted that $P_i$ escapes through an open 'backdoor' in the actin molecule, with residues R177 and methylated H73 potentially being mediators of $P_i$ release. However, a central limitation of the simulations was that they were performed on a G-actin structure[30], because high-resolution F-actin structures were unavailable at the time.

The first sub-4-Å cryogenic electron microscopy (cryo-EM) structures of F-actin revealed that, within the flattened conformation of the filament, $P_i$ cannot freely diffuse from the F-actin interior[31]; R177 participates in a hydrogen bonding network with the side chain of N111 and the backbones of H73 and G74, forming a closed backdoor conformation in F-actin. Accordingly, recently published cryo-EM structures of F-actin bound to ADP-$P_i$ and ADP at resolutions beyond 2.5 Å showed a closed backdoor in both the pre- and post-release states[8,32]. This suggests that $P_i$ escapes the filament interior when F-actin is in a transient, high-energy state that requires substantial rearrangements and that is difficult to capture using static imaging techniques such as cryo-EM. In a recent model of $P_i$ release, a rotameric switch of residue S14 from a hydrogen bond interaction with the backbone amide of G74 to that of G158 would enable $P_i$ to approach the R177-N111 backdoor and egress[33]. However, in the absence of experimental validation, the molecular principles of $P_i$ release from F-actin remain elusive, and further evidence is required to determine whether $P_i$ exits the filament interior through the postulated backdoor or another route. Additionally, the structural basis for why $P_i$ release from actin subunits at the barbed end is orders of magnitude faster than from core subunits is still unknown.

**Table 1 | Cryo-EM data collection, refinement and validation statistics**

| dataset | β/γ-actin barbed end | β-actin-R183W | β-actin-N111S |
|---|---|---|---|
| | EMD-16887, PDB 8OI6 | EMD-16888, PDB 8OI8 | EMD-16889, PDB 8OID |
| **Data collection and processing** | | | |
| Magnification | ×120,000 | ×130,000 | ×130,000 |
| Voltage (kV) | 200 | 300 | 300 |
| Electron exposure (e$^-$/Å$^2$) | 56 | 70 | 69 |
| Defocus range (μm) | −1.2 to −2.7 | −0.7 to −2.0 | −0.7 to −2.0 |
| Pixel size (Å) | 1.21 | 0.695 | 0.695 |
| Symmetry imposed | $C_1$ | $C_1$ | $C_1$ |
| Initial particle images (no.) | 252,982 | 1,569,882 | 2,001,281 |
| Final particle images (no.) | 43,618 | 1,286,604 | 1,756,928 |
| Map resolution (Å) | | | |
| 0.143 FSC threshold | 3.59 | 2.28 | 2.30 |
| Map resolution range (Å) | 3.5–5.5 | 2.2–3.7 | 2.2–3.9 |
| Measured helical symmetry[a] | | | |
| Helical rise (Å) | 27.47 | 27.58±0.08 | 27.58±0.05 |
| Helical twist (°) | −165.7 | −166.5±0.1 | −166.5±0.3 |
| **Refinement** | | | |
| Initial model used (PDB code) | 8A2T | 8OID | 8A2T |
| Model resolution (Å) | | | |
| 0.5 FSC threshold | 3.79 | 2.30 | 2.30 |
| Map sharpening *B* factor (Å$^2$) | −140 | −55 | −60 |
| Model composition | | | |
| Non-hydrogen atoms | 11,573 | 14,895 | 15,206 |
| Protein residues | 1,464 | 1,815 | 1,850 |
| Waters | 0 | 585 | 616 |
| Ligands | 8 | 10 | 10 |
| *B* factors (Å$^2$) | | | |
| Protein | 76.52 | 32.06 | 31.10 |
| Waters | – | 27.36 | 27.47 |
| Ligand | 69.41 | 24.35 | 21.49 |
| R.m.s. deviations | | | |
| Bond lengths (Å) | 0.002 | 0.004 | 0.004 |
| Bond angles (°) | 0.589 | 0.568 | 0.597 |
| Validation | | | |
| EM-ringer score | 1.83 | 5.56 | 5.49 |
| MolProbity score | 1.42 | 1.13 | 1.45 |
| Clashscore | 7.67 | 3.42 | 4.07 |
| Poor rotamers (%) | 0.24 | 0.65 | 0.96 |
| Ramachandran plot | | | |
| Favored (%) | 98.10 | 98.60 | 96.16 |
| Allowed (%) | 1.83 | 1.40 | 3.84 |
| Disallowed (%) | 0.07 | 0 | 0 |

[a]For the β/γ-actin barbed-end structure, the helical rise and twist between the ultimate (A$_0$) and penultimate (A$_1$) subunit are reported. For the filament structures, the helical parameters were estimated from the atomic model of five consecutive subunits independently fitted to the map, as described previously[8].

Here, we uncover that P$_i$ release from filament core and barbed-end subunits occurs through a common backdoor. For internal subunits, the backdoor is predominantly closed and its transient opening is kinetically limiting, whereas at the barbed end, the backdoor is open and P$_i$ can escape from the ultimate actin subunit without large rearrangements in the protein. Strikingly, we also characterize an actin disease variant (p.N111S) that adopts an open backdoor arrangement in internal subunits and hence releases P$_i$ without considerable delay.

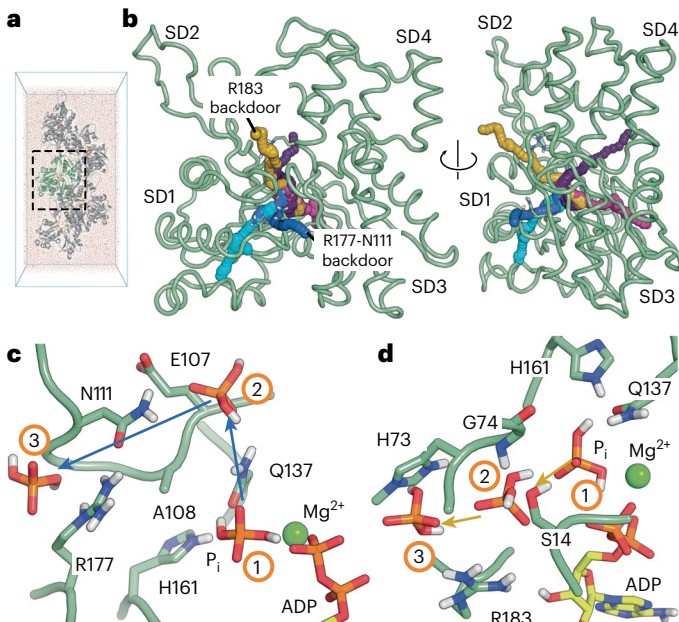

**Fig. 2 | Enhanced sampling simulations reveal several $P_i$-release paths from the F-actin core. a**, Simulation box containing an explicitly solvated actin pentamer. The core actin subunit is shown in pale green. **b**, Typical $P_i$-egress pathways from the actin core, obtained by enhanced sampling. Actin subdomains are annotated. The two plausible egress paths that were analyzed further are the putative R183 backdoor (gold) and R177-N111 backdoor (blue). Implausible $P_i$ release pathways are colored purple and magenta and are further shown in Extended Data Figure 4. The F-actin structure fluctuated during the simulations but is shown in a single, representative conformation for clarity. **c,d**, Close-up views of the plausible $P_i$-egress paths. Time-ordered representative $P_i$ positions (1–3) from enhanced sampling trajectories are shown and connected by arrows indicating the direction of $P_i$ movement. Similar to **b**, the F-actin structure is shown in a single, representative conformation for clarity. Close-up views are depicted for the potential R177-N111 backdoor path (**c**) and for the potential R183 backdoor path (**d**).

Our results provide a detailed molecular description of $P_i$ release from F-actin and outline a general approach of studying disease-linked actin variants.

## Results

### An open backdoor in the ultimate barbed-end subunit

To elucidate how the conformation of the barbed end of F-actin allows for much faster $P_i$-release kinetics during filament depolymerization, we first sought to solve the structure of the barbed end by cryo-EM. This required the generation of short filaments (<150 nm) so that we could pick enough filament ends per micrograph to determine the structure of the barbed end in high resolution. In vitro, actin polymerization generally results in long filaments (>500 nm) because filament growth is kinetically favored over nucleation. We therefore optimized a workflow for generating short filaments that featured DNase I, a G-actin-binding protein[34] that depolymerizes F-actin under physiological conditions. However, in the presence of the toxin phalloidin, DNase I does not disassemble F-actin and acts effectively as a pointed-end capper[35] that prevents filament reannealing. Accordingly, to generate short filaments, we mixed DNase I–G-actin complex with free G-actin and the FH2 domain of formin mDia1 (mDia1$_{FH2}$, which acts as an actin nucleator) in low-ionic-strength buffer, and added KCl to induce polymerization. Shortly after, we stabilized the formed filaments with phalloidin and then separated F-actin from unpolymerized actin and mDia1$_{FH2}$ using size-exclusion chromatography (SEC). Through this approach, we reproducibly formed short (~50–150 nm length) filaments and finally

obtained a cryo-EM structure of the barbed end at 3.6-Å resolution (Fig. 1a,b, Extended Data Fig. 1, Table 1 and Supplementary Video 1). Importantly, we did not find any evidence of an ABP that remained bound to the barbed end, indicating that our structure was the undecorated barbed end of F-actin.

The structure reveals the hallmark actin filament architecture of a double-stranded helix with a right-handed twist (Fig. 1b,d). All actin subunits in our reconstruction adopt the aged ADP-nucleotide state (Extended Data Fig. 2a,b), indicating that we captured the barbed-end structure in a state that resembles a depolymerizing filament. This is in line with our experimental setup, in which the filaments were separated from actin monomers after polymerization. In the structure, the arrangements of F-actin subunits that make all available inter-subunit contacts within the filament ($A_2$ and above), as well as the arrangement of the penultimate subunit $A_1$, are essentially the same as in previously determined F-actin structures (Extended Data Fig. 3, Cα r.m.s. deviation (r.m.s.d.) < 0.5 Å with PDB 8A2T) (ref. 8). Accordingly, in subunits $A_1$ and above, the predicted R177–N111 backdoor is closed and the $P_i$-binding site is shielded from the filament exterior (Fig. 1d and Extended Data Fig. 3e). Although the overall arrangement of the ultimate ($A_0$) subunit is also similar (Cα r.m.s.d. of 0.9 Å between subunits $A_0$ and $A_2$) and remains flattened (Extended Data Fig. 3c), we observed several differences when we compared it with the internal subunit $A_2$, including small rearrangements in the W-loop (residues 165–172) and in the hydrophobic plug region (residues 264–273), as well as a disordered F-actin carboxy terminus (residues 363–375) (Fig. 1c). However, the most striking rearrangement is the downward displacement of the Pro-rich loop (residues 107–112) by ~3 Å (Fig. 1c), which—unlike in internal actin subunits—is not stabilized by subdomain 4 (SD4) of an adjacent subunit (Extended Data Fig. 2d). Hence, N111 loses its hydrogen bonds with G74 in the sensor loop (residues 70–77) and SD3-residue R177 and instead interacts with E107, within its own local loop (Fig. 1d and Extended Data Fig. 3f). Interestingly, this E107-N111 interaction is commonly observed in G-actin structures (Extended Data Fig. 3f). Strikingly, residue H161, which flips rotameric position during the G- to F-actin transition and is important for ATP hydrolysis in F-actin[7,8], also adopts its G-actin-like position in the $A_0$ subunit and points towards R177. As a result, R177 can no longer interact with the sensor loop (Fig. 1d and Extended Data Fig. 3f). In conclusion, the hydrogen bonding network formed by R177, N111, H73, and G74 is fully abolished, which opens a hole of ~5 Å in diameter in the structure that connects the internal nucleotide-binding site to the filament exterior (Fig. 1e and Supplementary Video 1). This defines the predicted $P_i$ release backdoor[29] as open. Thus, our structure provides evidence that, under depolymerization conditions, $P_i$ can dissociate from the nucleotide-binding site at the barbed end without large protein rearrangements, thereby revealing the structural basis for why the ultimate barbed end subunit of actin releases $P_i$ at rates that are orders-of-magnitude faster than those from subunits within the filament core.

### Two plausible $P_i$-egress routes from the F-actin core

The barbed-end structure revealed a $P_i$-release pathway similar to the path that has been predicted in previous MD simulation studies[29]. Accordingly, when we performed MD simulations on $P_i$ release from barbed-end subunit $A_0$, we observed that virtually all $P_i$-escape events occurred through the open R177-N111 backdoor (Extended Data Fig. 4a,b). However, we reasoned that other $P_i$-exit routes may exist for F-actin core subunits, which make all available inter-subunit contacts and release $P_i$ at much slower rates. We therefore set out to develop an MD protocol to investigate the $P_i$-release mechanism from the F-actin filament core, using our recently reported ~2.2-Å structure of F-actin in the $Mg^{2+}$-ADP-$P_i$ state[8] as a high-quality starting model (Fig. 2a). Because $P_i$ release from the F-actin core is a slow, stochastic event with half-times >100 s, the diversity of possible $P_i$ escape paths cannot be studied using conventional MD simulations.

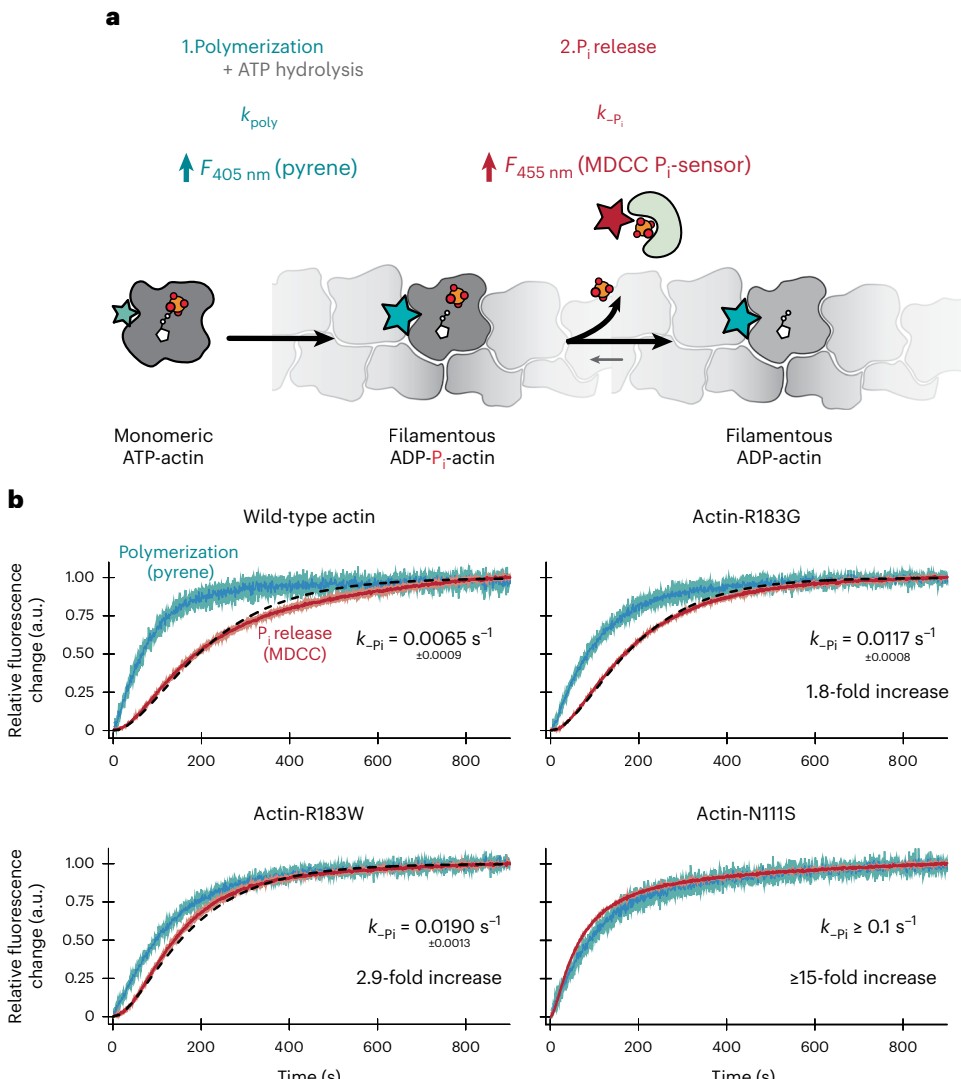

**Fig. 3 | Biochemical characterization of $P_i$ release from actin filaments.**
**a**, Scheme of the synchronous measurement of actin polymerization and subsequent $P_i$ release as measured by the fluorescence intensity increase of pyrene (cyan) and MDCC-PBP (red), respectively. $k_{poly}$, apparent rate of actin polymerization; $k_{-Pi}$, rate constant for phosphate release. $F_{405 nm}$ and $F_{455 nm}$ refer to the measured increase in fluorescent intensity at the specified emission wavelength. **b**, Timecourses of the normalized fluorescence intensities of 10 μM actin (either wild-type or mutants as indicated) containing 1.5% wild-type, pyrene

α-actin (cyan), and 30 μM MDCC-PBP (red) seeded with 160 nM spectrin-actin seeds after initiation of polymerization ($t = 0$ s). Dark colors indicate the average of three independent experiments, whereas the lighter colored areas indicate the s.d. The black dashed lines correspond to fits of the phosphate release data to a kinetic model (see Methods). Rates of $P_i$ release and relative rate enhancement over wild-type actin as determined either from fits to a kinetic model (wild-type, actin-R183G and actin-R183W) or by estimation from kinetic simulations (actin-N111S, see Methods and Extended Data Fig. 5c,d) are depicted in each graph.

Instead, we developed an enhanced-sampling simulation protocol based on metadynamics by applying a history-dependent repulsive potential to the $P_i$ Cartesian coordinates to progressively drive $P_i$ out of the nucleotide-binding site[36], without favoring any egress route a priori (see Methods). This protocol enables simulation of candidate $P_i$-egress pathways in about 10 ns.

Using this approach, we collected dozens of $P_i$-release events from the F-actin core and identified sterically accessible egress pathways (Fig. 2b). We then analyzed the diverse-sampled pathways for physical plausibility and first dismissed all pathways that entailed unrealistic distortions of F-actin or the nucleotide (Extended Data Fig. 4c–e). Second, we assumed that $P_i$ exits the F-actin interior near the binding sites of phalloidin and jasplakinolide, because both toxins strongly inhibit $P_i$ release[37–39]. This analysis resulted in two remaining egress pathways that were physically plausible. In the first, a side chain movement of Q137 allows $P_i$ to move into a hydrophilic pocket between E107 and H161, where it interacts with residues in the Pro-rich loop. From there,

$P_i$ escapes either by disrupting the R177-N111 hydrogen bond (leading to an open backdoor, similar to the conformation adopted by the barbed end) (Fig. 2c and Supplementary Video 2), by leaving close to residues N115 and R116, or by exiting near residues T120 and V370. Of note, phalloidin and jasplakinolide stabilize the R177-N111 interaction (Extended Data Fig. 2c), but would not interfere with $P_i$ egress near N115-R116 or T120-V370, suggesting that the exit path that requires the disruption of the R177-N111 interaction is the most probable. In the second plausible pathway, $P_i$ first breaks the hydrogen bond between S14 and G74 to enter a pocket between H73 and R183. Then, $P_i$ escapes to the intra-filament space upon breaking the strong electrostatic interaction with R183 (Fig. 2d). Interestingly, the disruption of the S14-G74 hydrogen bond has previously been proposed to play a role in $P_i$ release, albeit through a different mechanism[33]. Furthermore, phalloidin and jasplakinolide would prevent the opening of the H73-R183 pocket. Thus, in addition to the predicted route, there is another physically realistic egress pathway for $P_i$ from the F-actin core.

## Actin filaments with the N111S substitution release P$_i$ rapidly

To experimentally probe the two possible P$_i$-release pathways from the F-actin core, we mutated key residues in β-actin that pose a barrier for P$_i$ release in each pathway. For the first pathway, we introduced the p.N111S substitution to potentially disrupt the hydrogen bonding network of the R177-N111 backdoor. For the second pathway, we aimed to destabilize the putative R183-mediated backdoor using p.R183W and p.R183G mutations. Importantly, all actin mutants studied here are associated with human diseases; p.R183W was identified in β-actin from patients with deafness, juvenile-onset dystonia or development malfunctions[40,41], whereas p.R183G and p.N111S have been found in α-actin of patients with nemaline myopathy[42,43], highlighting the relevance of these actin variants.

We developed a fluorescence-based assay to synchronously monitor seeded actin polymerization using pyrene fluorescence and subsequent P$_i$ release via a fluorescent phosphate sensor in the same experiment (Fig. 3a). This allowed us to determine the respective reaction rates by fitting the data to a kinetic model (see Methods). Because we performed the experiments at actin concentrations (10 μM) that drive rapid filament growth and seeded the reaction with spectrin-actin seeds to circumvent slow nucleation, we effectively monitored P$_i$ release from core subunits and not from the barbed end. Wild-type β-actin had a slow P$_i$ release rate of 0.0065 s$^{-1}$ ($\tau \approx 107$ s) (Fig. 3b), which falls within the range of previously reported values for rabbit skeletal α-actin[10,21,25], indicating that slow P$_i$ release after polymerization and ATP hydrolysis is a conserved feature among mammalian actin isoforms. Actin-R183G and actin-R183W released P$_i$ at slightly increased rates of, respectively, 0.0117 s$^{-1}$ and 0.0190 s$^{-1}$, corresponding to a 1.8- and 2.9-fold increase compared with wild-type β-actin (Fig. 3b). Strikingly, actin-N111S exhibited ultrafast P$_i$ release kinetics without appreciable delay after polymerization. In fact, the reaction timecourses of P$_i$ release slightly outpaced the observed polymerization kinetics (Fig. 3b), making it impossible to determine an exact P$_i$ release rate for actin-N111S. However, when estimating the release rate conservatively (see Methods and Extended Data Fig. 5c,d), actin-N111S releases P$_i$ at a rate of ≥0.1 s$^{-1}$, which is ≥15-fold faster than the rate of wild-type actin. Thus, the R183 mutants release P$_i$ somewhat faster, but still display the characteristic delay between polymerization and P$_i$ release; by contrast, actin-N111S appears to release P$_i$ without appreciable delay.

## Structural basis for the fast P$_i$-release rates of actin-N111S

To structurally understand the differences in P$_i$-release rates between the R183W and N111S mutants, we determined the filament-core structures of these variants in the Mg$^{2+}$-ADP-bound state at ~2.3-Å by cryo-EM (Fig. 4a,b, Table 1 and Extended Data Figs. 6 and 7a,b). We modeled hundreds of water molecules bound to F-actin in both structures, as well as the exact rotameric positions of many amino acid side chains (Extended Data Fig. 7a–c). Globally, we observed no major differences between the two β-actin mutants and Mg$^{2+}$-ADP-bound α-actin filaments (PDB 8A2T, Cα r.m.s.d. < 0.7 Å) (Extended Data Fig. 7d) but, importantly, we identified small but impactful rearrangements.

We first examined the atomic arrangement near the mutated residues. In wild-type F-actin structures, the R183 side chain interacts with D157 and the carboxyl moieties of S14 and I71 (Fig. 4c). As expected for F-actin-R183W, these interactions are abolished and the W183 side chain points away from the mentioned residues (Fig. 4c). Nevertheless, the S14-G74 hydrogen bond remains intact in the mutant, and we did not observe an open cavity near W183 through which P$_i$ could escape (Fig. 4c); the R177-N111 backdoor is also intact (Fig. 4d,e).

The N111S substitution induces a more drastic conformational change in F-actin; residue S111 is too short to interact with R177 and G74. Instead, S111 forms a hydrogen bond with E107, the backbone of A108, and a water molecule within its local environment in the Pro-rich loop (Fig. 4d and Extended Data Fig. 7c). Additionally, the density for the side chain of R177 is fragmented (Extended Data Fig. 7c), indicating that the residue is more flexible than in wild-type F-actin structures. As a result, the interaction between the sensor loop and Pro-rich loop is disrupted and a ~4-Å-diameter hole is observed, yielding an open backdoor (Fig. 4d,e). Hence, the conformation of F-actin-N111S subunits is reminiscent of the arrangement of the ultimate subunit of the barbed-end structure (Fig. 1d). Accordingly, we identified that H161 adopts a mixture of rotameric states and that it partially adopts a G-actin-like conformation (Fig. 4d and Extended Data Fig. 7c). This observation suggests that repositioning of the H161 side chain is a key feature of backdoor opening. Conversely, the S14-G74 interaction is not affected in F-actin-N111S (Fig. 4c), indicating that disruption of this interaction is not required to open the backdoor, as has previously been proposed[33]. Thus, although the backbone atom shifts in actin-N111S are minor when compared with those in wild-type actin (< 1 Å), side chain rearrangements result in a broken hydrogen bonding network and an open backdoor (Fig. 4d,e, Extended Data Fig. 7c and Supplementary Video 3), defining the structural basis for the ultrafast P$_i$-release kinetics of F-actin-N111S.

We then inspected the conformation of ADP and the associated Mg$^{2+}$ ion in both mutant structures. In the F-actin-N111S structure, the nucleotide arrangement is essentially the same as that in Mg$^{2+}$-ADP wild-type α-actin; the Mg$^{2+}$ ion resides beneath the β-phosphate of ADP and interacts with five water molecules (Fig. 4b, Extended Data Fig. 7e and Supplementary Video 3). Interestingly, in F-actin-R183W, Mg$^{2+}$ changes position so that it is directly coordinated by both α- and β-phosphates, as well as by four water molecules (Fig. 4a and Extended Data Fig. 7e). Although R183 does not directly interact with ADP in wild-type F-actin, it resides in close proximity to the nucleotide and, importantly, it is positioned in a negatively charged cluster of acidic residues (Extended Data Fig. 8). Hence, F-actin-R183W has a more negatively charged nucleotide-binding site (Extended Data Fig. 8), which may explain why the positively charged Mg$^{2+}$ repositions in the mutant structure to compensate for the charge imbalance. Thus, F-actin-R183W has an altered nucleotide-binding site but does not reveal conformational changes that would enable P$_i$ egress, indicating that the R183 backdoor is not part of the dominant P$_i$-escape path in wild-type actin. Although we cannot exclude the possibility that this release path is marginally sampled, we propose that, alternatively, P$_i$ still mainly exits through the R177-N111 backdoor in the R183 mutants. In that scenario, the more negatively charged nucleotide-binding

**Fig. 4 | High-resolution cryo-EM structures of F-actin-R183W and F-actin-N111S. a,b,** Sharpened cryo-EM density maps of F-actin-R183W (**a**) and F-actin-N111S (**b**) in the Mg$^{2+}$-ADP nucleotide state. Five filament subunits are shown. For the R183 mutant, the central subunit is colored gold; the central subunit of the N111S mutant is colored blue. The helical rise and twist are annotated. Densities corresponding to water molecules are shown in red. For each mutant, a zoom-in of the nucleotide-binding pocket with cryo-EM densities for the nucleotide (depicted in yellow), the Mg$^{2+}$ ion (green), and water molecules (red) is shown. Water molecules that directly coordinate the Mg$^{2+}$ ion are colored magenta. The polar ends of the actin filament, (−) pointed and (+) barbed, are annotated. **c–e,** Structural comparison between filamentous wild-type α-actin (PDB 82AT, left, green), β-actin-R183W (middle, gold), and β-actin-N111S (right, blue) in the Mg$^{2+}$-ADP state. The panels depict the amino acid environment near the location of residue 183 (**c**), the amino acid environment near residues 177 and 111 (**d**), and a slice through the F-actin interior near the P$_i$-binding site (**e**). In **c** and **d**, F-actin is shown as a cartoon, and amino acids are shown as sticks and are annotated. In **e**, F-actin is depicted in surface representation. Water molecules are omitted from **c–e**. Although the depicted structures adopt the ADP state, P$_i$ from PDB 8A2S (F-actin in the Mg$^{2+}$-ADP-P$_i$ state) is fitted and shown semi-transparently to emphasize the P$_i$-binding site.

pocket in these mutants may explain why the negatively charged $P_i$ is released slightly faster (Fig. 3b,c). Interestingly, it has previously been shown that p.R183W results in perturbed nucleotide release from actin monomers and impaired binding of non-muscle myosin-2A to the actin filament[41]. It is therefore expected that these altered molecular properties of actin-R183W are the major cause of defects in actin organization that lead to disease, rather than the slightly altered $P_i$-release kinetics of this actin variant.

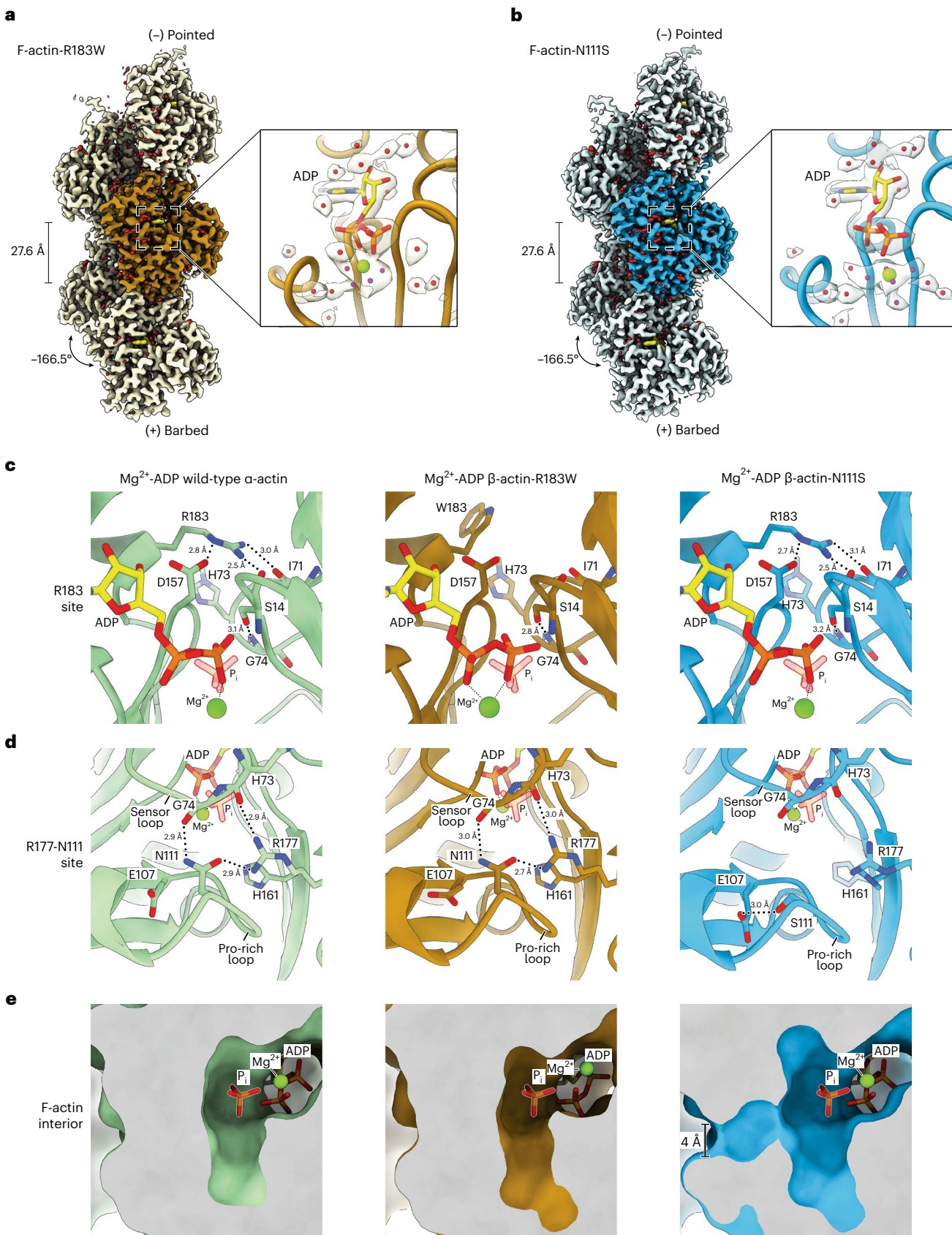

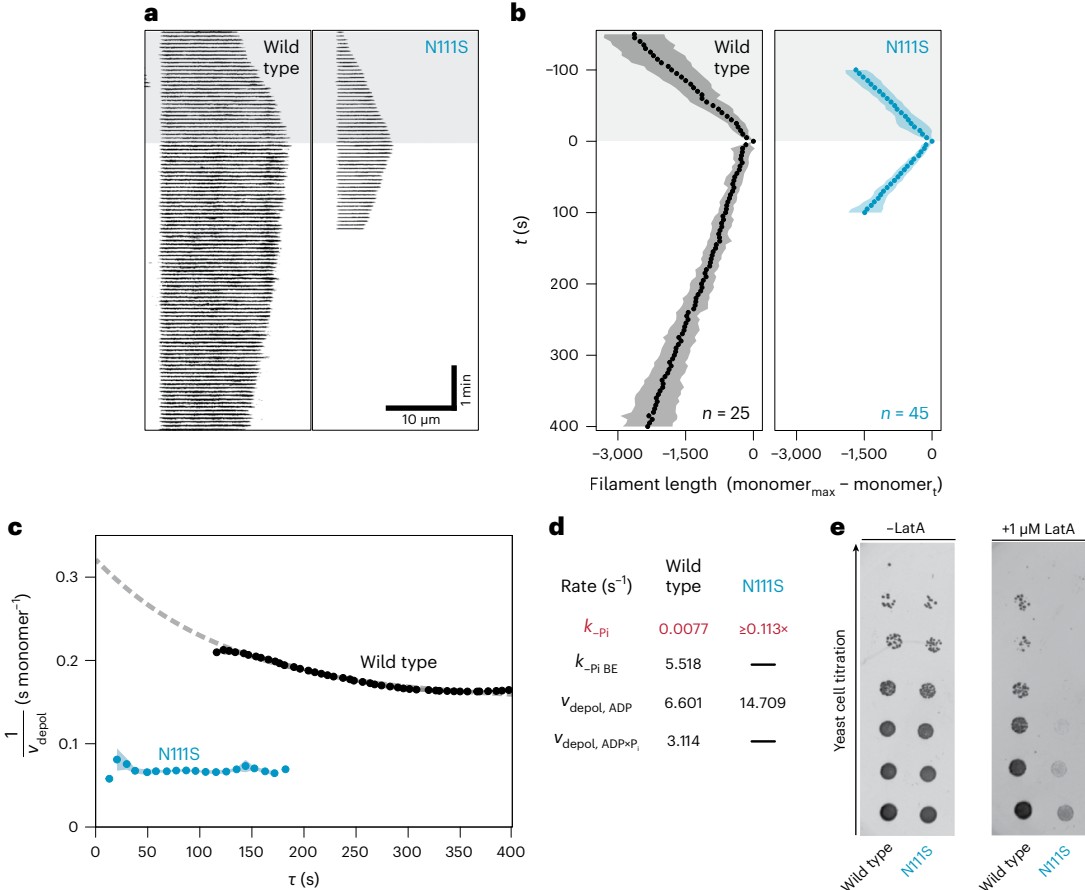

**Fig. 5 | Effects of the N111S substitution on single actin filaments in vitro and on yeast growth in vivo. a**, Timelapse TIRF imaging of single actin filaments (black, visualized by Lifeact-Alexa-Fluor-488) polymerized from spectrin-coated surfaces in microfluidic flow chambers. The vertical and horizontal axes indicate time and filament length, respectively. Polymerization and depolymerization phases are demarcated by the gray shaded and unshaded areas, respectively. **b**, Filament lengths tracked over the course of polymerization (gray shaded area) and depolymerization (unshaded area). Points indicate average filament length calculated from 25 and 45 filaments polymerized from either wild-type actin or actin-N111S, respectively. Shaded areas indicate the s.d. Wild-type and N111S filaments were analyzed from 6 and 12 independent experiments, respectively. *t*, time since depolymerization. Monomer$_{max}$ and monomer$_t$ are defined as the filament length in monomers at the maximum length and at time = *t*, respectively.

**c**, The inverse of the instantaneous depolymerization velocity ($1/v_{depol}$), calculated from the average filament length in **b**, as a function of filament age ($\tau$) upon depolymerization. Points and the shaded area around them indicate average $1/v_{depol}$ calculated from varying window sizes and the s.d., respectively. The dashed line indicates the fit to a kinetic model (see Methods). **d**, Rate constants of P$_i$ release from the filament interior and barbed end ($k_{-Pi}$ and $k_{-PiBE}$) and depolymerization velocities of ADP-P$_i$ or ADP subunits from the filament end ($v_{depol,ADP}$ and $v_{depol,ADP-Pi}$) determined from fits obtained in **c**. The asterisk indicates the lower bound estimate for N111S (see Methods and Extended Data Fig. 5h). **e**, Growth phenotype assay of yeast expressing either the wild type or N111S variant of *S. cerevisiae* actin. The two yeast strains were grown in the absence or presence of 1 µM latrunculin A (LatA).

## Actin-N111S filaments display markedly shortened ADP-P$_i$ caps

Next, we hypothesized that actin-N111S, which releases P$_i$ rapidly in bulk assays and adopts an open-backdoor conformation, should also display strong differences in nucleotide-state distribution at the single-filament level compared with wild-type actin. We reasoned that actin-N111S filaments should form drastically shortened ADP-P$_i$ caps, which we tested in microfluidic flow-out assays using total internal reflection fluorescence (TIRF) microscopy (Fig. 5a and Supplementary Video 4). Filaments were elongated from surface-immobilized spectrin-actin seeds using either wild-type or actin-N111S and were then rapidly switched to depolymerization with buffer lacking soluble actin (see Methods). For wild-type β-actin, the speed of filament depolymerization after flow-out gradually increased over a few minutes before reaching the maximum rate (Fig. 5b,c). Since ADP-P$_i$-bound actin depolymerizes at slower rates than does ADP-bound actin, this change is caused by the filament-depolymerizing region slowly maturing from an ADP-P$_i$-rich to an ADP-rich composition through P$_i$ release. The measured depolymerization velocities are well described by a kinetic model

(Fig. 5d and Methods), with parameters similar to those previously measured in single-filament assays on skeletal α-actin[21], as well as from our own bulk measurements (Figs. 3b and 5d). Strikingly, we observed that filaments grown from actin-N111S depolymerized at a high velocity ($v_{depol,ADP} = 14.71\ s^{-1}$) (Fig. 5b,c). More importantly, we found no appreciable change in the depolymerization velocity after buffer flow-out (Fig. 5c), indicating that the ADP-P$_i$ to ADP-actin transition is very fast and was not captured within our experimental resolution. This allowed us to estimate a lower bound for the phosphate-release rate from the filament interior $k_{-Pi} \geq 0.113\ s^{-1}$ for actin-N111S (Fig. 5d and Extended Data Fig. 5e). Hence, our data reveal that the rapid rate of P$_i$ release indeed results in drastically shortened ADP-P$_i$ caps in actin-N111S filaments.

To investigate the effect of the N111S substitution in vivo, we compared the growth rates of yeast strains expressing either wild-type or the N111S variant of *Saccharomyces cerevisiae* actin. Under normal conditions, we observed no major alterations in growth phenotype between the two strains (Fig. 5e). Accordingly, live-cell fluorescence imaging of the two yeast variants did not reveal large differences

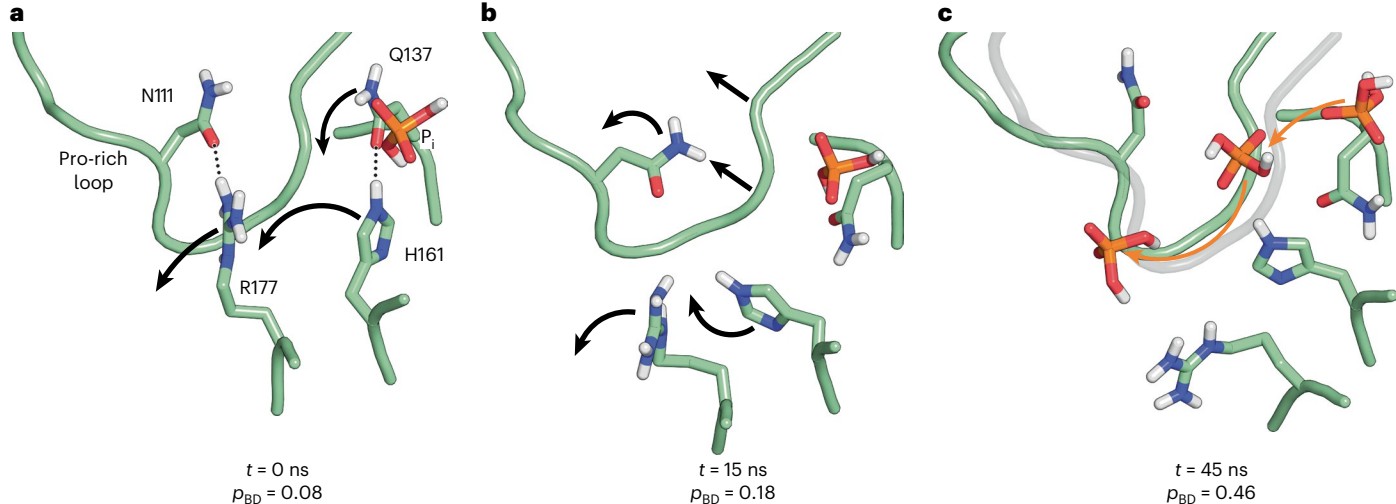

**Fig. 6 | Backdoor opening mechanism revealed by SMD simulations.**
**a–c**, Selected frames from the SMD trajectory. Dotted lines indicate hydrogen bonds. Arrows indicate the side chain and backbone movements involved in the opening of the backdoor. Time $t$ and $P_i$-release propensity ($p_{BD}$) along the trajectory are shown. In **c**, the orange arrows indicate the direction of $P_i$ movement through the open R177-N111 backdoor observed in a representative enhanced sampling simulation.

in the lifetime of endocytic actin patches (Extended Data Fig. 5i,j). However, when exposed to the toxin latrunculin A, which sequesters G-actin and accelerates F-actin depolymerization[44], the yeast expressing actin-N111S displayed a dramatically reduced growth rate compared with that of yeast expressing wild-type actin (Fig. 5e). This high sensitivity to latrunculin-A-induced stress suggests that actin-N111S filaments are more labile and prone to depolymerization. Thus, the phenotype observed for *S. cerevisiae* actin-N111S in vivo is in line with our in vitro experiments investigating human β-actin-N111S, for which we observed faster $P_i$-release and filament-depolymerization rates than were observed for wild-type actin.

Taken together, our experiments provide strong evidence that the first $P_i$-egress pathway identified by enhanced-sampling MD, encompassing the R177-N111 backdoor, is the dominant $P_i$-egress pathway from the F-actin interior. Hence, $P_i$ is released from the core and barbed end of F-actin through similar exit routes, although core subunits require additional conformational rearrangements to transiently open the R177-N111 backdoor.

**The transient F-actin core state that allows for $P_i$ release**
Finally, we sought to understand how $P_i$ is released through the R177-N111 backdoor from the wild-type F-actin core. Our enhanced-sampling MD suggested that there are $P_i$-escape pathways through the backdoor (Fig. 2c), but this protocol is aggressive by design, to explore diverse possible pathways, and may not reflect a realistic order of molecular motion. Therefore, we first ran unbiased MD simulations to investigate the backdoor conformation in F-actin. The simulations revealed the disruption and reformation of the R177-N111 interaction several times over a 1.1-μs period (Extended Data Fig. 9a). However, this never resulted in a functionally open backdoor (Extended Data Fig. 9b), indicating that more rearrangements are required for $P_i$ to escape, and these were not captured in the timeframe of our unbiased simulations. This is in line with the very slow $P_i$ release from wild-type actin (Fig. 3b).

Therefore, to gain further mechanistic insights into the formation of this transient state of the F-actin core that allows for $P_i$ release, we turned to steered-MD (SMD) simulations. We performed 40 SMD simulations of 100 ns each, in which selected regions of the backbone of an F-actin core protomer (with either charged or neutral methylated H73) were transitioned to the barbed-end conformation, which is a prototypical example of a functionally open backdoor (see

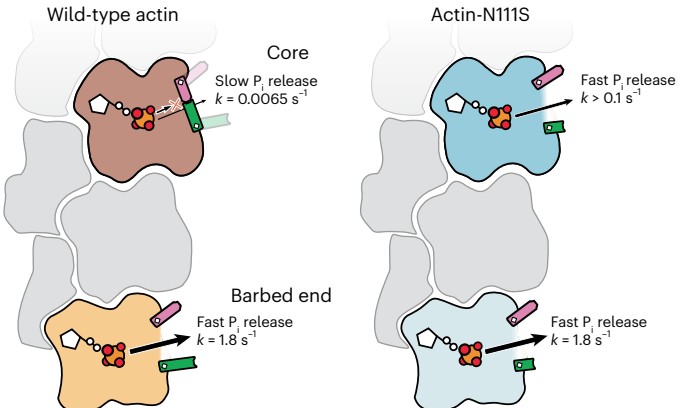

**Fig. 7 | Cartoon model of $P_i$ release from wild-type and N111S-F-actin.** In wild-type F-actin, subunits that reside in the filament core predominantly adopt a closed-backdoor conformation and hence release $P_i$ at slow rates. In the ultimate subunit at the barbed end, the backdoor is open, leading to 300-fold faster $P_i$ release during actin depolymerization. In F-actin-N111S, the amino acid substitution results in an open backdoor in internal subunits, leading to increased $P_i$-release rates.

Methods). Because this protocol steers the conformational transition in a quasi-deterministic fashion, structural rearrangements happen much faster than for stochastic backdoor opening. The majority of simulations showed movement of the Pro-rich loop and disruption of the R177-N111 hydrogen bond within 25 ns (Fig. 6a,b and Extended Data Figs. 9c–k and 10), representing the first major step of backdoor opening. The rotation of H161 to a G-actin-like conformation was also observed in many simulations and generally occurred after disruption of the R177-N111 interaction within 40–50 ns (Fig. 6b,c and Extended Data Fig. 10), while inter-subunit contacts within the filament remained intact (Extended Data Fig. 9k). We then evaluated the $P_i$-release efficiency of structures extracted along these key timepoints in the SMD trajectories using our previously introduced metadynamics-based protocol. Although we did not observe a major effect of the charge of methylated H73 on backdoor opening, our simulations revealed a high probability (Fig. 6c, Extended Data

Fig. 10a,b and Methods) that $P_i$ would escape through the backdoor when the R177-N111 interaction was disrupted and the H161 side chain was flipped, suggesting that both events are required to stabilize the open-backdoor conformation. Thus, although only the backbone movement was steered in our SMD setup, the simulations elucidate the side chain rearrangements that lead to the transient opening of the backdoor in the F-actin core (Fig. 6).

## Discussion

By combining cryo-EM with in vitro reconstitution and MD simulations, we have elucidated how $P_i$ is released from F-actin (Figs. 6 and 7) and that the hydrogen bonding network between the Pro-rich loop, sensor loop, and R177 strand forms the predominant backdoor. At the barbed end, the door for $P_i$ release is open in the ultimate subunit, indicating that $P_i$ can egress without large protein rearrangements. However, the backdoor is closed in F-actin core subunits and opens only transiently through a high-energy state. The opening of the door is likely initiated by a stochastic disruption of the hydrogen bonding network and is further induced by the change of rotameric position of H161 to a G-actin-like conformation. After $P_i$ egress, H161 flips back to its original position and the hydrogen-bonding network is re-established, allowing F-actin to adopt its low-energy state with a closed backdoor. In actin-N111S, the backdoor is predominantly open in all subunits because the introduced S111 side chain is too short to maintain the interactions that keep the door closed. Hence, F-actin-N111S releases $P_i$ rapidly upon polymerization, thereby drastically reducing the fraction of ADP-$P_i$-bound subunits in the filament. Thus, our experimental and simulation data provide conclusive evidence that the $P_i$-release rate from F-actin is controlled by steric hindrance through the backdoor.

Our barbed-end structure also provides further implications for actin polymerization. During the G- to F-actin transition, the side chain of residue H161 flips towards ATP, which triggers the relocation of water molecules near the nucleotide and, as a result, creates a favorable environment for ATP hydrolysis[7,8]. By contrast, at the barbed end, H161 adopts its G-actin-like rotameric position and points away from the nucleotide in the ultimate subunit, which suggests that, regardless of the growth velocity of the filament, the last subunit becomes ATP-hydrolysis-competent only when another actin subunit is added to the filament. This means that the full G- to F-form transition of an actin subunit should be considered a multi-step process, which not only encompasses the initial incorporation of that subunit into the filament, but also requires the subsequent binding of the next actin subunit. Formal proof for such a mechanism will require future investigations of the barbed end of F-actin in the ATP state. In general, we believe that investigating the structure of actin ends, either undecorated or bound by diverse classes of ABPs, will remain a prevalent theme of future research, guided by recent advances in actin-end structure determination by our lab and others[45–48].

It has recently been shown that the presence of $P_i$ at the nucleotide-binding site affects the bending structural landscape of F-actin[32], which raises the question of whether backdoor opening could be affected by filament bending. Although backdoor opening involves a confined region of SD1 (the Pro-rich loop and sensor loop) and SD3 (R177-strand) in actin, the subdomains that display the largest displacements during filament bending are SD2 and SD4 (ref. 32). Accordingly, the backdoor is closed in all subunits of bent F-actin structures in the ADP (PDB 8D15) and ADP-$P_i$ (PDB 8D16) states, suggesting that the backdoor state is not strongly affected by filament bending. In addition, if filament bending dramatically affected the $P_i$-release kinetics of actin, one would expect $P_i$ release to be cooperative, that is, actin subunits in the ADP state would bend more and thereby stimulate $P_i$ release from neighboring subunits that adopt the ADP-$P_i$ state. However, $P_i$ release from the F-actin core is stochastic[21]. Therefore, we do not anticipate that filament bending majorly affects the dynamic opening and closing of the backdoor.

How are the kinetics of $P_i$ release regulated in a complex cellular environment? Interestingly, the rates of $P_i$ release from the filament interior observed in vitro appear to be slower than the turnover of some cellular actin structures, such as lamellipodial networks and endocytic patches[49]. This strongly suggests that $P_i$ release in vivo is accelerated by ABPs through yet poorly defined mechanisms, which can be discussed in the context of our results. The $P_i$-release backdoor opens towards the inner side of the filament, adjacent to where the two strands of the actin helix interact (Fig. 1d). This site is targeted by the small-molecule toxins phalloidin and jasplakinolide, which block backdoor opening and hence sterically inhibit $P_i$ release[37–39]. Importantly, however, the backdoor and its immediate surroundings are not known to be directly engaged by many ABPs, presumably because molecules larger than phalloidin and jasplakinolide cannot easily enter the narrow cavity between the two actin strands. Accordingly, factors implicated in actin aging and turnover, such as ADF/cofilin and associated regulators like coronin[50,51], all bind at the filament periphery[52–54], suggesting that these ABPs can affect $P_i$ release only allosterically. Such a mechanism has indeed been postulated for ADF/cofilin proteins, which efficiently bind and sever F-actin only following $P_i$ release, but also accelerate the release of $P_i$ from the filament[15,24]. Structurally, cofilin binding changes the tilt of F-actin[52,53], resulting in rearrangements at the nucleotide-binding site that are incompatible with the presence of $P_i$, rendering actin tilting and $P_i$ binding mutually exclusive[8,53]. Hence, initial cofilin binding to the filament likely results in an actin conformation that releases $P_i$ more rapidly. However, how such ABP-induced conformational changes promote backdoor opening, and how this relates to actin disassembly in cells, remains to be elucidated.

Finally, can our results explain how the N111S substitution leads to nemaline myopathy, a disease affecting α-actin in skeletal muscle? Although F-actin in striated muscle is expected to undergo less turnover than cytoplasmic actin isoforms, it is well established that actin-severing proteins, such as cofilins, are expressed in muscle sarcomeres to control actin (thin) filament length during sarcomerogenesis and actin turnover[55–57]. Since cofilins preferably sever ADP- over ADP-$P_i$-bound F-actin, we propose that the ultrafast $P_i$-release kinetics of actin-N111S may contribute to the pathophysiology in patients with this disease-linked mutation. Moreover, it will be interesting to study the effects of the N111S substitution in actin in non-muscle tissue. Because actin-N111S does not majorly populate the ADP-$P_i$ state, this mutant could be a unique tool for investigating the role of the metastable ADP-$P_i$ nucleotide state both in vivo and in vitro. We envision that our approach of studying mutants actin-R183W and actin-N111S, in which we combined biochemical experiments with high-resolution cryo-EM, will be instrumental in elucidating the molecular mechanisms of other disease-associated actin mutants. Specifically, visualizing the impact of substitutions on the atomic structure of F-actin will allow for the formulation of new hypotheses on how these affect cellular processes and how this is linked to disease. Therefore, this approach may ultimately contribute to the development of new therapeutic strategies for the treatment of human diseases that are characterized by mutations in actin genes.

## Online content

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

## Methods

### DNA constructs and yeast strains

Throughout the manuscript, we used human β-actin amino acid numbering that is consistent with the numbering in the corresponding UniProt entry (P60709, ACTB_HUMAN), with the initiator methionine numbered as residue 1, even though this methionine is cleaved off during actin maturation. Hence, the amino acid numbers for β-actin used in our paper align with the sequences of mature human and rabbit skeletal α-actin, as well as *S. cerevisiae* actin, facilitating a direct comparison between all actin isoforms used in this study.

The plasmid for the expression of recombinant human β-actin (p2336 pFL_ACTB_C272A) was described previously[58]. All mutations were introduced *via* QuikChange PCR, using p2336 as template (Supplementary Table 1). All β-actin constructs contain the C272A substitution (including the protein referred to as wild type), as C272 is prone to oxidation in aqueous solutions[59]. The equivalent residue in human skeletal α-actin is also alanine. The FH2 domain of *Mus musculus* mDia1 (mDia1$_{FH2}$) (amino acids 750–1163) was cloned into an pETMSumoH10 expression vector by Gibson assembly.

*S. cerevisiae* strains were grown in minimal medium containing yeast nitrogen base without amino acids (Difco) containing glucose and supplemented with tryptophane, adenine, histidine, and/or uracil if required. *S. cerevisiae* strains were transformed using the lithium-acetate method[60]. Yeast actin mutagenesis was performed as described previously[61]. Yeast viability in the presence or absence of latrunculin A was analyzed by drop test assays: fivefold serial dilutions of cell suspensions were prepared from overnight agar cultures by normalizing measurements of the optical density at 600 nm ($OD_{600}$), then plated onto agar plates and incubated at 30 °C for 2 d. For live-cell imagining, yeast strains were transformed with plasmid 2477 (ref. 62) encoding Lifeact-mCherry under the constitutive promoter of *abp140*.

### Live-cell imaging of *S. cerevisiae* variants

Yeast cells expressing either wild-type actin or actin-N111S were cultured at 30 °C in minimal medium (2% glucose, 6.7 g L$^{-1}$ yeast nitrogen base without amino acids, supplemented with 20 mg L$^{-1}$ tryptophan and uracil) for at least one full cell cycle and were imaged when at an $OD_{600}$ of between 0.5 and 1. Cells were seeded between the slide and coverslip, and all videos were acquired in less than 10 min after slide preparation. Cells were imaged using a spinning disk confocal device on the 3i Marianas system equipped with an Axio Observer Z1 microscope (Zeiss), a CSU-X1 confocal scanner unit (Yokogawa Electric Corporation), a Plan-Apochromat ×100/1.4 NA Oil Objective (Zeiss), and Orca Flash 4.0 sCMOS Camera (Hamamatsu), which was operated with Slidebook version 2022 software (3i). Temperature was maintained at 30 °C ± 1 °C at all times during imaging. Time-lapse images were collected every 500 ms for a total of 60 s of recording Lifeact intensity, which resulted in videos with 120 frames. To study in vivo actin dynamics, endocytic actin patches were detected and analyzed using the Fiji plugin TrackMate[63]. Patch detection was performed using a median filter and a Laplacian of Gaussian particle-detection algorithm for spots of 0.7 µm in diameter. Patches recognized by TrackMate were monitored, and the detected intensities that did not represent actin patches, that is, actin cables, were manually removed. Patch tracks were assigned using a simple LAP tracker. Resulting data were curated using a custom Python script to pre-process the data. Only patches in which the full lifespan was captured within the video were considered, and patches with a lifespan shorter than 4 s were not taken into account to avoid cable detection artifacts.

### Protein expression and purification

A mixture of native bovine cytoplasmic β-actin and γ-actin (from now on referred to as β,γ-actin) was purified from bovine thymus tissue, as described previously[26,46]. Bovine β and γ-actin both display 100% amino acid sequence identity to their corresponding human orthologs.

Human cytoplasmic β-actin variants were recombinantly expressed as fusion proteins, with thymosin β4 and a deca-His-tag fused to the actin C terminus[58,64]. β-actin was expressed in BTI-Tnao38 insect cells (CVCL_Z252, provided by S. Wohlgemuth and A. Musacchio, – Max Planck Institute of Molecular Physiology) using baculovirus infection. At ~48 h after infection, the cells were pelleted (5,000$g$, 10 min) and stored at −80 °C until use. On the day of purification, the insect cells were thawed, resuspended in lysis buffer containing 10 mM Tris pH 8, 50 mM KCl, 5 mM CaCl$_2$, 1 mM ATP, 0.5 mM TCEP, and cOmplete protease inhibitor (Roche), and were subsequently lysed using a fluidizer. The lysed cells were subjected to ultracentrifugation (100,000$g$, 25 min) to remove cell debris. The supernatant was filtered and then loaded onto a 5 ml HisTrap FF crude column (Cytiva). The column was washed with 10 column volumes of lysis buffer supplemented with 20 mM imidazole. The fusion protein was eluted from the column using a linear imidazole gradient (20 mM to 160 mM in 10 min). The eluted fusion protein was then dialyzed overnight in G-buffer (5 mM Tris pH 8, 0.2 mM CaCl$_2$, 0.1 mM NaN$_3$, 0.5 mM ATP, 0.5 mM TCEP), followed by incubation with chymotrypsin (1/250 wt/wt ratio of chymotrypsin to actin) for 20 min at 25 °C to cleave thymosin β4 and the deca-His-tag from the β-actin. Proteolysis was terminated by the addition of 0.2 mM PMSF (final concentration), and the mixture was applied to a clean HisTrap FF crude column. The flowthrough was collected and β-actin was polymerized overnight by the addition of 2 mM MgCl$_2$ and 100 mM KCl (final concentrations). The next morning, actin filaments were pelleted by ultracentrifugation (210,000$g$, 2 h) and resuspended in G-buffer. β-actin was depolymerized by dialysis in G-buffer for 2–3 d. Afterwards, the protein was ultracentrifuged (210,000$g$, 2h) to remove any remaining filaments. The supernatant was, if necessary, concentrated in a 10-kDa concentrator (Amicon) to 2–3 mg ml$^{-1}$, flash frozen in liquid nitrogen in 50-µl aliquots, and stored at −80 °C until further use.

Native rabbit α-actin was purified from muscle acetone powder and labeled with pyrene (N-1-pyrene-Iodoacetamide, Sigma). Ten grams of rabbit muscle acetone powder was mixed with 200 ml G-Buffer and stirred for 30 min at 4 °C. Later, the solution was centrifuged at 16,000$g$ in a GSA rotor for 1 h at 4 °C to separate the supernatant containing actin from the debris. Actin polymerization was initiated in the supernatant by adding polymerization buffer (50 mM KCl, 1.5 mM MgCl$_2$, 1 mM EGTA, 10 mM imidazole pH 7) and an additional 0.1 mM ATP. The solution was allowed to polymerize at room temperature for 1 h while being stirred, after which the solution was moved to 4 °C to polymerize for another hour. Later, the concentration of KCl in the solution was increased to 800 mM, and it was stirred for another 30 min. After a total of 2.5 h of polymerization, the solution was centrifuged at 96,000$g$ in a Ti45 rotor for 1 h at 4 °C and the actin pellet was resuspended in F-Buffer (2 mM Tris pH 8.0, 50 mM KCl, 1.5 mM MgCl$_2$, 1 mM EGTA, 10 mM imidazole, 0.1 mM CaCl$_2$, 0.2 mM ATP, 0.01% NaN$_3$, 0.5 mM TCEP). Polymerized actin was reduced by the addition of 1 mM TCEP, and the mixture was centrifuged at 266,000$g$ in a TLA-110 rotor for 30 min at 4 °C. The resulting actin pellet was resuspended in labeling buffer (50 mM KCl, 1.5 mM MgCl$_2$, 1 mM EGTA, 10 mM imidazole pH 7, 0.2 mM ATP), to which pyrene was added in five times molar excess of actin, and incubated on ice for 2 h. The reaction was quenched by the addition of 10 mM DTT and was dialyzed in G-buffer to depolymerize actin.

For the purification of heterodimeric DNase I–actin complexes, deoxyribonucleaseI from bovine pancreas (DNase I, Serva, cat. no. 18535.02) was dissolved at a concentration of 666 µM (20 mg ml$^{-1}$) in 1×KMEI Buffer (0.5 mM ATP, 1 mM TCEP, 50 mM KCl, 1.5 mM MgCl$_2$, 1 mM EGTA, 10 mM imidazole pH 7) containing 1× complete protease inhibitors and 1 mM PMSF. Then, 4 ml of 90 µM filamentous β,γ-actin was mixed with 1.1 ml of 666 µM DNase I, resulting a 1:2 molar ratio, and depolymerized by dialysis for 1 week against G-buffer. The dialyzed sample was centrifuged for 30 min at 80,000 r.p.m. in a TLA-110 rotor, and the supernatant was gel filtered into G-Buffer over a Superdex 200 16/600 column (Cytiva). Fractions corresponding to the heterodimeric

DNase I–actin complex were pooled, concentrated, and stored at 4 °C for up to 3 months.

mDia1$_{FH2}$ was expressed with an amino-terminal 10×His-SUMO3-tag in *Escherichia coli* BL21 Star pRARE cells for 16 h at 18 °C. The cells were lysed in Lysis Buffer-v2 (50 mM NaH$_2$PO$_4$ pH 8.0, 400 mM NaCl, 0.75 mM β-mercaptoethanol, 15 µg ml$^{-1}$ benzamidine, 1× complete protease inhibitors, 1 mM PMSF, DNase I) and the protein was purified by IMAC using a 5-ml HisTrap column. The protein was eluted using Elution Buffer (50 mM NaH$_2$PO$_4$ pH 7.5, 400 mM NaCl, 400 mM imidazole, 0.5 mM β-mercaptoethanol) in a gradient and the 10×His-SUMO3-tag was directly cleaved using SenP2 protease overnight. After cleavage, proteins were desalted into lysis buffer and recirculated over a 5-ml HisTrap column, followed by gel filtration over a Superdex 200 16/600 into Storage Buffer (20 mM HEPES pH 7.5, 200 mM NaCl, 0.5 mM TCEP, 20% glycerol), concentrated, flash frozen in liquid nitrogen, and stored at −80 °C.

Spectrin-actin seeds were purified as described previously[65,66]. The purified spectrin-actin seeds were biotinylated by addition of a fivefold molar excess of maleimide-PEG$_2$-biotin. After incubation for 30 min on ice, biotinylated spectrin-actin seeds were separated from free, unreacted maleimide-PEG$_2$-biotin by two rounds of desalting over NAP-5 columns. Glycerol was added to 50% (vol/vol), and biotinylated spectrin-actin seeds were stored at −20 °C until use.

### Synchronous measurement of actin polymerization and P$_i$ release in bulk assays

On the day of the assay, aliquots of all purified β-actin variants (frozen as G-actin) were thawed and centrifuged at 100,000$g$ for 20–30 min to remove aggregates. To ensure that all variants were in exactly the same buffer, we exchanged the buffer to G-buffer-v2 (5 mM Tris pH 8, 0.2 mM CaCl$_2$, 0.1 mM NaN$_3$, 0.1 mM ATP, 0.5 mM TCEP) using Micro Bio-Spin Chromatography Columns (Bio-Rad). For each measurement, 40 µl G-actin solution containing 30.5 µM unlabeled β-actin variant, 0.5 µM pyrene-labeled, wild-type α-actin, and 15 µM P$_i$-sensor (MDCC-labeled phosphate binding protein, Thermo Fisher Scientific) was prepared. We confirmed that the presence of trace amounts (1.5%) of pyrene-labeled, wild-type α-actin, which releases P$_i$ with slow kinetics, did not substantially contribute to the overall readout by the phosphate sensor, which was dominated by P$_i$ release from the actin mutant present in vast excess (98.5%) (Extended Data Fig. 5a). Thirty-six microliters of G-actin solution was then mixed with 4 µl 10×ME (5 mM EGTA pH 7.5, 1 mM MgCl$_2$) and incubated at room temperature for 2 min, in order to exchange the ATP-associated divalent cation from Ca$^{2+}$ to Mg$^{2+}$. We then took 36 µl of this solution and mixed it with 64 µl polymerization buffer (16 mM HEPES pH 7, 160 mM KCl, 3 mM MgCl$_2$, 1.5 mM EGTA, 38 µM P$_i$-sensor, 160 nM spectrin-actin seeds, 0.1 mg ml$^{-1}$ β-casein, 0.1 mM ATP, 0.5 mM TCEP) in a quartz cuvette to start the experiment. This yielded final concentrations of 10 µM actin (1.5% pyrene-labeled), 100 nM spectrin-actin seeds, 100 mM KCl, and 30 µM P$_i$-sensor. The spectrin-actin seeds were added to ensure rapid polymerization in order to (1) minimize potential differences in the polymerization kinetics between the β-actin variants (Extended Data Fig. 5b) and (2) create a pronounced separation between the time courses of polymerization and P$_i$ release, at least in the case of wild-type actin. Measurements were taken in a spectrofluorometer (PTI QM-6) under constant excitation at 365 nm and synchronous monitoring at 1-s intervals of pyrene (wavelength $\lambda$ = 410 nm) and MDCC (wavelength $\lambda$ = 455 nm) fluorescence intensities, which report on actin polymerization and phosphate release, respectively.

### Determination of P$_i$-release rates from bulk assays

Timecourses of pyrene or MDCC fluorescence from individual experiments were first normalized by subtracting the minimal signal at the beginning of the experiment and dividing by the maximal signal at saturation at $t$ = 900 s. Observed rates of actin polymerization ($k_{poly}$)

were then determined by fitting time courses of normalized pyrene fluorescence from individual experiments by a mono-exponential function in Origin Pro version 9.0G:

$$I(t) = 1 - e^{-k_{poly}t}$$

with $I(t)$ being the normalized fluorescence intensity as a function of time. Average $k_{poly}$ values for each actin variant were calculated from three independent experiments (Extended Data Fig. 5b). To determine $k_{-Pi}$, we first averaged time courses of normalized MDCC fluorescence from three individual experiments for each actin variant. The averaged data were fitted by a simple kinetic model using the KinTeK Explorer software (version 6.3):

$$A \xrightarrow{k_{poly}} B \underset{k_{+Pi}}{\overset{k_{-Pi}}{\rightleftharpoons}} C + P_i$$

with A being monomeric ATP-actin, B being filamentous ADP-P$_i$-actin, and C being filamentous ADP-actin. The model contains three kinetic parameters, two of which were fixed. The first-order rate of polymerization ($k_{poly}$), which formally is the sum of the rates of both polymerization and ATP hydrolysis, was fixed to the experimentally measured polymerization rate for each actin variant (see above). The second-order association rate constant for binding of inorganic phosphate ($k_{+Pi}$) was fixed to 0.000002 µM$^{-1}$ s$^{-1}$, as measured previously for wild-type α-actin and assumed to be the same for all actin variants[11,27]. This assumption likely does not hold, because the chosen substitutions can be anticipated to similarly accelerate both release and binding of P$_i$. However, we determined that $k_{+Pi}$ can be varied by more than 1,000-fold without majorly affecting the obtained first-order rate of phosphate release ($k_{-Pi}$). More importantly, we can exclude the possibility that rebinding of P$_i$ contributes substantially under our experimental conditions: P$_i$ was (1) generated only in minor amounts (10 µM) during the course of the assay and (2) potently sequestered by the phosphate sensor that was present in molar excess (30 µM) and that binds P$_i$ with 10,000-fold-higher affinity (dissociation constant ($K_D$) = 0.1 µM) (ref. 67) than that of actin ($K_D$ = 1.5 mM) (refs. 11,27).

For the N111S mutant, we could not determine the exact rate constant of P$_i$ release in this manner, because the average observed rate of P$_i$ release slightly exceeded the average observed polymerization rate (Fig. 3b). This should formally not be possible because the latter has to precede the former, and the reason for this inversion remains unknown. To nonetheless obtain a conservative estimate for the increase in the P$_i$-release rate in this case, we carried out kinetic simulations in KinTek Explorer to systematically explore the dependence of the observed P$_i$-release reaction kinetics on the rate enhancement of P$_i$ release (Extended Data Fig. 5c,d). This showed that a rate enhancement of P$_i$ release by more than 15-fold is required for the observed P$_i$ release rate to fall within the error margin of the observed polymerization rate. Hence, we consider 0.1 s$^{-1}$ the lower bound for the rate of P$_i$ release for the N111S mutant.

### Preparation of functionalized glass slides

Functionalized glass slides coated with 5% biotin-PEG and 95% hydroxy-PEG were prepared as has been described[68]. Briefly, high-precision glass coverslips (22 × 60 mm, 1.5H) were asymmetrically cut at one corner using a diamond pen to distinguish the functionalized surface from the non-functionalized one. Glass slides were cleaned by incubation in 3 M NaOH solution for 15 min, rinsed in water, and incubated in freshly prepared Piranha solution (3:2 mixture of 95–97% sulfuric acid and 30% hydrogen peroxide) for 30 min. Slides were rinsed with water to remove residual acid and were then air dried using nitrogen gas. Dried glass slides were sandwiched with 3 drops of GOPTS (3-glycidyloxypropyl trimethoxysilane) and then stored in closed Petri dishes, which were further incubated in

an oven at 75 °C for 30 min. Sandwiched glass slides were rinsed in acetone and separated with a pair of tweezers. Following separation, the glass slides were rinsed again in fresh acetone. Separated glass slides were air dried with nitrogen gas and placed in pre-warmed Petri dishes with their functionalized surface facing up. The functionalized surfaces were sandwiched with 75 µL of a 150 mg ml$^{-1}$ mixture of 95% hydroxy-amino-PEG (α-hydroxy-ω-amino PEG-3000) and 5% biotinyl-amino-PEG (α-biotinyl-ω-amino PEG-3000) dissolved in acetone. Sandwiched glass surfaces were incubated in closed Petri dishes in an oven at 75 °C for 4 h. Following incubation, the glass slides were separated with a pair of tweezers, rinsed multiple times in water, and air dried using nitrogen gas.

## Preparation of microfluidic devices
Microfluidic devices were prepared by casting polydimethylsiloxane (PMDS) onto a silicon mold designed to incorporate up to four inlets and one outlet. A mixture of PDMS and its curing agent (10:1 mass ratio, SYLGARD 184 Silicone Elastomer Kit) was thoroughly mixed and poured onto the silicon mold. The PDMS cast was degassed in a desiccator under vacuum for 3 h to remove bubbles from the cast. After removal of bubbles, the cast was incubated in an oven at 75 °C for another 4 h to complete the curing process. On completion of the curing process, the PDMS microfluidic devices were cut from the cast, rinsed with isopropanol, and air dried with nitrogen gas.

The functionalized glass cover slips and the flow surface of the microfluidic devices were plasma cleaned at 0.35 mbar pressure and 80% ambient air for 3 min. Parts of the functionalized glass cover slips that would eventually line up with the observation chamber on the microfluidic device were protected from plasma treatment by placing blocks of PDMS on the region. The microfluidic device was sealed by placing the PDMS block onto the functionalized glass surface and incubating at 75 °C for 1 h.

## Microfluidic experiments and TIRF microscopy of single filaments
The surface of a flow channel was prepared by first passivating the surface to avoid nonspecific interactions, then coating the biotinylated surface with streptavidin, and finally coating the surface with biotinylated spectrin-actin seeds. Channels were passivated by washing with 1× KMEI buffer supplemented with 1% Pluronic F-127, 0.1 mg ml$^{-1}$ β-casein, and 0.1 mg ml$^{-1}$ κ-casein. Passivated surfaces were coated with streptavidin by flowing in 1× KMEI containing 0.75 nM streptavidin. Streptavidin-coated surfaces were then coated with spectrin-actin seeds by washing with 1× KMEI containing 10 nM biotinylated spectrin-actin seeds. Each of the above steps was interleaved by a wash step with in 1× KMEI after each step. All steps were carried out at flowrates of 20 µl min$^{-1}$ for 5 min each, except the streptavidin flow step which was 2 min long. After the surfaces were coated with spectrin seeds, polymerization was started by flowing in assay buffer (0.1 mg ml$^{-1}$ β-casein, 1 mM ATP, 1 mM DABCO, 20 mM β-mercaptoethanol, 100 mM KCl, 1.5 mM MgCl$_2$, 1 mM EGTA, 20 mM HEPES) containing ~2.5 µM profilin:actin (2.71 µM wild-type β-actin or 2.50 µM β-actin-N111S) and 50 nM Alexa-Fluor-488-Lifeact for visualization of filament growth. The profilin:actin amount was varied to equalize minor differences in the polymerization velocity between the two actin variants. Image acquisition was started immediately after the profilin:actin buffer flow was initiated. Depolymerization was initiated after about 3–5 min of polymerization by stopping the flow of profilin:actin and starting the wash with depolymerization buffer (0.1 mg ml$^{-1}$ β-casein, 1 mM ATP, 1 mM DABCO, 20 mM β-mercaptoethanol, 100 mM KCl, 1.5 mM MgCl$_2$, 1 mM EGTA, 20 mM HEPES, 50 nM Alexa-Fluor-488-Lifeact).

Image acquisition was carried out using a TIRF microscope with a ×60 objective and a ×1.5 zoom lens under TIRF conditions. Time-lapse images were acquired every 5 s with 2% laser power, 2-s exposure time,

and an electronic gain setting of 150, and emitted light was filtered through an emission filter of 525 nm with a 50-nm bandpass. All images were acquired at a bit-depth of 16 bits.

## Filament tracking and data analysis
Timelapse TIRF microscopy images of actin filaments were first denoised using the non-local means algorithm implemented with a custom Python script. Lengths of single actin filaments were tracked in time with ImageJ using the JFilament plugin[69] with the following parameters: alpha = 15, beta = 10, gamma = 20,000, weight = 0.5, stretch = 2,000, deform iterations = 200, spacing = 1.0, smoothing = 1.01, curve type = open. Foreground and background values were assigned by selecting regions of the filament and the surrounding background, respectively.

Some of the filaments stopped depolymerizing at different time points during the depolymerization phase; therefore, we restricted our analysis to time scales under which the filament depolymerization was uninterrupted (Extended Data Fig. 5g). Pauses in the depolymerization of single filaments have been reported to be a result of photo-induced dimerization and were extensively characterized in a previous study[70]. Our tracked filaments demonstrate pausing behavior similar to those observed previously, wherein the cumulative distribution of the pause probability fits a sigmoidal model as a function of time[70] (Extended Data Fig. 5g). Filament lengths tracked from different filaments were averaged and used to calculate depolymerization velocity. Instantaneous depolymerization velocities were calculated by performing local linear regression of varying sizes centered around a time point. We found that the combined standard prediction error for the three fit parameters of the kinetic model displays a broad minimum as a function of window size (Extended Data Fig. 5e). Altering the window size within this minimum (sizes 17–29) yields correspondingly smaller or larger delay times in the velocity data, with marginal effects on the obtained fit parameters (Extended Data Fig. 5f). We systematically chose window sizes that reliably predict fit parameters with an average prediction standard deviation error of less than 10% (Extended Data Fig. 5e,f). Instantaneous velocities calculated for each time point were then averaged over different window sizes. An exponential decay function of the form $y = ae^{-bx} + c$ was used to fit $1/v_{depol}$ versus filament age ($\tau$) curves, where:

$$y = \frac{1}{v_{depol}}$$

$$b = k_{-Pi}$$

$$c = \frac{1}{v_{depol,ADP}}$$

$$a + c = \frac{1}{v_{depol,ADP \times Pi}}$$

Filament age during the depolymerization phase was calculated as follows:

$$\tau = t + \frac{L_0 - L_t}{v_{pol}}$$

Where $t$ is the time since depolymerization, $L_0$ is the filament length at the start of depolymerization, $L_t$ is the filament length at time $t$, and $v_{pol}$ is the polymerization velocity.

The barbed-end P$_i$-release rate was calculated as follows:

$$\kappa_{-Pi\,BE} = v_{depol,ADP} \left( \frac{\kappa_{off}^{ADP-Pi} - v_{depol,ADP \times Pi}}{v_{depol,ADP \times Pi} - v_{depol,ADP}} \right)$$

Where $k_{-Pi}$ and $k_{-PiBE}$ are the phosphate-release rates from the interior of the filament and the barbed end, respectively. $v_{depol}$, $v_{depol,ADP}$, and $v_{depol,ADP\times Pi}$ are the observed depolymerization rate, depolymerization rate of ADP-actin subunits, and depolymerization rate of ADP-$P_i$-actin subunits, respectively. ADP-$P_i$-actin subunits can dissociate from the filament in two different ways; either as actin–ADP-$P_i$ with a rate of $\kappa_{off}^{ADP-Pi}$ or first by releasing the $P_i$ at a rate of $k_{-PiBE}$ and then dissociating at a rate of $v_{depol,ADP}$ (ref. 21). For our calculations, we fixed $\kappa_{off}^{ADP-Pi}$ to 0.2 s$^{-1}$, as measured previously[27].

We estimated lower bounds for the $P_i$-release rate of actin-N111S by assuming that $v_{ADP-Pi-depol}$ was either equal to or twice that of wild-type actin. The latter assumption was motivated by the observation that the $v_{depol,ADP}$ rate of the N111S mutant was about 2.2 times that of wild-type actin (Fig. 3d and Extended Data Fig. 5h).

### Preparation of short filaments for cryo-EM
The in vitro polymerization of actin filaments from purified G-actin generally results in long filaments (>500 nm). Cryo-EM imaging of such filaments at high magnification typically yields 0–2 actin ends per micrograph, which does not allow for the averaging of enough end particles to obtain a high-resolution structure. Hence, previous studies have relied on capping protein (CP) as a potent actin nucleator and barbed-end capper to obtain short filaments[45,46]. However, CP binds the F-actin barbed end with high affinity, and we aimed to reconstruct an undecorated barbed end. We therefore designed a new protocol in which we used two other ABPs to create short filaments: DNase I and mDia1$_{FH2}$, which act as pointed-end capper and actin nucleator, respectively. Eighty micromolar β/γ-actin–DNase I complex was mixed with 2 μM free β/γ-actin and 25 μM mDia1$_{FH2}$ (final concentrations) in a total volume of 50 μl G-buffer. The sample was then incubated with 5.6 μl 10× ME on ice for 1 min to exchange the ATP-associated divalent cation in actin from Ca$^{2+}$ to Mg$^{2+}$. Actin polymerization was induced by the addition of 6.2 μl 10× KMEH (100 mM HEPES pH 7.0, 1000 mM KCl, 20 mM MgCl$_2$, 10 mM EGTA), resulting in a salt concentration of 100 mM KCl and 2 mM MgCl$_2$. After the mixture was incubated in a room-temperature water bath for 60 s, we added 1 μl phalloidin (in DMSO) to a final concentration of 80 μM. The sample was placed on ice for another 30 min, and was then injected onto a Superdex 200 increase 5/150 column (Cytiva) pre-equilibrated in 10 mM imidazole pH 7.1, 100 mM KCl, 2 mM MgCl$_2$ and 1 mM EGTA on an ÄKTA Micro system (Cytiva). The eluted filament fractions were collected and directly used for cryo-EM grid preparation within 30 min.

### Cryo-EM sample preparation of actin-R183W and actin-N111S
The recombinant human β-actin filaments with the N111S or R183W substitution were prepared similarly to what has described previously for native rabbit skeletal α-actin filaments[8]. Frozen G-actin aliquots (53 μM of actin-N111S, 58 μM of actin-R183W) in G-buffer were thawed and centrifuged at 100,000g for 20–30 min to remove aggregates. Fifty microliters of G-actin was then mixed with 5.6 μl 10× ME and incubated for 5 min on ice to exchange the ATP-associated divalent cation from Ca$^{2+}$ to Mg$^{2+}$. Actin polymerization was induced by moving the sample to room temperature, followed by the addition of 6.2 μl 10× KMEH, resulting in a final salt concentration of 100 mM KCl and 2 mM MgCl$_2$. The two mutants were polymerized for 1 h at room temperature, followed by 1–2 h on ice. The long incubation period following polymerization ensured that both β-actin variants adopted the 'aged' ADP nucleotide state. The formed filaments were subsequently centrifuged at 200,000g for 30 min, and the F-actin pellet was resuspended in 1× KMEH (10 mM HEPES pH 7.1, 100 mM KCl, 2 mM MgCl$_2$, 1 mM EGTA) supplemented with 0.02% Tween 20 (vol/vol), to a final concentration of ~15 μM F-actin. The resuspended material was used for cryo-EM grid preparation within 1 h.

### Cryo-EM grid preparation
All cryo-EM grids were prepared using the same protocol; 2.8 μl of F-actin sample was applied to a glow-discharged R2/1 Cu 300 mesh holey-carbon grid (Quantifoil). Excess solution was blotted away, and the grids were plunge frozen in liquid ethane or a mixture of liquid ethane and propane using a Vitrobot Mark IV (Thermo Fisher Scientific, operated at 13 °C). The short β/γ-actin filaments, which allowed for the structural determination of the barbed end, were blotted with a blotting force of 0 for 3 s. The long filaments of recombinant β-actin with the N111S or R183W substitution were blotted with a blotting force of −20 for 8 s.

### Cryo-EM grid screening and data collection
The barbed-end dataset was collected on a 200 kV Talos Arctica cryo-microscope (Thermo Fisher Scientific) with a Falcon III direct electron detector (Thermo Fisher Scientific) operated in linear mode. Using EPU (Thermo Fisher Scientific), 1,316 movies were collected at a pixel size of 1.21 Å in 40 frames, with a total electron exposure of ~56 e$^-$ Å$^{-2}$. The defocus values in EPU were set from −2.7 to −1.2 μm.

The β-actin-N111S (9,516 movies) and β-actin-R183W (7,916 movies) datasets were collected on a 300 kV Titan Krios G3 microscope (Thermo Fisher Scientific) equipped with a K3-direct electron detector (Gatan) and a post-column BioQuantum energy filter (Gatan, slit width 15 eV). For both datasets, movies were obtained in super-resolution mode in EPU at a pixel size of 0.3475 Å in 60 frames (total electron exposure of ~70 e$^-$ Å$^{-2}$). The defocus values in EPU were set from −2.0 to −0.7 μm.

### Cryo-EM image processing
All datasets were pre-processed on the fly using TranSPHIRE[71]. Within TranSPHIRE, gain and beam-induced motion correction was performed using UCSF MotionCor2 v1.3.0 (ref. 72). The super-resolution mode collected data (β-actin-N111S and β-actin-R183W datasets) were binned twice during motion correction (resulting pixel size, 0.695 Å). CTFFIND4.13 (ref. 73) was used to estimate the contrast transfer function, and particles or filament segments were picked using SPHIRE-crYOLO[74].

For the β/γ-actin barbed-end dataset, particles were picked using SPHIRE-crYOLO in regular single-particle mode[74], resulting in 252,982 particles. The picked particles were binned four times and extracted in a 96 × 96 box (pixel size, 4.84 Å) in RELION 3.1.0 (ref. 75). The extracted particles were then imported into CryoSPARC v.3.3.2 (ref. 76), which was used for the majority of image processing of this dataset. Importantly, the particles were processed as regular single particles without any applied helical symmetry or restraints. We first performed a 2D classification into 100 classes, thereby removing 11,275 junk particles. The remaining 241,709 particles were subjected to two rounds of heterogeneous refinements. In the first round, we supplied a reference of the pointed end and a reference of a complete filament core. Through this approach, we removed all particles that represented the pointed end (106,675 particles) and kept all particles that were classified into the full filament. For the second round, the remaining particles (135,034 particles) were again heterogeneously refined against two references: a full filament map and a barbed-end map. This allowed for the removal of 72,877 filament core particles and the isolation of 62,157 barbed-end particles. These barbed-end particles were then un-binned (pixel size, 1.21 Å) and subjected to non-uniform refinement, resulting in a map at a resolution of 4.07 Å. We then converted the particles to RELION (using csparc2star.py) for Bayesian polishing[77]. Afterwards, we re-imported them into CryoSPARC and ran iterative 2D classifications and three-dimensional (3D) classifications without image alignment to remove any remaining junk particles. Following one extra round of Bayesian polishing, the final 43,618 barbed-end particles were non-uniformly refined, with the per-particle defocus estimation option switched on, to a reconstruction of 3.51-Å resolution. To further improve the density of subunits at the barbed end, we created a soft mask around the first four actin subunits (from the end) and ran a local refinement in CryoSPARC. The resulting cryo-EM density map was refined to a slightly lower resolution (3.59 Å) but showed an

improved local resolution for the penultimate and ultimate subunits at the barbed end.

For the β-actin-N111S and β-actin-R183W datasets, filament segments were picked using the filament mode of SPHIRE-crYOLO[78], with a box distance of 40 pixels between segments (corresponding to 27.8 Å) and a minimum of six boxes per filament. This yielded 2,001,281 and 1,569,882 filament segments (from now on referred to as particles) for the N111S and R183W datasets, respectively, which were extracted (384 × 384 box) and further processed in helical SPHIREv1.4 (ref. [79]). The processing strategy was essentially the same as that reported in our previous work[8]. Briefly, the extracted particles were first 2D classified using ISAC2 (refs. [80],[81]) in helical mode, and all non-protein picks were discarded. The particles were then refined using meridian alpha, which imposes helical restraints to limit particle shifts to the helical rise (set to 27.5 Å) to prevent particle duplication, but does not apply helical symmetry[71]. For both datasets, the first refinement was performed without a mask using EMD-15109 as the initial reference, low-pass filtered to 25 Å. From the reconstruction that we obtained, a soft mask was created that covered 85% of the filament length within the box (326 pixels in the z direction). Masked meridian alpha refinements then yielded density maps at resolutions of 2.9 Å (N111S dataset) and 3.0 Å (R183W dataset). Subsequently, the particles of both datasets were converted to be readable by RELION through sp_sphire2relion.py. In RELION, the particles were subjected to Bayesian polishing (two times) and CTF refinements[82], followed by a 3D classification without image alignment into eight classes. We selected the particles that classified into high-resolution classes and removed duplicates, which yielded a final set of 1,756,928 and 1,386,604 particles for the N111S and R183W datasets, respectively. Finally, these particles were refined from local searches (sampling 0.9°) with solvent flattening Fourier shell correlations (FSCs) in RELION, yielding reconstructions at resolutions of 2.30 Å (N111S) and 2.28 Å (R183W). RELION was also used to estimate the local resolutions of each density map.

### Model building, refinement and analysis

The barbed-end β/γ-actin structure was modeled as β-actin. Of the four amino acid substitutions between β- and γ-actin, three represent N-terminal amino acids (D2, D3 and D4 in β-actin; E2, E3 and E4 in γ-actin) that are not visible in the density map owing to flexibility. Accordingly, the only other residue (V10 in β-actin, I10 in γ-actin) that is different between both isoforms was modeled as valine. To construct an initial model for human β-actin, chain C of the 2.24-Å cryo-EM structure of rabbit skeletal α-actin in the Mg²⁺-ADP state (PDB 8A2T—94% sequence identity to human β-actin, all water molecules removed) was rigid-body fitted in the third actin subunit from the barbed end (A₂) in the cryo-EM density map using UCSF ChimeraX-1.5 (ref. [83]). Rabbit skeletal α-actin residues were substituted with the corresponding residues of human β-actin in Coot[84]. The resulting model was then also fitted in the densities for the A₀, A₁, and A₃ actin subunits. The model for the cyclic toxin phalloidin was taken from PDB 6T1Y (ref. [39]) and rigid-body fitted into the density map. The barbed structure was iteratively refined by manual model building in Coot and Phenix real-space refine[85] with applied Ramachandran and rotamer restraints. As described in the manuscript, the model of the ultimate A₀ actin subunit had to be substantially altered from the starting models.

The model of recombinant β-actin with the N111S substitution was obtained through a similar approach; chain C of PDB 8A2T (including all water molecules) was fitted into the central actin subunit of the density map. After substitution of all α-actin specific amino acids to the corresponding β-actin residues, introduction of the N111S substitution, and further manual model building in Coot, the resulting model was fitted in four more actin subunits (chains A, B, D, E) in the density map. The filament was modeled as a pentamer to capture the full interaction interface of the central subunit with its four neighboring subunits. All water molecules were first manually built, inspected and adjusted in the central subunit, and were then copied to the other chains with non-crystallographic symmetry (NCS). Because the local resolution was worse at the periphery of the reconstruction, we removed water molecules that displayed poor corresponding cryo-EM density in the non-central actin chains. The model was refined in Phenix real-space refine with NCS restraints but without Ramachandran and rotamer restraints. The model of β-actin-R183W was constructed and refined in the same way, except that the β-actin-N111S structure was used as the starting model for model building. A summary of the refinement quality of the structures is provided in Table 1. Figures depicting cryo-EM density maps and protein structures were prepared using ChimeraX-1.5.

### Preparation of structural models for MD simulations

All-atom models of the core and barbed end of F-actin were prepared from, respectively, PDB 8A2S (F-actin core in the Mg²⁺-ADP-Pᵢ state, ref. [8]) and PDB 8OI6 (the F-actin barbed-end structure presented in this study). Missing residues 46–48 in the D-loop were built de novo with MODELLER v10.2 (ref. [86]). To ensure that simulations could be compared between structural states, the barbed-end structure was back-mutated from human β-actin to human/rabbit α-actin using MODELLER. The missing C-terminal residues (363–375) of the ultimate subunit in the barbed-end structure were modeled using the corresponding resolved residues in the adjacent subunit as template. Phalloidin molecules were removed in the barbed-end model. Hydrogen atoms were added with CHARMM c43b2 (ref. [87]). The Pᵢ ion was modeled in the H₂PO₄⁻ state, using previously established parameters[88]. This is consistent with a recent quantum-mechanical analysis of ATP hydrolysis by F-actin[7], and also promotes sampling of Pᵢ-egress events by reducing the strength of electrostatic interactions[36]. Protonation states of histidine residues were determined with ProPka[89]. Histidine 73 was modeled as a protonated methyl-histidine (meH73⁺). For the F-actin core, a separate model with a neutral methyl-histidine 73 (meH73) was also prepared. Non-histidine titratable residues were modeled in their standard state. For the actin core models, structural water molecules were kept. As the barbed-end structure was determined in the ADP-bound state, we added the Pᵢ ion in the same position relative to ADP as seen in the actin core structure. Structural models were embedded in an orthorhombic simulation box (20 nm × 10.4 nm × 8.8 nm for actin core, 18.8 nm × 11.6 nm × 8.8 nm for barbed end), solvated with TIP3P water molecules, and supplemented with Na⁺ and Cl⁻ ions to ensure electroneutrality and reach a total salt concentration of 150 mM. CHARMM, VMD[90], and scripts obtained from CHARMM-GUI[91] were used for model preparation. Each model was then energy-minimized for 5,000 steps of steepest descent, with harmonic position restraints (force constant 4,184 kJ mol⁻¹ nm⁻²) applied on backbone atoms.

### MD simulation parameters

All simulations were performed with GROMACS 2021.5 (refs. [92],[93]) patched with the February 2022 version of the colvars module[94]. Energetics were described with the CHARMM36m force-field[95]. Van der Waals forces were smoothly switched to zero between 1.0 nm and 1.2 nm. Long-range electrostatics was treated by the particle mesh ewald (PME) method. The cut-off between short-range and long-range electrostatics was initially set to 1.2 nm, and then was adjusted automatically by GROMACS to improve simulation performance. The length of covalent bonds involving hydrogen atoms was constrained with LINCS[96]. The leapfrog integrator, with a 2-fs timestep, was used for molecular dynamics time integration. An average temperature of 310 K was maintained with the v-rescale thermostat[97] (Berendsen for equilibration[98]); an average pressure of 1 bar was maintained with the Parrinello–Rahman barostat[99] (Berendsen for NPT equilibration, that is, at a fixed number of particles N, average pressure P and average temperature T). Minimized systems were equilibrated under active backbone restraints by 1 ns of NVT simulation (that is, at a fixed number of particles N, box volume V and average temperature T) followed by

1 ns of NPT simulation. For production simulations, backbone restraints were turned off, and the filament was kept parallel to the box using a harmonic restraint on the orientation quaternion (force constant $10^5$ kJ mol$^{-1}$), as implemented in colvars. MD trajectories were visualized and analyzed with VMD, Pymol (Schrödinger) and MDAnalysis[100].

## Metadynamics protocol to sample P$_i$ release

To efficiently sample structurally diverse P$_i$-release events from actin, we developed an enhanced sampling protocol based on simulating 'swarms' of short metadynamics trajectories[36,101]. Metadynamics applies a history-dependent potential on a user-defined collective variable to make previously visited configurations more unstable, thus enhancing the sampling. Here, we apply metadynamics on P$_i$ to drive it out of the active site. For this purpose, we concurrently applied separate conventional metadynamics biases on the three Cartesian coordinates $x_P$, $y_P$, and $z_P$ of the phosphorus atom of the P$_i$ ion, expressed in the internal reference frame of the actin filament. The bias was applied only on the P$_i$ ion of the relevant actin subunits, namely the ultimate subunit in the barbed end and the central subunit in the F-actin core. The bias deposition frequency was set to 0.125 ps$^{-1}$, the bias strength to $W = 1$ kJ mol$^{-1}$, and the Gaussian potential s.d. to 0.05 nm. These parameters ensure the capture of P$_i$-escape events. We stress that they are not suited for the evaluation of the converged free-energy profile along a P$_i$-egress path, which was not the purpose of our simulations. Each swarm was composed of 50 independent 10-ns-long metadynamics simulations launched from the same structure. Swarms were analyzed by visual inspection and machine learning.

## Clustering and analysis of P$_i$-egress paths

To get a broad picture of the main P$_i$-egress paths sampled by metadynamics, we used hierarchical clustering in path space. For this purpose, P$_i$ was considered to have escaped when the distance of the P atom from its initial position in the frame of the actin filament exceeded a cut-off value of 1.4 nm, and frames following escape were discarded for the rest of the analysis. We clustered escape paths as follows. First, each P$_i$-egress trajectory (represented by the time-series of the $x_P$, $y_P$, $z_P$ coordinates defined above) was smoothed using B-spline interpolation. Then, 100 equally spaced points were extracted along the fitted splines for each trajectory, resulting in 50 discretized paths in Cartesian space per swarm. Finally, the Euclidean distance matrix between the 50 discretized paths was computed. From this matrix, we performed hierarchical clustering of P$_i$-egress paths. To visualize the paths corresponding to each cluster, we first computed each cluster's centroid path as the cluster average in discretized path space. Then, we used the closest cluster member to the centroid as this cluster's representative path. We found that a user-defined value of four clusters was sufficient to uncover the main sterically accessible P$_i$-egress paths from actin. Spline interpolation was performed with SciPy[102], and clustering was performed with scikit-learn[103]. To complement the agnostic clustering analysis, we also evaluated the fraction of paths within a swarm for which P$_i$ egress took place through the R177-N111 backdoor. We visually inspected the individual P$_i$-egress trajectories and labeled the trajectories in which escape through the R177-N111 backdoor was observed. Escape through the R177-N111 backdoor was defined as P$_i$ passing in between the side chains of R177 and N111, or in their close vicinity.

## Unbiased molecular dynamics of actin core

To obtain insights into the conformational dynamics of actin, a 1.1-μs unbiased MD simulation of the F-actin core (meH73$^+$) was performed. For the central actin subunit, a soft positional restraint (force constant 625 kJ mol nm$^{-2}$) was applied on the coordinates $x_P$, $y_P$, and $z_P$ of the P$_i$ ion of the central subunit to keep it close to the post-hydrolysis position. We analyzed the opening state of the R177-N111 backdoor by measuring the distance between R177CZ and N111CG atoms (N111-R177 distance) along the simulation trajectory.

## Classifier model to detect P$_i$ egress through the R177-N111 backdoor

We trained a logistic regression binary classifier to automatically detect trajectories in which P$_i$ egress happens through the R177-N111 backdoor. Our classifier takes as input a featurized P$_i$-egress trajectory from a metadynamics simulation, and returns 1 if this trajectory is identified as escaping through the R177-N111 backdoor, and returns 0 otherwise. To make the model insensitive to differences in absolute Cartesian coordinates between simulations of different structures, we developed an input trajectory featurization based on the distances of P$_i$ to key residues during egress. First, to deal with escape trajectories of unequal time lengths, we computed for each trajectory the longitudinal path collective variable $s$ (pathCV[104]) ranging from 0 (P$_i$ in the post-hydrolysis position) to 1 (P$_i$ escapes). Thus, $s$ provides a common rescaled time that enables the comparison of trajectories of different lengths. The pathCV calculation was performed with an in-house Python script. The $\lambda$ parameter was set to 325 nm$^{-2}$, and the reference path was obtained by B-spline interpolation of the $x_P$, $y_P$, and $z_P$ time-series, and then was discretized into 30 equally spaced points. For each simulation frame of a given P$_i$-egress trajectory, we then computed $s$, along with distances between the P atom of P$_i$ and the CA atom of 48 residues involved in typical egress pathways detected in the clustering and/or visual analyses (residues 12–18, 71–74, 106–116, 120, 134–138, 141, 154–161, 177, 183, 300–301, 334–338, 370, 374). Distances were turned into Boolean contact values using a distance cut-off of 0.75 nm. Then, contact values along egress paths were grouped into four bins according to the pathCV $s$ value, and the average P$_i$ contact occupancies per bin and per residue were computed. This procedure yielded a P$_i$ contact map along discretized rescaled time, which has the same 48 × 4 size for every P$_i$-egress trajectory regardless of the actual time to P$_i$ escape. Each contact map was finally flattened into a column vector whose entries were used as input features for the classifier. Training of the classifier was performed using a manually labeled data set of 300 P$_i$-egress trajectories, including 70 positives. This data set was randomly split into training and test sets in a 75%/25% proportion using class balancing to preserve the proportion of true positives. The classifier was trained on the training set, and the confusion matrix was evaluated on the test set. scikit-learn was used for these tasks. Out of 75 labeled trajectories in the test set, the classifier predicted 52 true negatives, 6 false positives, 16 true positives and 1 false negative.

## Steered MD simulations

To simulate the opening of the R177-N111 backdoor from the filament interior, we used SMD to drive the central actin core subunit towards the barbed-end configuration and away from the actin core configuration. For this purpose, we applied a harmonic restraint (force constant 334,124 kJ mol$^{-1}$ nm$^{-2}$ or 800 kcal mol$^{-1}$ Å$^{-2}$) moving with constant velocity on the Δr.m.s. deviation (Δr.m.s.d.), that is, the difference in r.m.s.d. between actin core and barbed-end structures. The biasing potential $U$ acting on atomic configuration $x$ at time $t$ was:

$$U(x, t) = \frac{1}{2}k \left( \text{r.m.s.d}(x, \text{actin core}) - \text{r.m.s.d}(x, \text{barbed end}) \right.$$
$$\left. - \text{r.m.s.d}(\text{actin core, barbed end}) - vt \right)^2$$

with $v = 0.0022$ nm ns$^{-1}$ and where r.m.s.d.$(x$, actin core) refers to the r.m.s.d. of atomic configuration $x$ with respect to the actin core structure, r.m.s.d.$(x$, barbed end) refers to the r.m.s.d. of atomic configuration $x$ with respect to the barbed-end structure and r.m.s.d. (actin core, barbed end) refers to the r.m.s.d. of the actin core structure with respect to the barbed end structure. R.m.s.d. values were computed on the Cα atoms of residues 8–21, 29–32, 35–38, 53–68, 71–76, 79–93, 103–126, 131–145, 150–166, 169–178, 182–196, 203–216, 223–232, 238–242, 246–250, 252–262, 274–284, 287–295, 297–300, 308–320, 326–332, 338–348, 350–355, 359–365 and 367–373. To mimic the influence of a long actin

filament in cellular conditions, static r.m.s.d. restraints (force constant 4,184 kJ mol$^{-1}$ nm$^{-2}$) were applied separately on the C$\alpha$ atoms of two pairs of lateral actin subunits flanking the central subunit. To decouple backdoor opening from P$_i$ movement, a positional harmonic restraint was applied on the P atom Cartesian coordinates $x_P$, $y_P$, and $z_P$ (force constant 625 kJ mol$^{-1}$ nm$^{-2}$). Forty SMD simulations were run (20 with meH73$^+$ and 20 with meH73). The following structural observables were computed along each trajectory: N111-R177 distance (see above), sensor/Pro-rich loops distance (defined as the distance between the centers of geometry of C$\alpha$ atoms of sensor-loop residues 70–75, and C$\alpha$ atoms of Pro-rich loop residues 105–115), H161-Q137 distance (defined as the distance between H161-atom NE2 and Q137-atom OE1) and the H161 side chain torsion angle ($\chi_1$). To evaluate the capacity of structures sampled by SMD to release P$_i$ through the R177-N111 backdoor, frames were extracted at $t = 0$ ns (equilibrated structure), $t = 15$ ns, 25 ns, 35 ns, 45 ns, 50 ns, 65 ns, 75 ns, 85 ns, and 100 ns. Then, each of these frames was used to launch a swarm of 50 metadynamics simulations to probe P$_i$ release according to the protocol described above. A static $\Delta$r.m.s.d. restraint with the same force constant and reference value as the SMD potential was applied to preserve the configuration of the actin core subunit. For each swarm, all trajectories were analyzed with the logistic regression classifier. Finally, the proportion $p_{BD}$ of trajectories for which backdoor escape through R177-N111 was predicted was computed for each swarm.

### Reporting summary

Further information on research design is available in the Nature Portfolio Reporting Summary linked to this article.

## Data availability

The cryo-EM maps generated in this study have been deposited in the Electron Microscopy Data Bank (EMDB) under accession codes (dataset in brackets): EMD-16887 (β/γ-actin barbed end), EMD-16888 (F-actin-R183W) and EMD-16889 (F-actin-N111S). These depositions include sharpened and unsharpened maps, unfiltered half-maps and the masks used for refinements. The associated protein models have been deposited in the Protein Data Bank (PDB) with accession codes 8OI6 (β/γ-actin barbed end), 8OI8 (F-actin-R183W) and 8OID (F-actin-N111S). The following previously published protein models were used for data analysis and comparisons: 8A2S, 8A2T and 2V52. We used EMD-15109 as a 3D model for the first refinements of F-actin-R183W and F-actin-N111S. The sequence of human β-actin (P60709, ACTB_HUMAN) was retrieved from UniProt. MD simulation models and protocols, MD simulation datasets, and Jupyter notebooks to reproduce the analyses reported in Extended Data Figs. 4, 9 and 10 have been deposited in Zenodo (https://doi.org/10.5281/zenodo.7765025). All other materials are available from the corresponding authors upon request. Source data are provided with this paper.

## Code availability

All scripts used for data analysis of single-filament assays can be retrieved from https://github.com/iamankitroy/Actin-Pi-Release.

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

## Acknowledgements

We thank S. Bergbrede for overall wet-lab support and for assistance with the purification of actin mutants; J. Funk and F. Merino for initial experiments; T. Wagner for assistance with data analysis; N. Schmidt for assistance with fluorescence microscopy; and S. Pospich and R. S. Goody for valuable discussions and critical proofreading of the manuscript. P.B. thanks P. Bastiaens for continuous support. This work was supported by funds from the Max Planck Society (to G.H., P.B and S.R.), the German Research Foundation (DFG, grant no. BI 1998/2-1 to P.B.) and the European Research Council under the European Union's Horizon 2020 Programme (ERC-2019-SyG, grant no. 856118 to S.R). A.B. is supported by an EMBO long-term fellowship. W.O. and M.B.S. are supported by a postdoctoral fellowship from the Alexander von Humboldt foundation.

## Author contributions

W.O., G.H., P.B. and S.R. conceived the project. A.R., P.B. and W.O. performed protein purifications and biochemical assays. W.O. and P.B. prepared samples for cryo-EM. W.O. and O.H. collected and W.O. processed cryo-EM data and built the protein models. F.E.C.B. carried out all molecular dynamics simulations. A.B. and M.B.S. performed all experiments that involved yeast. G.H., P.B. and S.R. supervised the project. W.O., F.E.C.B., P.B. and S.R. wrote the manuscript, with critical input from all authors.

## FundingInformation

## Competing interests

The authors declare no competing interests.

## Additional information

**Extended data** is available for this paper at https://doi.org/10.1038/s41594-023-01101-9.

**Correspondence and requests for materials** should be addressed to Gerhard Hummer, Peter Bieling or Stefan Raunser.

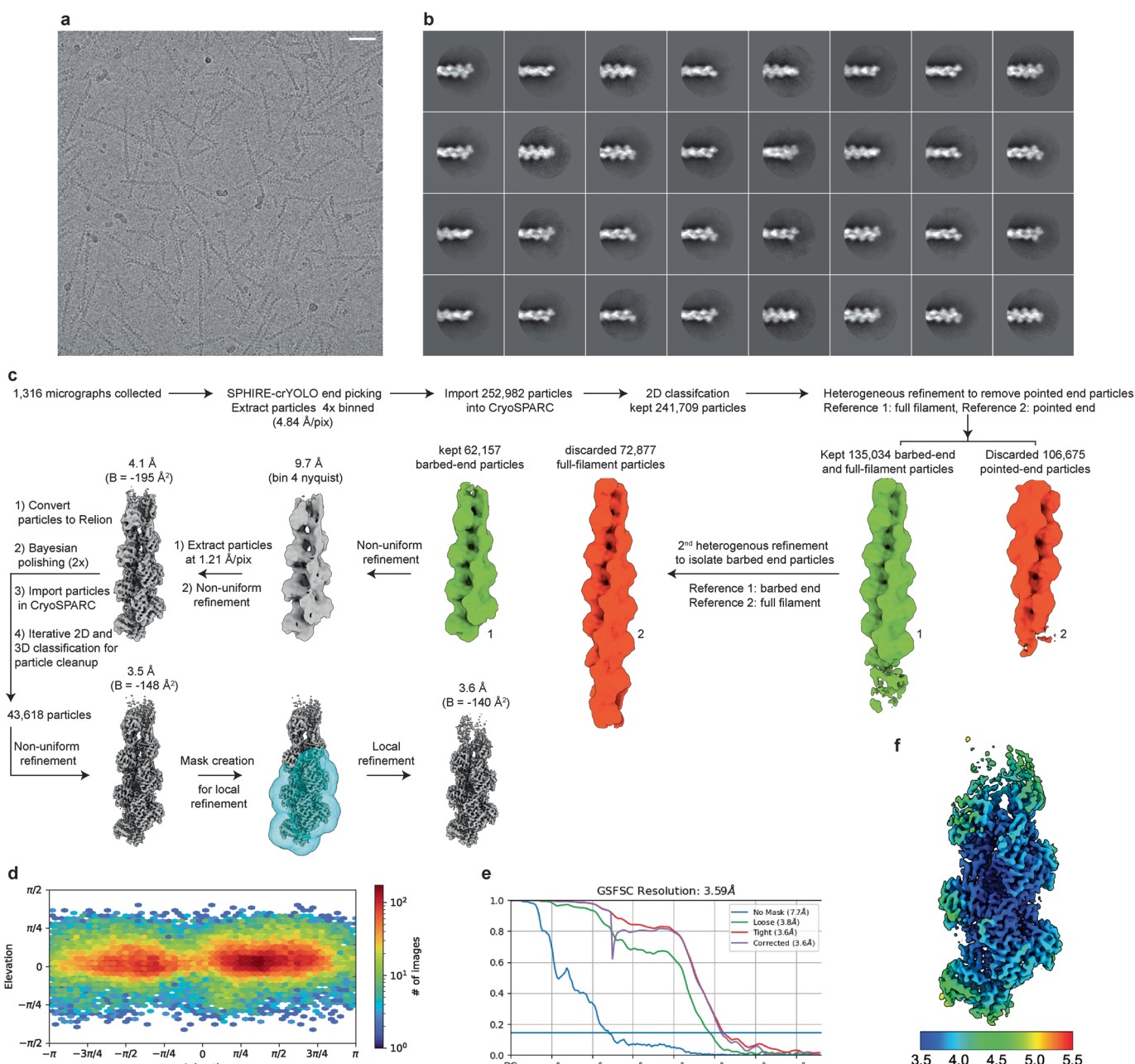

**Extended Data Fig. 1 | Cryo-EM image-processing of the F-actin barbed end. a,** Micrograph depicting short β/γ-actin filaments frozen in vitreous ice, at a defocus of −2.2 μm. The shown micrograph is an example image from a total dataset of 1,316 micrographs. The scale bar is 400 Å. **b**, Exemplary 2D-class averages of the F-actin barbed end reconstruction, generated by RELION. The box size is 465 × 465 Å². **c,** Image processing strategy that was employed to determine the β/γ-actin barbed end structure. All maps are shown in the same orientation. **d,** Angular distribution of the barbed end particles used to reconstruct the final cryo-EM map, generated by CryoSPARC. **e,** Fourier-shell correlation plots for the barbed structure of gold-standard refined half-maps, computed by CryoSPARC. The FSC = 0.143 threshold is annotated. **f,** Local-resolution estimation of the barbed end reconstruction, computed through RELION.

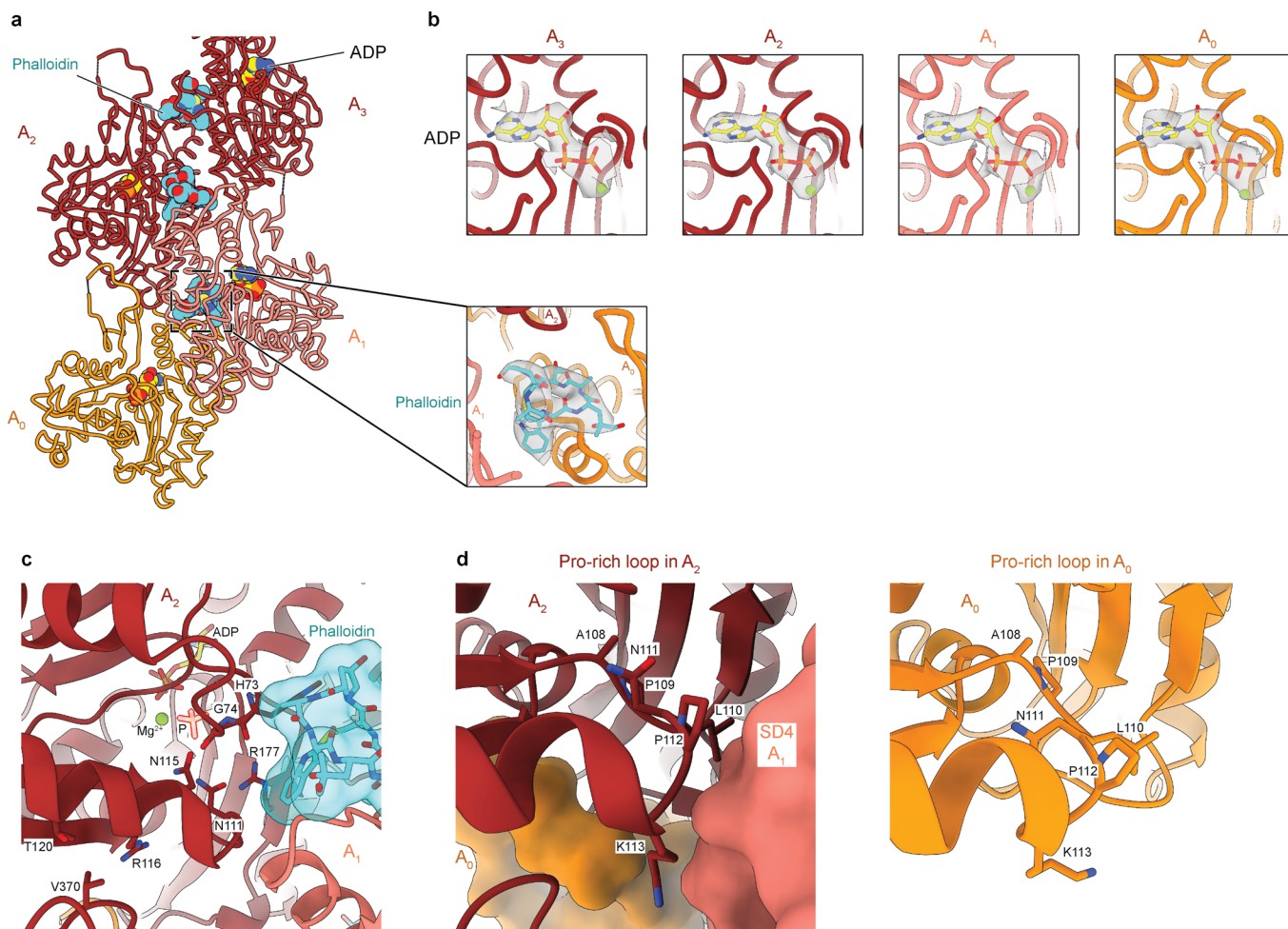

**Extended Data Fig. 2 | Small-molecule binding sites and Pro-rich loop arrangement in the F-actin barbed end structure. a,** Simplified cartoon representation of the barbed end structure. Actin subunits are annotated. ADP and phalloidin are shown as spheres with carbon atoms colored cyan and yellow, respectively. **b,** Cryo-EM densities of the ADP and phalloidin binding sites with fitted models. F-actin is shown as simplified cartoon. **c,** Zoom of the phalloidin-binding site near the R177-N111 backdoor in actin subunit $A_2$. Phalloidin is shown in stick representation with a semi-transparent surface. Residues of the proposed backdoor (R177, N111, H73, G74) are annotated, as well as other residues (N115/ R116, T120/V370) that may represent other $P_i$ escape sites. Phalloidin would only interfere with disruption of the classical backdoor. **d,** Arrangement of the Pro-rich loop in subunits $A_2$ (left panel) and $A_0$ (right panel) in the barbed end structure. In subunit $A_2$, the Pro-rich loop conformation is stabilized by SD4 of subunit $A_1$. Conversely, the Pro-rich loop of subunit $A_0$ does not interact with other subunits.

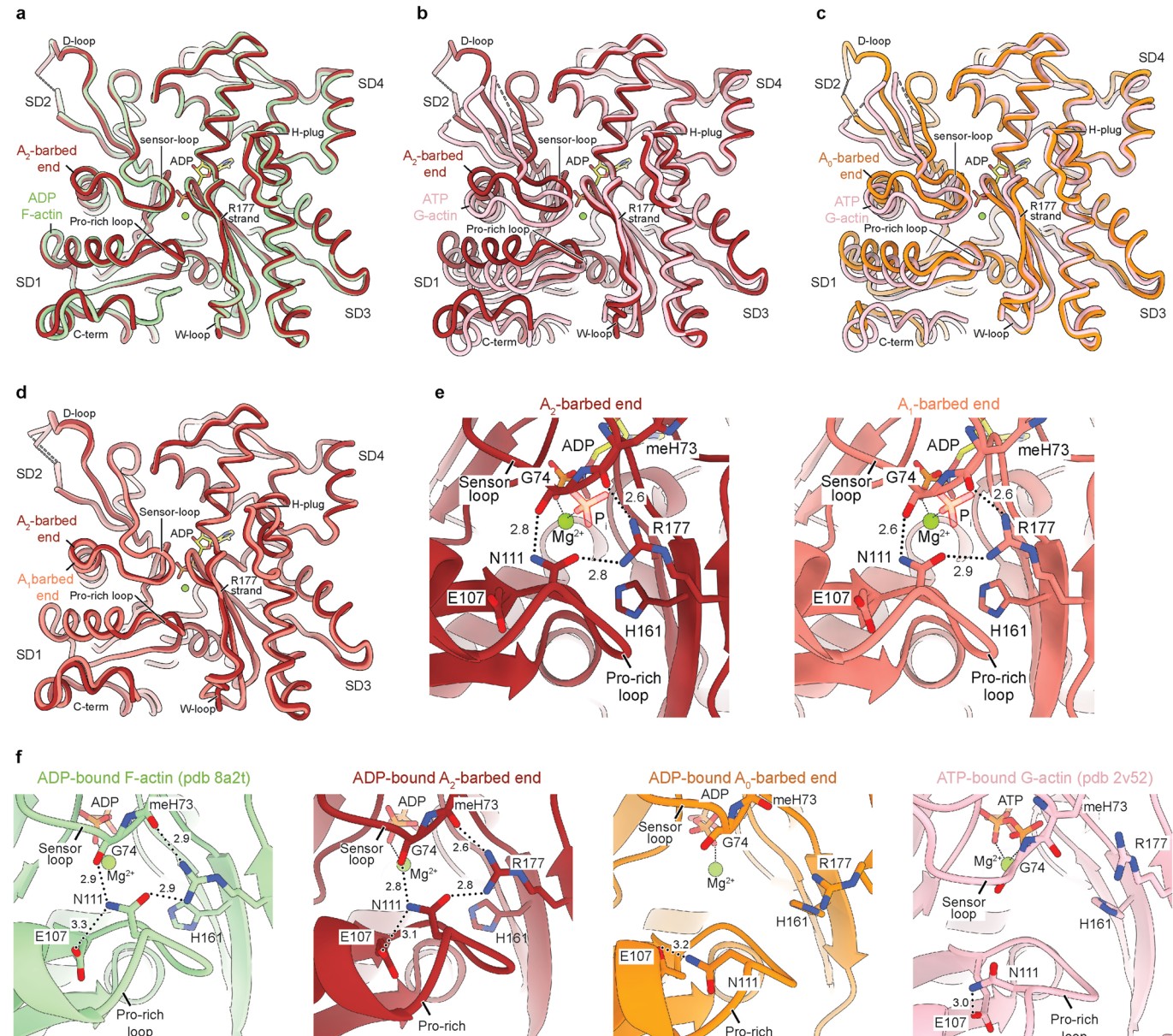

**Extended Data Fig. 3 | Alignment of barbed-end subunit structures with previously determined actin structures. a,** Superimposition of the $A_2$ subunit of the barbed end structure (colored dark-red) with the structure of ADP-bound filamentous α-actin (pdb 8a2t, colored green). **b,** Superimposition of the $A_2$ subunit of the barbed end structure with the structure of ATP-bound monomeric α-actin (pdb 2v52, colored pink). **c,** Superimposition of the $A_0$ subunit of the barbed end structure (colored orange) with the structure of ATP-bound monomeric α-actin (pdb 2v52, colored pink). **d,** Superimposition of the $A_2$ (colored dark-red) and $A_1$ (colored salmon) subunits of the barbed end structure. The only observed change in penultimate subunit $A_1$ is a small rearrangement of the W-loop. This change in the W-loop is expected, because this region would interact with the D-loop of the missing subunit. **e,** Zoom of the R177-N111S backdoor as seen from the F-actin exterior for the $A_2$ and $A_1$ subunits of the barbed end structure. Amino acids that form the backdoor are annotated. Phalloidin is hidden for clarity. Although the subunits adopt the ADP state, $P_i$ from pdb 8a2s (F-actin in the $Mg^{2+}$-ADP-$P_i$ state) is shown semi-transparently to emphasize the $P_i$-binding site. **f,** Architecture of the $P_i$-release backdoor in structures of ADP-bound F-actin (left), the ADP-bound $A_2$ subunit at the F-actin barbed end (middle-left), the ADP-bound $A_0$ subunit at the F-actin barbed end (middle-right) and ATP-bound G-actin (right).

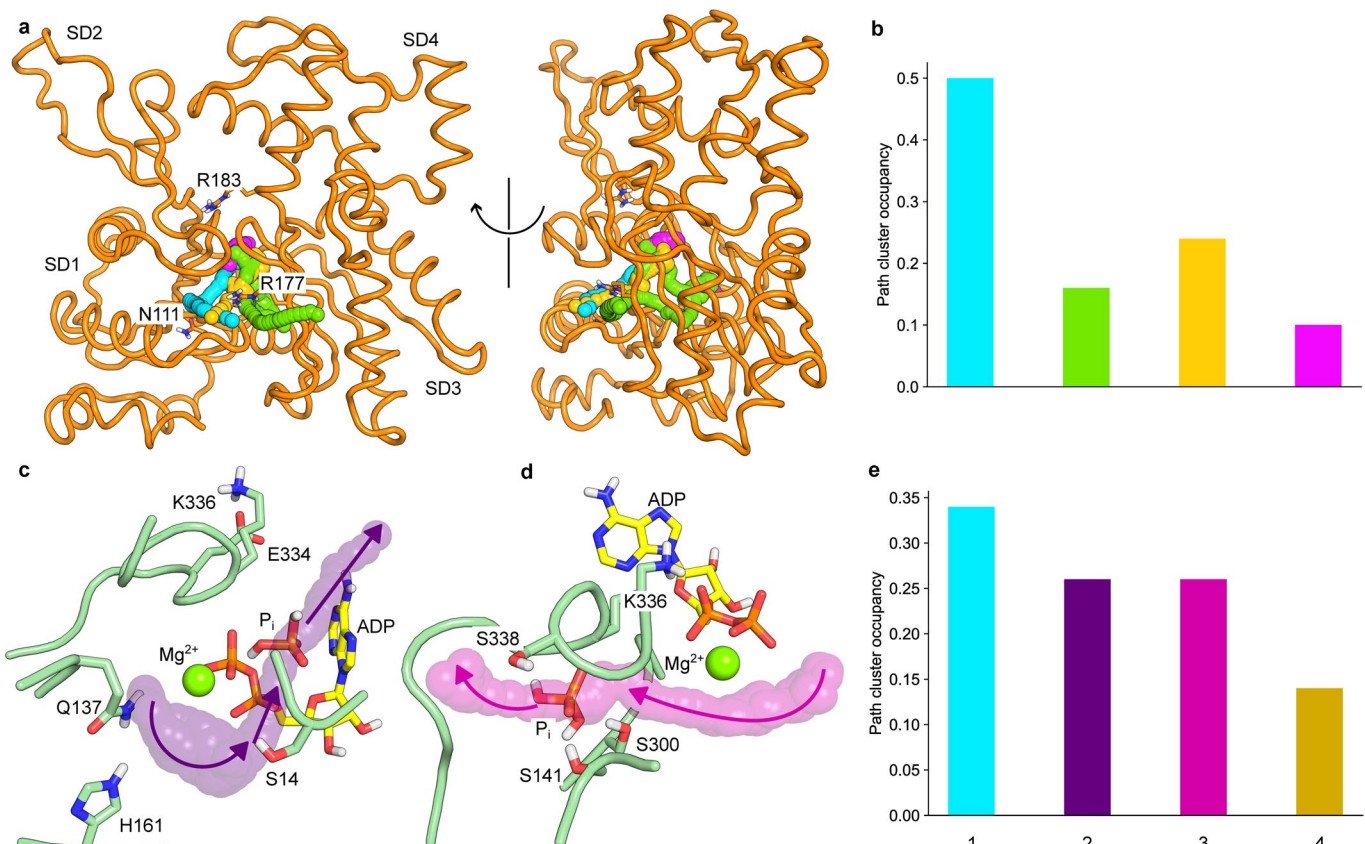

**Extended Data Fig. 4 | Enhanced sampling simulations of P$_i$ release from the F-actin barbed end and core. a,** Representative egress paths from the barbed end show a clear preference for egress through the open R177-N111 backdoor. The F-actin structure fluctuated during the simulations but is shown in a single, representative conformation for clarity. **b,** Path cluster occupancies in barbed end simulations with the same color code as in panel **a**. **c,** A representative path of P$_i$ egress through pathway 2 (purple) in F-actin core simulations. The simulation revealed a highly bent ADP conformer that allowed P$_i$ to escape. Such a drastic rearrangement of the nucleotide binding pocket is physically implausible.

In addition, this egress path would not be directly affected by phalloidin and jasplakinolide binding. Hence, this pathway was not considered for further experimental validation. **d,** A representative path of P$_i$ egress through pathway 3 (magenta) in F-actin core simulations. This escape would also not be directly affected by phalloidin and jasplakinolide binding. Hence, it was not considered for further experimental validation. Similar to panel a, the F-actin structure is shown in a single, representative conformation for clarity. **e,** Path cluster occupancies in actin core simulations with the same color code as in Fig. 2. Pathways 1 and 4 are further discussed in the main text.

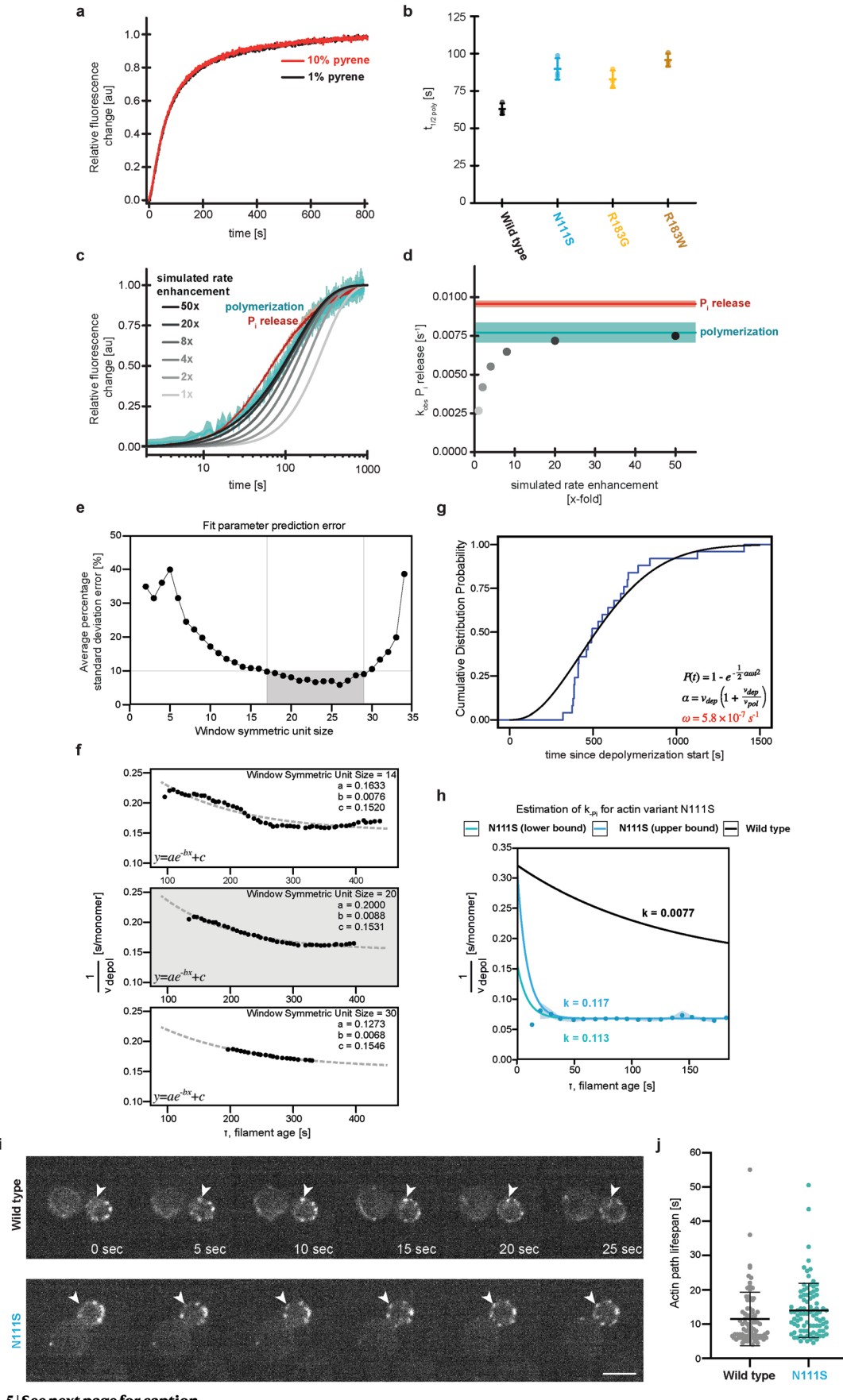

**Extended Data Fig. 5 | See next page for caption.**

**Extended Data Fig. 5 | Kinetic analysis of $P_i$ release from actin filaments and dynamics of actin variants in yeast. a,** Time courses of the normalized fluorescence intensity of 30 μM MDCC-PBP ($P_i$ sensor) from 10 μM N111S actin containing either 1% (black) or 10% (red) wild-type, pyrene α-actin, seeded with 100 nM spectrin-actin seeds after initiation of polymerization (t = 0s). **b,** Characteristic half-times of polymerization for indicated actin variants as determined from mono-exponential fits to the observed polymerization time-courses (Fig. 3b). Dots are values from individual experiments, whereas lines represent the mean from three experiments. Error bars are SD. **c,** Semi-logarithmic plot of simulated $P_i$ release reaction kinetics (gray to black) depending on the enhancement of $P_i$ release rate constant (x-fold over wild-type actin as indicated) compared to the observed time courses of polymerization (cyan) and of $P_i$ release (red) for N111S actin. **d,** Apparent rates of $P_i$ release, obtained from mono-exponential fits of the simulated data shown in c as a function of the enhancement of $P_i$ release rate constant (x-fold over wild-type actin as indicated). The cyan and red areas indicate the observed apparent rates of polymerization (cyan) and $P_i$ release (red) with the center of the error bands (dark colors) being the average from three experiments and the error bands (light colors) representing SD. Note that an enhancement of the $P_i$ release rate constant by at least 15-fold is required for the apparent rate of $P_i$ release to fall within the error margin of the observed polymerization rate. **e,** Effect of varying window sizes on the average prediction error of the three fit parameters from the kinetic model. The average prediction error is large (>10%) for small and large window sizes with a shallow optimum in between (grey shaded region, size 17 to 29). At smaller window sizes the prediction error is large as a result of a higher variance in calculated velocities and for larger window sizes the prediction error increases with decrease in the number of data points available for fitting. **f,** Exemplary fits (grey dashed lines) calculated from velocities (black points) obtained with different window sizes from within (shaded grey, size 20) and flanking (size 14

and 30) the optimum. The exponential decay function used for these fits and their fit parameters are provided in the bottom-left and top-right parts of each sub-plot respectively. **g,** Cumulative distribution probability of pauses as a function of time from the onset of depolymerization. Blue line indicates the fraction of tracked filaments that have paused at any given time and the black line is obtained by fitting an analytic model of pause probability (described previously in ref. 70) as a function of time. The model is described by the equations in the bottom-right corner of the plot. From this fit we obtained for the fit parameter $\omega$ (protomer transition rate) a value of $5.8 \times 10^{-7}$ $s^{-1}$. **h,** Estimation of lower bounds for the phosphate release rate of N111S mutant. Rates were calculated by fitting exponential decay functions to the observed depolymerization velocity data, so that the observed velocities converged to the mutant's $v_{depol,ADP}$ and the $v_{depol,ADP-Pi}$ rate was either assumed to be equal (high estimate, blue) or twice that of the wild type (low estimate, turquoise). The latter assumption was motivated by the observation that the $v_{depol,ADP}$ rate of N111S mutant was about 2.2 times that of wild-type actin (Fig. 5d). **i,** Time-lapse series of confocal microscopy images showing Lifeact-mCherry fluorescence every 5 seconds for actin path detection in yeast cells expressing either wild-type or N111S-actin. The white arrow highlights an exemplary dynamic actin path. The scale bar is 4 μm. The shown images of yeast cells are example images from a total of eight cells per strain. Four independent colonies per strain were imaged. The full experiment was performed in duplicate, obtaining similar results. **j,** Plot of endocytic actin patch lifespans per strain. Each color dot represents one patch. The average lifespans are 12 s ± 7 s (wild-type actin) and 14 s ± 7 s (N111S-actin). Error bars correspond to mean ± standard deviation. Sample size corresponds to n = 86 patches for WT and n = 89 patches for N111S, from a total of eight cells per strain from four independent colonies. There are no large differences in actin-patch lifespans between the two yeast strains.

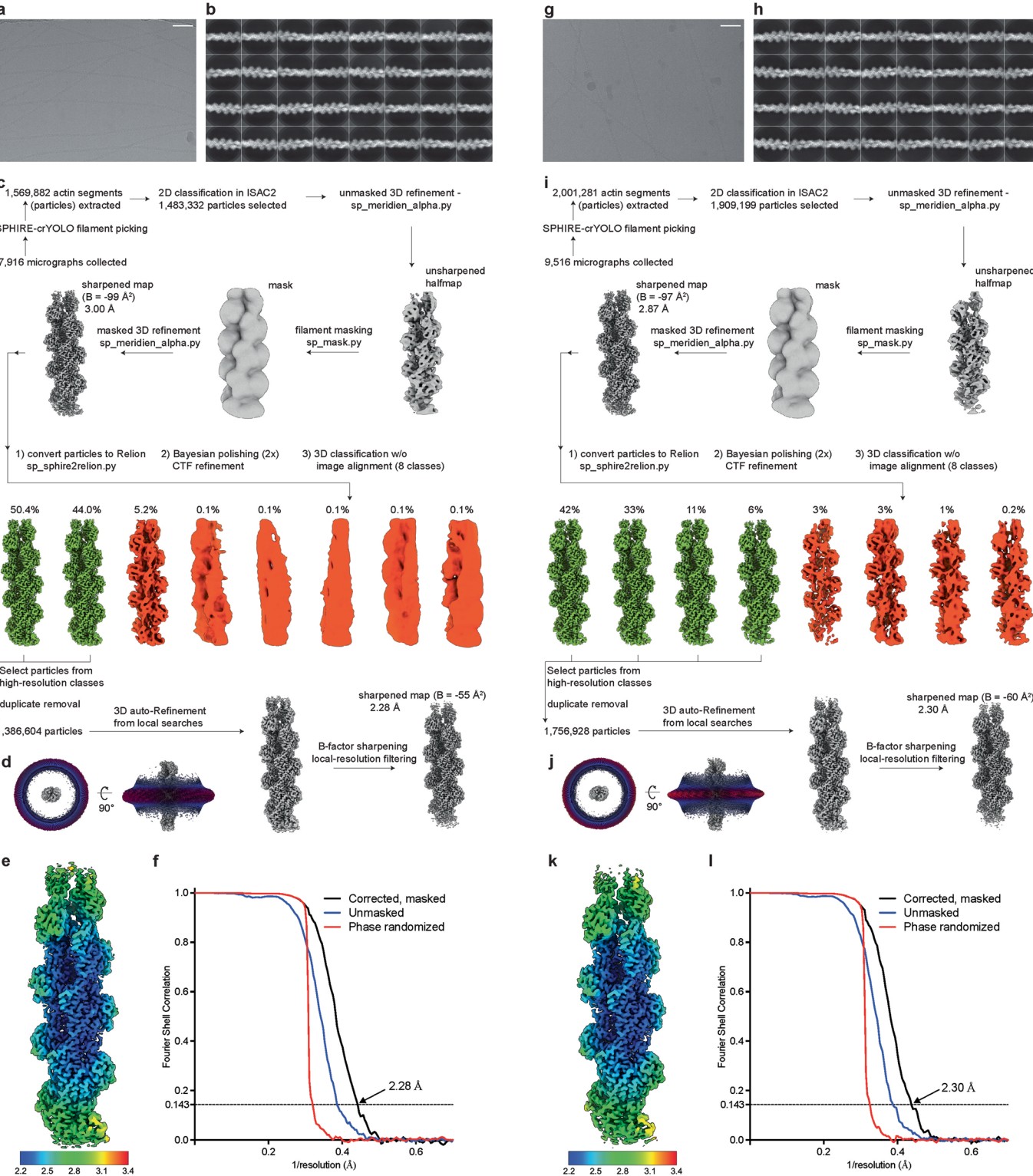

**Extended Data Fig. 6 | Processing of the F-actin-R183W and F-actin-N111S datasets. a, g,** Micrographs depicting R183W (**a**) and N111S (**g**) variants of F-actin frozen in vitreous ice, at, respectively, defocus values of −2.3 μm and −1.9 μm. The shown micrographs are example images from total datasets of 7,916 (R183W) and 9,516 (N111S) micrographs. The scale bars are 400 Å. **b, h,** Exemplary 2D-class averages of the R183W- (**b**) and N111S-F-actin (**h**) particles, computed through RELION. The box size is 267 × 267 Å². **c, i,** Image processing strategies that were employed to determine the actin-R183W (**c**) and actin-N111S (**i**) structures. All maps are shown in the same orientation. **d, j,** Angular distribution of the particles used to reconstruct the final cryo-EM maps of F-actin-R183W (**d**) and F-actin-N111S (**j**), shown along the filament axis (left) and orthogonal to the filament axis (right). **e, k,** Local-resolution estimations of the R183W- (**e**) and N111S-F-actin (**k**) density maps, calculated by RELION. The bar depicts local resolution in Å. **f, l,** Fourier-shell correlation plots for gold-standard refined masked (black), unmasked (blue) and high-resolution phase randomized (red) half-maps of F-actin-R183W (**f**) and F-actin-N111S particles (**l**). The FSC = 0.143 threshold is shown as a dashed line.

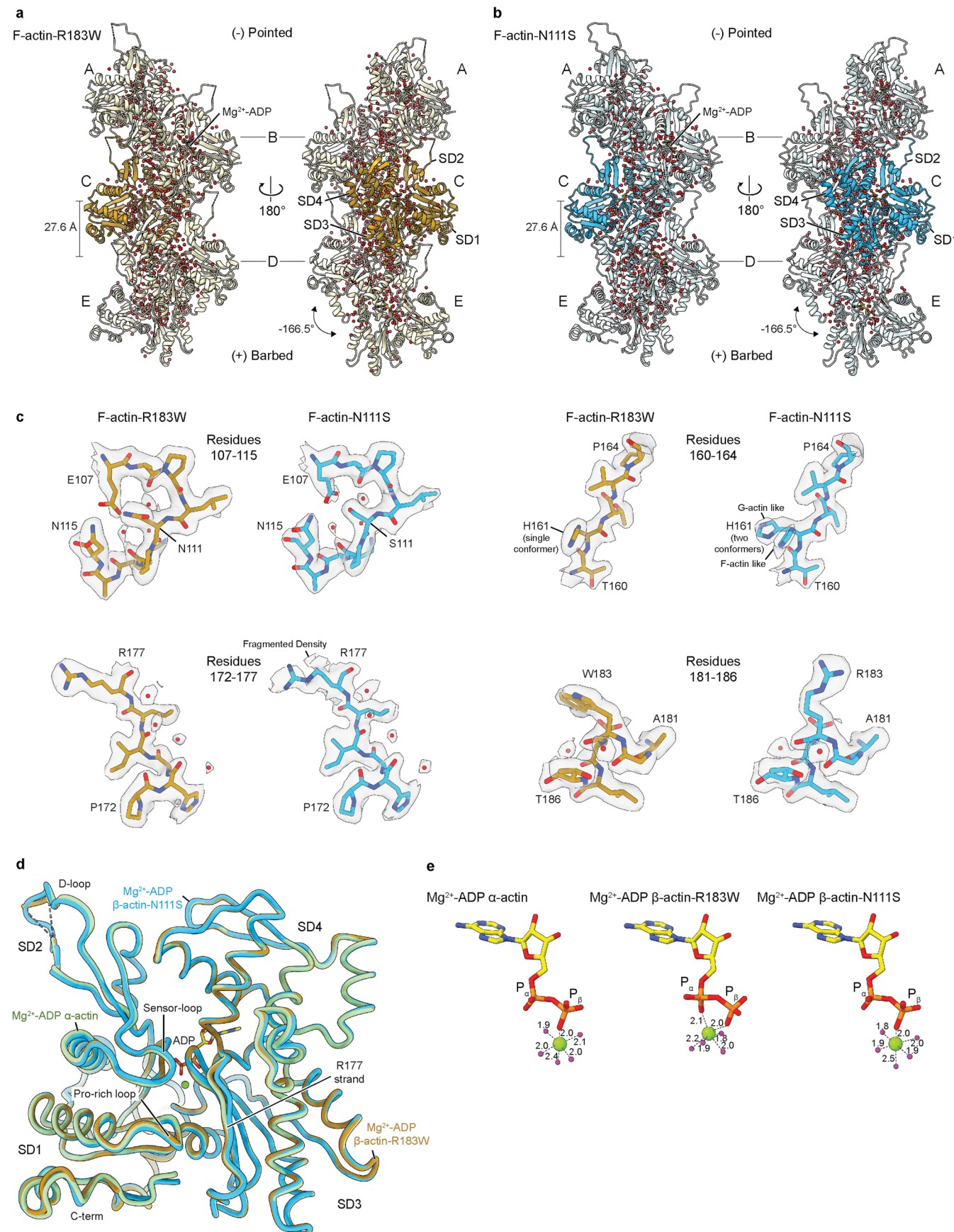

**Extended Data Fig. 7 | See next page for caption.**

**Extended Data Fig. 7 | High-resolution structures of β-actin variants R183W and N111S. a, b,** Structures of F-actin-R183W (**a**) and F-actin-N111S (**b**) shown as cartoon orthogonal to the filament axis. The central subunits are colored gold and blue, respectively. Water molecules modeled in both structures are depicted as red spheres. The pointed and barbed end directions are annotated, as well as the helical rise and twist of both mutant structures. **c**, Cryo-EM densities and fitted atomic models for selected regions of both reconstructions. Specifically, amino-acid environments near the mutated residues are shown to highlight differences between both structures. **d**, Alignment of single subunits of filamentous wild-type α-actin (pdb 8a2t, colored green), R183W-β-actin (gold) and N111S-β-actin (blue) in the $Mg^{2+}$-ADP state. The F-actin subdomains and regions important for $P_i$ release are annotated. **e,** Nucleotide arrangement in high-resolution $Mg^{2+}$-ADP-bound F-actin structures of wild-type α-actin (pdb 8a2t, left), β-actin-R183W (middle) and β-actin-N111S (right). The arrangements in wild-type α-actin and β-actin-N111S are similar, whereas the $Mg^{2+}$ ion adopts a different position in the β-actin-R183W structure.

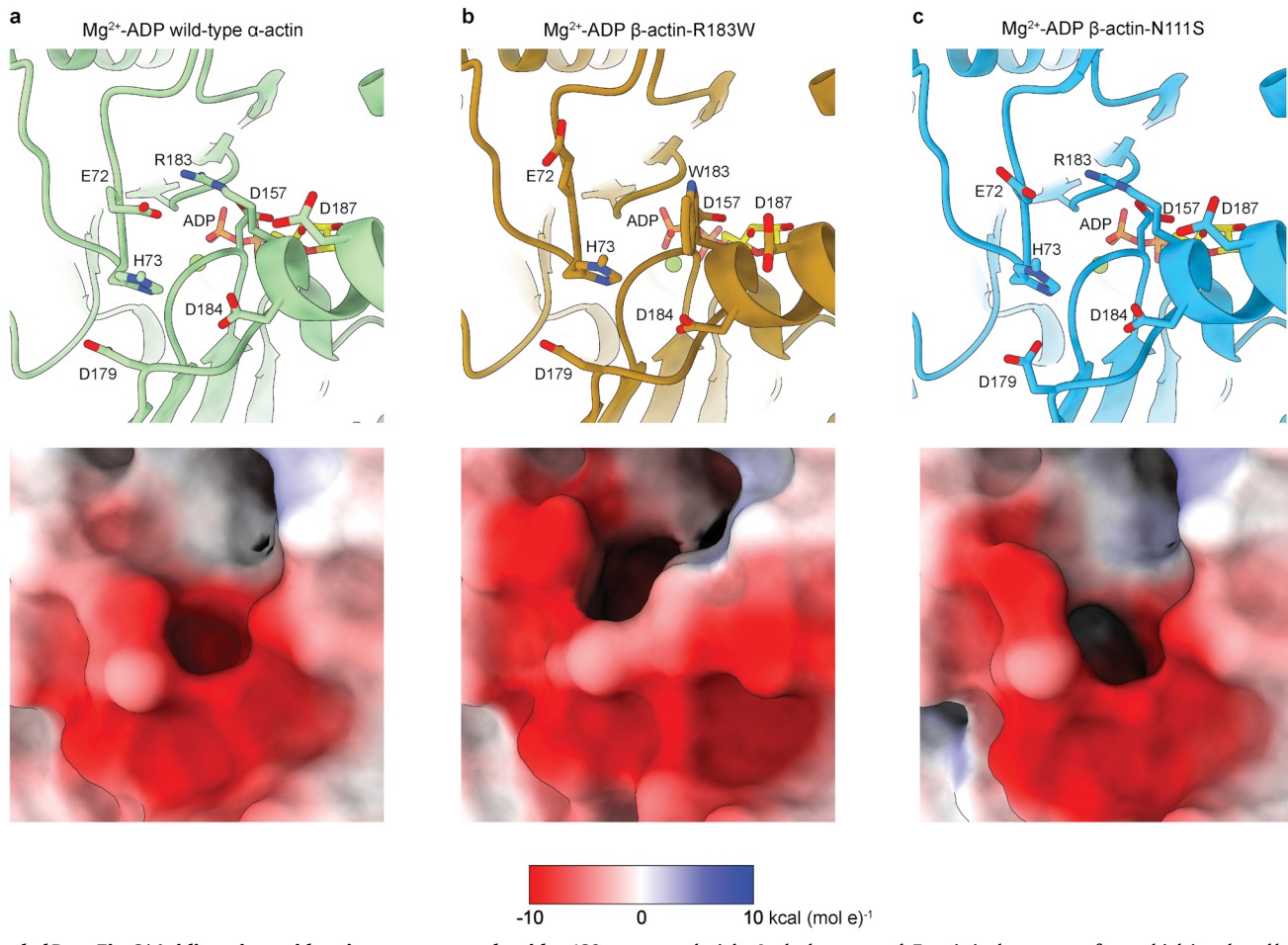

**Extended Data Fig. 8 | Acidic amino-acid environment around residue 183.**
**a–c,** Arrangement near residue 183, as show from the filament exterior for
filamentous wild-type α-actin **(a)** (pdb 8a2t, colored green), β-actin-R183W
(gold) **(b)** and β-actin-N111S (blue) **(c)** in the Mg²⁺-ADP state. The top image
depicts F-actin as cartoon and charged amino acids near residue-183 as cartoon

and sticks. In the lower panel, F-actin is shown as surface, which is colored by
electrostatic Coulomb potential ranging from −10 kcal (mol e)⁻¹ (red) to +10 kcal
(mol e)⁻¹ (blue). The surface is more negatively charged in the F-actin-R183W
structure.

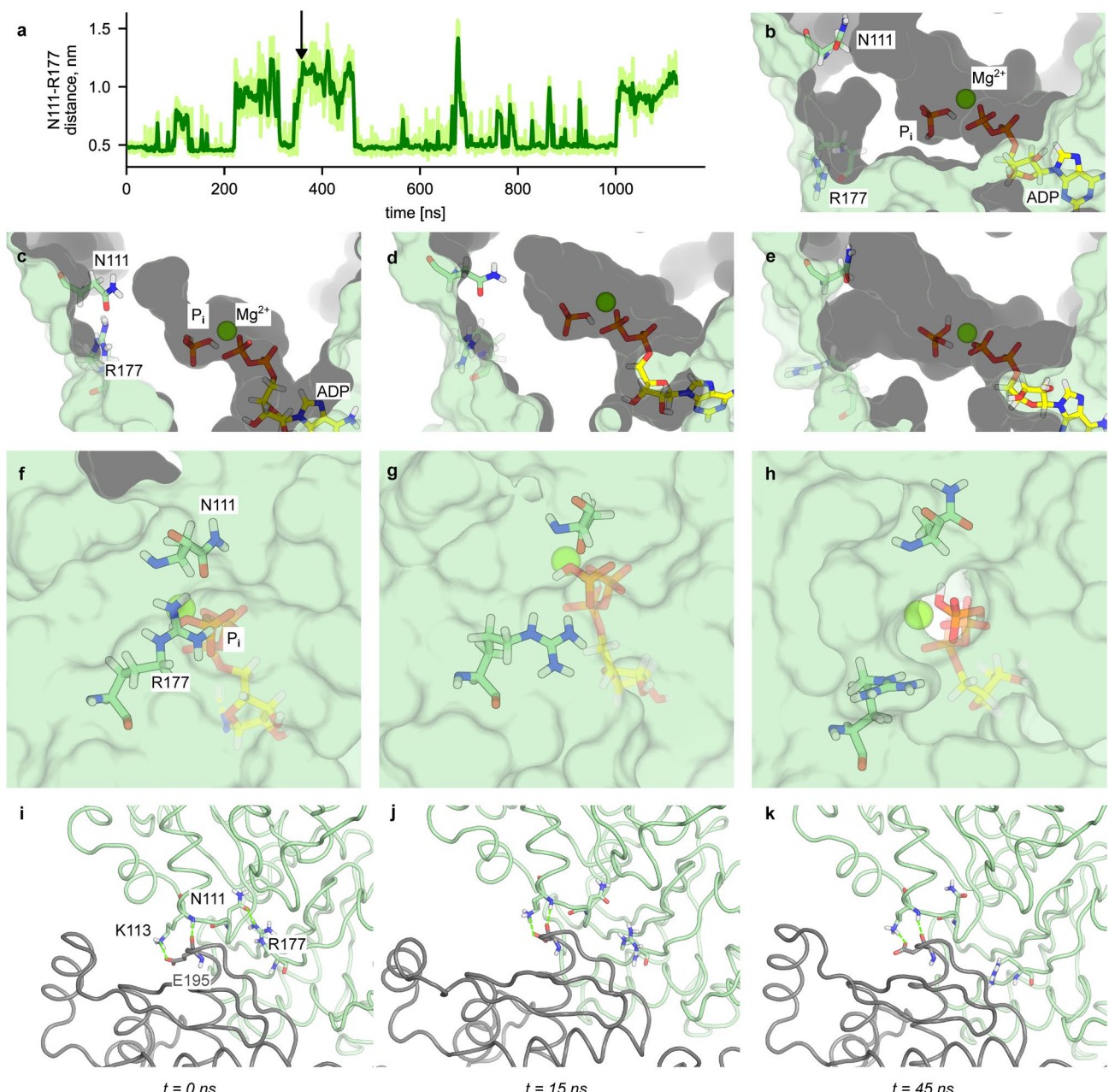

*t = 0 ns*          *t = 15 ns*          *t = 45 ns*

**Extended Data Fig. 9 | Structural dynamics of the R177-N111 backdoor in unbiased and steered MD simulations. a,** Time-series of the R177CZ-N111CG distance (depicted in light green) in an unbiased MD simulation, revealing that the R177-N111 hydrogen bond reversibly breaks and re-forms in unbiased MD simulations. A distance larger than 0.6 nm indicates disruption of the hydrogen bond. For clarity, a centered 2 ns-moving average is shown in dark green. The arrow indicates the used time-frame to capture the F-actin conformation shown in panel b. **b**, Example configuration with a broken R177-N111 hydrogen bond sampled at 360 ns during the unbiased MD simulation. The surface

representation indicates that the backdoor is closed despite the broken R177-N111 hydrogen bond. Solvent accessible area for $P_i$ is depicted in dark grey. There is no accessible path for $P_i$ to escape from the F-actin interior. **c-k** Snapshots from a representative SMD simulation (neutral meH73, replicate 1). Frames are the same as in Fig. 6. **c–e,** Solvent-accessible surface representation of the nucleotide binding site and R177-N111 backdoor, side view. The solvent-accessible volume is shown in dark grey. **f–h,** Solvent-accessible surface representation of the R177-N111 backdoor, external view. Note that in **h,** $P_i$ is visible from the outside. **i–k,** Close-up on the inter-subunit contacts.

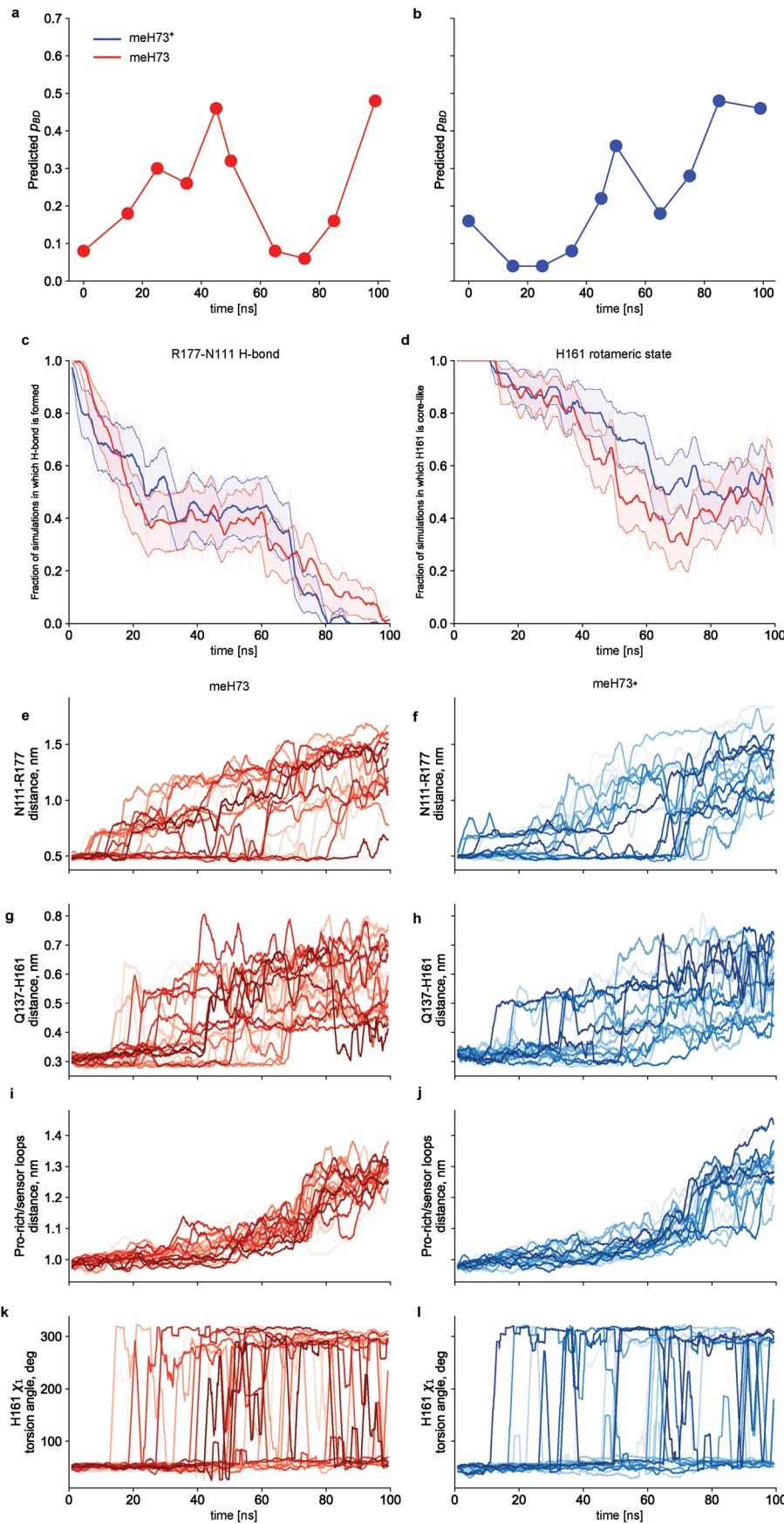

**Extended Data Fig. 10 | See next page for caption.**

**Extended Data Fig. 10 | Evolution of key observables during SMD simulations.**
**a,** Predicted fraction $_{pBD}$ of $P_i$ egress through the R177-N111 backdoor along SMD replicate 1, with neutral meH73, as obtained by classifier analysis of the enhanced sampling trajectories. **b**, Predicted fraction $_{pBD}$ of $P_i$ egress through the R177-N111 backdoor along SMD replicate 1, with positively charged meHis73, as obtained by classifier analysis of the enhanced sampling trajectories. **c**, Survival analysis of the R177-N111 hydrogen bond in SMD simulations. For each protonation state of meH73, we computed the fraction of trajectories with the R177-N111 hydrogen bond still formed at time t. A distance cut-off of 0.6 nm between atoms CZ of R177 and CG of N111 was used to define a formed hydrogen

bond. The solid lines represent the 2 ns-rolling average of the survival fraction. The dotted lines represent the 2 ns-rolling averages of the survival fraction ± SEM. The background overlays represent the raw values for the survival fraction ± SEM. **d**, Survival analysis of the rotameric state of H161. For each protonation state of meH73, we computed the fraction of trajectories with H161 in a core-like rotameric state. H161 side-chain rotameric states were defined by $\chi_1$ torsion angle being above or below 180 deg. Representation conventions are the same as for panel **c**. **e**–**l**. Evolution of structural observables during for all SMD simulation replicates. For clarity, 2 ns-moving averages are shown.

|---|---|

# Reporting Summary

## Statistics

For all statistical analyses, confirm that the following items are present in the figure legend, table legend, main text, or Methods section.

| n/a | Confirmed | |
|---|---|---|
| ☐ | ☒ | The exact sample size (*n*) for each experimental group/condition, given as a discrete number and unit of measurement |
| ☐ | ☒ | A statement on whether measurements were taken from distinct samples or whether the same sample was measured repeatedly |
| ☒ | ☐ | The statistical test(s) used AND whether they are one- or two-sided<br>*Only common tests should be described solely by name; describe more complex techniques in the Methods section.* |
| ☒ | ☐ | A description of all covariates tested |
| ☐ | ☒ | A description of any assumptions or corrections, such as tests of normality and adjustment for multiple comparisons |
| ☐ | ☒ | A full description of the statistical parameters including central tendency (e.g. means) or other basic estimates (e.g. regression coefficient) AND variation (e.g. standard deviation) or associated estimates of uncertainty (e.g. confidence intervals) |
| ☒ | ☐ | For null hypothesis testing, the test statistic (e.g. *F*, *t*, *r*) with confidence intervals, effect sizes, degrees of freedom and *P* value noted<br>*Give P values as exact values whenever suitable.* |
| ☒ | ☐ | For Bayesian analysis, information on the choice of priors and Markov chain Monte Carlo settings |
| ☒ | ☐ | For hierarchical and complex designs, identification of the appropriate level for tests and full reporting of outcomes |
| ☒ | ☐ | Estimates of effect sizes (e.g. Cohen's *d*, Pearson's *r*), indicating how they were calculated |

*Our web collection on statistics for biologists contains articles on many of the points above.*

## Software and code

Policy information about availability of computer code

| Data collection | Cryo-EM data was collected using the commercially available software EPU version 2.8 (Thermofisher Scientific).<br><br>Molecular dynamics simulations were performed using GROMACS 2021.5 and colvars 2022-02-20-dev. |
|---|---|
| Data analysis | Cryo-EM data collection was monitored and preprocessed on the fly using TranSPHIRE version 1.5.13. The preprocessing steps in TranSPHIRE involved gain and drift correction using UCSF MotionCor2 v1.3.0, CTF estimation with CTFFIND4 v4.1.13, and particle picking using SPHIRE-crYOLO v1.5.8. All cryo-EM data were further processed using helical SPHIRE v1.4, RELION v3.1.0 and CryoSPARC v3.3.2. Protein model building was performed in COOT v0.9.8.1 and the models were refined using phenix real-space refine v1.20.1-4487-000. Protein models were validated within the phenix suite v1.20.1-4487-000. Figures and videos that depict cryo-EM density maps and protein structures were prepared using UCSF ChimeraX v1.5.<br><br>Bulk biochemical data was analyzed using Origin Pro version 9.0G. Kinetic simulations were carried out in KinTek Explorer version 6.3.<br><br>All scripts used for data analysis of single-filament assays can be retrieved from https://github.com/iamankitroy/Actin-Pi-Release/. Single filament microfluidic timelapse images were denoised using custom python script, NL-Means_Denoise.py. Single filaments were tracked with the JFilament 1.02 plugin on Image 2.9.0/1.53t. Calculation of filament length and alignment of length and depolymerization time was performed with analyze_filament_tracking.py. Instantaneous depolymerization velocity and their fits were computed with filament_instant-depolVelocity_analysis.py. Kinetic constants were calculated for WT filaments using calc_kinetic-constants.py and estimated for N111S mutant filaments using mutant-kinetic-contants-estimation.ipynb. All scripts were executed with Python 3.9.16 with the following version of modules: Scikit-Image 0.19.3, Numpy 1.24.2, SciPy 1.10.1 and Pandas 1.5.3. All single molecule data were plotted in R 4.2.2 with ggplot 3.3.6.<br><br>Molecular dynamics simulations were analyzed using Python 3.9.12, MDAnalysis 2.3.0, numpy 1.20.3 and pandas 1.5.2. Machine learning was performed using scikit-learn 1.1.3, scipy 1.9.3. Figures that visualize molecular dynamics trajectories were created using Python 3.9.12 and VMD 1.9.4. Model rebuilding for MD was performed using MODELLER v10.2. Hydrogen atoms were added with CHARMM c43b2. |

Live cell imaging movies were analyzed using Python 3.10.10, pandas 2.0.0 and numpy 1.23.5. Figures were created using GraphPad Prism 9.

For manuscripts utilizing custom algorithms or software that are central to the research but not yet described in published literature, software must be made available to editors and reviewers. We strongly encourage code deposition in a community repository (e.g. GitHub). See the Nature Portfolio guidelines for submitting code & software for further information.

## Data

Policy information about availability of data

All manuscripts must include a data availability statement. This statement should provide the following information, where applicable:
- Accession codes, unique identifiers, or web links for publicly available datasets
- A description of any restrictions on data availability
- For clinical datasets or third party data, please ensure that the statement adheres to our policy

The cryo-EM maps generated in this study have been deposited in the Electron Microscopy Data Bank (EMDB) under accession codes (dataset in brackets): EMD-16887 (β/γ-actin barbed end), EMD-16888 (R183W-F-actin) and EMD-16889 (N111S-F-actin). These depositions include sharpened and unsharpened maps, unfiltered half-maps and the masks used for refinements. The associated protein models have been deposited in the Protein Data Bank (PDB) with accession codes 8OI6 (β/γ-actin barbed end), 8OI8 (R183W-F-actin) and 8OID (N111S-F-actin). The following previously published protein models were used for data analysis and comparisons: 8A2S [https://doi.org/10.2210/pdb8A2S/pdb], 8A2T [https://doi.org/10.2210/pdb8A2T/pdb] and 2V52 [https://doi.org/10.2210/pdb2V52/pdb]. We used EMD-15109 [https://www.ebi.ac.uk/emdb/EMD-15109] as 3D model for the first refinements of R183W- and N111S-F-actin. The sequence of human β-actin (P60709, ACTB_HUMAN) was retrieved from UniProt [https://www.uniprot.org/uniprotkb/P60709/entry]. MD simulation models and protocols, MD simulation datasets and Jupyter notebooks to reproduce the analyses reported in Extended Data Figs. 4, 9, 10 have been deposited in Zenodo (10.5281/zenodo.7765025). All other materials are available from the corresponding authors upon request.

# Field-specific reporting

Please select the one below that is the best fit for your research. If you are not sure, read the appropriate sections before making your selection.

☒ Life sciences ☐ Behavioural & social sciences ☐ Ecological, evolutionary & environmental sciences

For a reference copy of the document with all sections, see nature.com/documents/nr-reporting-summary-flat.pdf

# Life sciences study design

All studies must disclose on these points even when the disclosure is negative.

| Sample size | Sample sizes for the three cryo-EM datasets presented in this study: For the barbed end dataset, 1,316 micrographs were collected. 262,982 total particles were picked and 43,618 particles were used for the final reconstruction. For the R183W-F-actin dataset, 7,916 micrographs were collected. 1,569,882 total particles were picked and 1,286,604 particles were used for the final reconstruction. For the N111S-F-actin dataset, 9,516 micrographs were collected. 2,001,281 total particles were picked and 1,756,928 particles were used for the final reconstruction.<br>These sample sizes of ~1000-10,000 micrographs are common in the cryo-EM field for obtaining high-resolution protein structures, see for example Oosterheert et al. Nature (2022): https://doi.org/10.1038/s41586-022-05241-8.<br><br>The in vitro bulk phosphate release assays were performed as three independent experiments per actin variant. The sample size of n=3 is common for in vitro assays with purified proteins, see for example Belyy et al. Plos Biol. (2020): https://doi.org/10.1371/journal.pbio.3000925<br><br>For single filament biochemical assays, we similarly adhered to established practices in the field, by analyzing more than 20 filaments per experimental condition, see Jegou et al Plos Biol (2011): https://doi.org/10.1371/journal.pbio.1001161.g002. Such number is sufficient, because hundreds of depolymerizing events occur per filament depolymerization phase.<br><br>The yeast drop test assays were performed in three independent experiments. Live cell imaging experiments were performed for four independent colonies per yeast strain. More than 85 patches were analyzed per strain, which corresponds to a statistical power of 98% to detect effects in patch lifespan of 5 seconds. For other similar studies see Kaksonen et al., Cell (2005): https://doi.org/10.1016/j.cell.2005.09.024. |
|---|---|
| Data exclusions | During the cryo-EM image processing, particles that represented false picks or particles that did not contribute high-resolution information to the reconstructions were discarded through 2D and 3D classification procedures. This process, which is required to obtain high-resolution reconstructions, is a standard procedure in cryo-EM image processing.<br><br>In TIRF microscopy experiments determining actin filament depolymerization velocities, filaments that stick to the surface or exhibited prolonged pauses during depolymerization were excluded from the analysis. The latter is a well-known artefact resulting from photo-induced oligomerization, see for example Niedermeyer et al. PNAS (2012): https://doi.org/10.1073/pnas.1121381109.<br><br>For the live cell imaging experiments only patches where the full lifespan was captured within the movies were considered, and patches with a lifespan shorter than 4 seconds were not taken into account to avoid cable detection artifacts, see for example Planade, Belbahri et al., PLoS Biol., (2019): https://doi.org/10.1371/journal.pbio.3000500. |
| Replication | All cryo-EM datasets were collected in one session per structure and were not repeated. It is unattainable from a time and cost perspective to repeat cryo-EM data collection and processing on the exact same sample. |

The in vitro bulk phosphate release assays were performed in triplicate. They were performed as independent experiments, with new protein aliquots from the same purification batch. All attempts at replication were successful.

The single filament microfluidic experiments were performed in replicates of 6 for wild-type actin and 12 for the N111S mutant. They were performed as independent experiments, with new protein aliquots from the same purification batch. All attempts at replication were successful.

The in vivo experiments were performed in duplicate for four independent colonies per strain. They were performed as independent experiments, growing new cultures of cells at different days. All attempts at replication were successful.

Randomization — For the 3D refinement of cryo-EM structures, particles were randomly split into two half sets. For all other experiments, randomization was not required because there were no confounding factors in our experiments that could have led to biased results. Covariates were not controlled.

Blinding — This study does not involve any experiments where blinding would be applicable, because there were no confounding factors in our experiments that could have led to biased results.

# Behavioural & social sciences study design

All studies must disclose on these points even when the disclosure is negative.

Study description — *Briefly describe the study type including whether data are quantitative, qualitative, or mixed-methods (e.g. qualitative cross-sectional, quantitative experimental, mixed-methods case study).*

Research sample — *State the research sample (e.g. Harvard university undergraduates, villagers in rural India) and provide relevant demographic information (e.g. age, sex) and indicate whether the sample is representative. Provide a rationale for the study sample chosen. For studies involving existing datasets, please describe the dataset and source.*

Sampling strategy — *Describe the sampling procedure (e.g. random, snowball, stratified, convenience). Describe the statistical methods that were used to predetermine sample size OR if no sample-size calculation was performed, describe how sample sizes were chosen and provide a rationale for why these sample sizes are sufficient. For qualitative data, please indicate whether data saturation was considered, and what criteria were used to decide that no further sampling was needed.*

Data collection — *Provide details about the data collection procedure, including the instruments or devices used to record the data (e.g. pen and paper, computer, eye tracker, video or audio equipment) whether anyone was present besides the participant(s) and the researcher, and whether the researcher was blind to experimental condition and/or the study hypothesis during data collection.*

Timing — *Indicate the start and stop dates of data collection. If there is a gap between collection periods, state the dates for each sample cohort.*

Data exclusions — *If no data were excluded from the analyses, state so OR if data were excluded, provide the exact number of exclusions and the rationale behind them, indicating whether exclusion criteria were pre-established.*

Non-participation — *State how many participants dropped out/declined participation and the reason(s) given OR provide response rate OR state that no participants dropped out/declined participation.*

Randomization — *If participants were not allocated into experimental groups, state so OR describe how participants were allocated to groups, and if allocation was not random, describe how covariates were controlled.*

# Ecological, evolutionary & environmental sciences study design

All studies must disclose on these points even when the disclosure is negative.

Study description — *Briefly describe the study. For quantitative data include treatment factors and interactions, design structure (e.g. factorial, nested, hierarchical), nature and number of experimental units and replicates.*

Research sample — *Describe the research sample (e.g. a group of tagged Passer domesticus, all Stenocereus thurberi within Organ Pipe Cactus National Monument), and provide a rationale for the sample choice. When relevant, describe the organism taxa, source, sex, age range and any manipulations. State what population the sample is meant to represent when applicable. For studies involving existing datasets, describe the data and its source.*

Sampling strategy — *Note the sampling procedure. Describe the statistical methods that were used to predetermine sample size OR if no sample-size calculation was performed, describe how sample sizes were chosen and provide a rationale for why these sample sizes are sufficient.*

Data collection — *Describe the data collection procedure, including who recorded the data and how.*

Timing and spatial scale — *Indicate the start and stop dates of data collection, noting the frequency and periodicity of sampling and providing a rationale for these choices. If there is a gap between collection periods, state the dates for each sample cohort. Specify the spatial scale from which the data are taken*

| | |
|---|---|
| Data exclusions | *If no data were excluded from the analyses, state so OR if data were excluded, describe the exclusions and the rationale behind them, indicating whether exclusion criteria were pre-established.* |
| Reproducibility | *Describe the measures taken to verify the reproducibility of experimental findings. For each experiment, note whether any attempts to repeat the experiment failed OR state that all attempts to repeat the experiment were successful.* |
| Randomization | *Describe how samples/organisms/participants were allocated into groups. If allocation was not random, describe how covariates were controlled. If this is not relevant to your study, explain why.* |
| Blinding | *Describe the extent of blinding used during data acquisition and analysis. If blinding was not possible, describe why OR explain why blinding was not relevant to your study.* |

Did the study involve field work? ☐ Yes ☒ No

# Reporting for specific materials, systems and methods

We require information from authors about some types of materials, experimental systems and methods used in many studies. Here, indicate whether each material, system or method listed is relevant to your study. If you are not sure if a list item applies to your research, read the appropriate section before selecting a response.

## Materials & experimental systems

| n/a | Involved in the study |
|---|---|
| ☒ | ☐ Antibodies |
| ☐ | ☒ Eukaryotic cell lines |
| ☒ | ☐ Palaeontology and archaeology |
| ☒ | ☐ Animals and other organisms |
| ☒ | ☐ Human research participants |
| ☒ | ☐ Clinical data |
| ☒ | ☐ Dual use research of concern |

## Methods

| n/a | Involved in the study |
|---|---|
| ☒ | ☐ ChIP-seq |
| ☒ | ☐ Flow cytometry |
| ☒ | ☐ MRI-based neuroimaging |

## Eukaryotic cell lines

Policy information about cell lines

| | |
|---|---|
| Cell line source(s) | BTI-Tnao38, species of origin - Trichoplusia ni. The cells were provided by S. Wohlgemuth and A. Musacchio (MPI Dortmund, Germany). The Musacchio lab obtained the cell line from G. Blissard (Cornell University, NY, USA) in 2011 - see Hashimoto et al. BMC Biotechnol. (2012) - doi: 10.1186/1472-6750-12-12. The BTI-Tnao38 cell line was used in our study for the production of recombinant β-actin variants. |
| Authentication | The BTI-Tnao38 cell line was not authenticated |
| Mycoplasma contamination | The BTI-Tnao38 cell line was not tested for mycoplasma contamination. |
| Commonly misidentified lines (See ICLAC register) | Research Resource Identifier: CVCL_Z252. |

