## [Peer Review File · Nature Structural & Molecular Biology]

Peer Review Information

Manuscript Title: Molecular mechanisms of inorganic-phosphate release from the core and barbed end of actin filaments

Corresponding author name(s): Gerhard Hummer, Peter Bieling, Stefan Raunser

Reviewer Comments & Decisions:

Decision Letter, initial version:

Message: 16th May 2023

Dear Dr. Raunser,

Thank you again for submitting your manuscript "Molecular mechanisms of inorganic-phosphate release from the core and barbed end of actin filaments". We now have comments (below) from the 3 reviewers who evaluated your paper. In light of those reports, we remain interested in your study and would like to see your response to the comments of the referees, in the form of a revised manuscript.

You will see that while all reviewers appreciate the results, they do raise concerns which will need to be addressed in a revision. Specifically, in line with reviewer's #2 comments, please validate that the mini-filaments are an appropriate tool to measure actin dynamics. We would also ask that you revisit the analysis of single filament experiments to follow suggestions of reviewer #1. We agree with reviewer #2 that the *in vivo* investigation of actin dynamics using fluorescence microscopy would further strengthen the manuscript, if feasible. We would encourage further discussion to put the findings in a context of cellular environment.

Please be sure to address/respond to all concerns of the referees in full in a point-by-point response and highlight all changes in the revised manuscript text file. If you have comments that are intended for editors only, please include those in a separate cover letter.

We expect to see your revised manuscript within 6 weeks. If you cannot send it within this

time, please contact us to discuss an extension; we would still consider your revision, provided that no similar work has been accepted for publication at NSMB or published elsewhere.

Reporting Summary:

When submitting the revised version of your manuscript, please pay close attention to our [href="https://www.nature.com/nature-portfolio/editorial-policies/image-integrity">Digital Image Integrity Guidelines. and to the following points below:](https://www.nature.com/nature-portfolio/editorial-policies/image-integrity)

Please note that all key data shown in the main figures as cropped gels or blots should be presented in uncropped form, with molecular weight markers. These data can be aggregated into a single supplementary figure item. While these data can be displayed in a relatively informal style, they must refer back to the relevant figures. These data should be submitted with the final revision, as source data, prior to acceptance, but you may want to start putting it together at this point.

SOURCE DATA: we request that authors provide, in tabular form, all the data underlying the graphical representations used in figures. This is to further increase transparency in data reporting, as detailed in this editorial (<http://www.nature.com/nsmb/journal/v22/n10/full/nsmb.3110.html>). Spreadsheets can be submitted in excel format. Only one (1) file per figure is permitted; thus, for multi-paneled figures, the source data for each panel should be clearly labeled in the Excel file; alternately the data can be provided as

multiple, clearly labeled sheets in an Excel file. When submitting files, the title field should indicate which figure the source data pertains to. We request our authors to provide source data at the revision stage, so that they are part of the peer-review process. Please also include the uncropped blots and gels in the Source data file.

Data availability: this journal strongly supports public availability of data. All data used in accepted papers should be available via a public data repository, or alternatively, as Supplementary Information. If data can only be shared on request, please explain why in your Data Availability Statement, and also in the correspondence with your editor. Please note that for some data types, deposition in a public repository is mandatory - more information on our data deposition policies and available repositories can be found below: <https://www.nature.com/nature-research/editorial-policies/reporting-standards#availability-of-data>

[Redacted]

Sincerely,

Katarzyna Ciazynska
(she/her)
Associate Editor
Nature Structural & Molecular Biology
<https://orcid.org/0000-0002-9899-2428>

Referee expertise:

Referee #1: actin assembly dynamics, biochemistry

Referee #2: cryo-EM, cytoskeleton

Referee #3: actin dynamics, cell biology

Reviewers' Comments:

Reviewer #1:

Remarks to the Author:

This manuscript by Oosterheert and colleagues tackles the long-standing question of how inorganic phosphate (Pi) is released from the actin subunits in an actin filament, following ATP hydrolysis. This reaction is of key importance since it embodies the aging of the filament, and changes its mechanical properties as well as its interaction with regulatory proteins.

Recent work from the same authors (reference 8 of this manuscript) had shown that the "back door" through which Pi has been proposed to escape was actually closed for the actin subunits within the filament. Here, they use molecular dynamics simulations to show how thermal fluctuations could lead to the opening of the back door, and Pi release.

More interestingly, the authors also provide high resolution structure of the actin filament barbed end. I believe this is the first time this is done, and it provides a trove of new information. For instance, the data show that the barbed end subunits adopt a flattened conformation, similar to the F-actin conformation of subunits in the core of the filament, and that the rapid release of Pi from subunits at the filament barbed end results from a more permanent opening of the back-door. The data also reveal that only the very last terminal subunit at the barbed end differs significantly from the F-actin structure, adopted by all the other subunits. The data also provides interesting insights into the triggering of ATP hydrolysis (resulting in ADP+Pi) during filament elongation.

Finally, the authors use actin mutants (related to pathologies) to test and expand their structural understanding of Pi release.

I find the manuscript well written and clear. The sketches, cartoons and movies nicely illustrate the submolecular processes at hand, and ease their understanding. I believe it will be of interest to a broad readership.

I have some suggestions to improve the manuscript. I recommend its publication,

provided that the authors address the following points:

1. About the steered molecular dynamics simulations: to the non-expert reader, it may seem very puzzling that the opening of the back door, which is an infrequent event in the filament (taking place over minutes), can be efficiently described with simulations that cover a very short time-span (10 ns). I understand that this is because of the metadynamics protocol used to enhance the sampling, as explained in the Methods (line 860 and following). However, for the sake of clarity, I think it would help if this point was briefly explained in the main text.

2. About the cryo-EM structure of filament barbed ends: the rationale for obtaining a high density of short filaments is unclear. What is the purpose of FH2 and DNase-I? Please explain.

Also, isn't it odd that no ABP is observed at the barbed end? Why do you think that is? Could the presence of FH2 on a fraction of the observed barbed ends be missed and alter the conclusions? Please comment.

Finally, one could expect the terminal subunit's conformation at the barbed to fluctuate considerably more than for the rest of the subunits, in the filament. Do the cryo-EM data provide any insights into this?

3. About the single filament experiments with microfluidics: some filaments seem to pause, or stop fully and keep a constant length, during their depolymerization (supp movie 4). How are these pauses taken into account when analyzing the data? Isn't there a risk that short pauses could be missed and affect the measured depolymerization rate?

The paused filaments do not appear stuck on the surface. Could these pausing events be similar to the previously reported photo-induced pauses, observed in similar experiments (Niedermayer et al. PNAS 2012) though with a different fluorescent labeling method?

Also, isn't there a chance that the presence of LifeAct might impact the depolymerization rate? (could LifeAct cause the pauses observed during depolymerization?). Please provide data, or a reference, indicating that this impact is negligible in these experiments.

4. About the analysis of the single filament experiments: the x-axis of Fig 5c indicates that the time since depolymerization is taken into account, yet this seems incorrect. It would be more appropriate to take into account the time since each portion of filament was polymerized, as this marks the onset of the Pi release process (if one neglects the brief delay of hydrolyzing ATP into ADP+Pi). This should be corrected (i.e. recalculating the correct x for each data point) and I expect that it would shift and stretch the data set to the right, and that the fit would then result in a lower rate of Pi release. This could in fact bring it closer to the value estimated from solution assays (Fig 3b).

Also, in fig 5c, why do the data points for wild type start so late? Even with a sliding window of size 20 to determine the instantaneous depolymerization velocity (as explained line 652), it seems that there should be data points at shorter time scales. These missing data points would make the parameters estimated from the fit more accurate. Please explain. If possible, please use these data points at short time scale.

More minor points:

- Fig 1b, it is not easy to distinguish the orange-red from orange (at least on my screen).
- Fig 2b, is the F-actin structure fluctuating during that time? The image gives the impression that the Pi is travelling through a very static structure, so perhaps this should be clarified in the legend.
- Fig 3a, in the sketch, the MDCC Pi-sensor seems to emerge from the filament. The sketch should be drawn to show Pi being released from the filament and then interacting with the MDCC sensor.
- Ref 51 is cited for the purification and biotinylation of spectrin-actin seeds. However, this reference does not say anything about biotinylating the seeds. Please describe (even briefly) how the seeds were biotinylated. Also, the original protocol for the purification of spectrin-actin seeds is in Casella, Maack & Lin, JBC 1986, which should be cited along with ref 51.

Reviewer #2:

Remarks to the Author:

Work in the manuscript builds on recent understanding of coupling between F-actin structure and nucleotide-state revealed with high-resolution precision (PMID: 36289337; PMID: 36289330). The current study provides critical new evidence about the mechanism of phosphate release from F-actin using an elegant combination of cryo-EM structure determination, molecular dynamics calculations and in vitro actin filament dynamics. The resulting clarity about this important but poorly understood transition represents a step-change in comprehension of the actin cytoskeleton.

Using actin filament end structure as a starting point, sets of MD calculations, further structures of actin point mutants and in vitro filament dynamics reconstitution are characterized to provide a coherent view of the backdoor phosphate release mechanism of F-actin. The manuscript is very high quality both in terms of the work performed and its presentation, and the findings will be of wide significance for both technical and biological readers. A few caveats and additional question should be addressed/clarified:

- 1) What is the evidence that the mini-filaments used to determine the actin barbed end structure (Fig. 1) capture a depolymerization state, especially since they are formed in the presence of an actin stabilizing drug? It is crucial that the logic here is clear, since the findings concerning the phosphate release backdoor from this structure frame the rationale of the rest of the study.
- 2) Relatedly, although the literature context for studying actin depolymerization conditions with respect to phosphate released is provided (paragraph from line 78), the relevance to conditions of ATP-driven actin polymerization and in-filament ATPase are less obvious to the more general reader, and could be more clearly explained.
- 3) Reynolds et al (PMID: 36289330) recently described how bound phosphate profoundly affects the mechanical properties of F-actin. What is the effect of filament bending on the behaviour of the phosphate release backdoor and its dynamics?
- 4) The coloring of plausible/improbable release pathways needs to be explicit in the legend of figure 2b

Reviewer #3:

Remarks to the Author:

Oosterheert and colleagues present molecular dynamics and in vitro reconstitution experiments aimed at explaining the mechanism by which the inorganic phosphate Pi is released from actin subunits of polymerised filaments. They conclude that the Pi escapes the actin through a molecular backdoor mechanism based on concerted backbone displacements and rotameric arrangements of keys residues close to the nucleotide binding site. The authors show that: 1) Pi is released more rapidly at the barbed end ultimate subunit, where the molecular backdoor is open. Using cryo-electron microscopy, they solved the structure of short ADP-loaded filaments and demonstrate that H161 of the ultimate subunit adopts a G-like rotameric position, with a disrupted N111-R177 interaction and a stabilised E107-N111 one, leaving a 5 Angstrom diameter hole that probably facilitates the Pi release. 2) the interaction between N111 and R177 constitutes the main pathway for the Pi release. Using molecular dynamics simulations and point mutation experiments, the authors convincingly show that N111S mutation strongly accelerates the rate of Pi release from the filament core. Filament structure determined by cryo-EM revealed that N111S mutant present an open backdoor and a repositioned H161, similarly to the ultimate filament subunit.

This is a very nice study that addresses an important question in actin biology – that is, the molecular mechanism controlling the inorganic phosphate release, which is sensed by many actin-binding proteins controlling actin organisation and dynamics. Overall, the conclusions, which are of immediate interest in the actin field, are well supported by the data presented. I therefore recommend the publication in Nature Structural & Molecular Biology.

Although no additional experiments are absolutely required, the authors need to discuss more explicitly the following issues:

- 1) The discussion remains superficial for understanding the implication and control of Pi release in the cellular context. Given the group's extended knowledge of actin and the abundance of structural and functional data on actin binding proteins, the authors should discuss how the rate of Pi release could be controlled in vivo. For instance, as the nucleotide state of the actin subunits controls the selective binding of accessory proteins, how could these proteins inhibit/stimulate the phosphate release? The authors could discuss the matter for at least the known protein families present at the cellular leading edge. On the other hand, could accessory proteins known to have a higher affinity for ADP-actin, e.g. ADF/cofilin, tropomyosin, myosin, etc., induce a structural reorganisation that accelerates the release?
- 2) The authors investigated the effect of the mutations R183W, R183G and N111S, all of which are associated with severe pathological diseases. Although the effects of R183 mutations were limited in vitro, a broader discussion of how the R183W mutant could explain such defects in actin organisation and/or dynamics in vivo is lacking. For instance, the authors could discuss how the subsequent structural reorganisation of actin subunits may affect the recruitment of key actin binding proteins.
- 3) A limitation of this study may be it has not investigated actin dynamics in vivo. A potential experiment to strengthen the authors' statement could be a direct observation of actin dynamics inside living N111S mutant yeast to follow actin dynamics using high resolution fluorescence microscopy.

Author Rebuttal to Initial comments

Point-to-point response to the reviewers' comments

We thank the reviewers for their positive feedback and insightful comments, which aided us to further improve the manuscript. Below is a point-by-point response to all comments and a detailed description of all changes we have made to our manuscript after considering their suggestions. The changes are highlighted in yellow in the revised text.

Reviewer #1

[1.1] About the steered molecular dynamics simulations: to the non-expert reader, it may seem very puzzling that the opening of the back door, which is an infrequent event in the filament (taking place over minutes), can be efficiently described with simulations that cover a very short time-span (10 ns). I understand that this is because of the metadynamics protocol used to enhance the sampling, as explained in the Methods (line 860 and following). However, for the sake of clarity, I think it would help if this point was briefly explained in the main text.

Reply 1.1: We thank the reviewer for pointing out this lack of clarity. We note that two different enhanced sampling/biasing protocols are used in this study. Short, 10-ns metadynamics simulations are used to push P_i out of the nucleotide binding site in route-agnostic fashion. These simulations make no assumptions about the nature of the backdoor nor the mechanism of its opening, and are used to map sterically accessible egress routes (shown in Fig. 2). Longer, 100-ns Steered MD (SMD) simulations are used to simulate the opening transition of the backdoor after we had acquired independent evidence that this mechanism involves residues N111 and R177 (shown in Fig. 6). These simulations are possible because our barbed end structure is representative of an open-backdoor structure, providing a set of coordinates towards which we can steer the F-actin core structure to efficiently simulate backdoor opening. In both cases, a biasing force is indeed used to accelerate otherwise rare, slow events. To emphasize this important point, we have added the following sentences in the Results section.

On page 6:

“Instead, we developed an enhanced-sampling simulation protocol based on meta-dynamics, which applies a history-dependent repulsive potential on the P_i Cartesian coordinates to progressively drive it out of the

nucleotide-binding site³⁷, without favoring any egress route a priori (see Methods for details). This protocol makes it possible to simulate P_i egress events in about 10 ns.”

On page 12:

“Because this protocol steers the conformational transition in quasi-deterministic fashion, structural rearrangements happen much faster than for stochastic backdoor opening.”

[1.2a] About the cryo-EM structure of filament barbed ends: the rationale for obtaining a high density of short filaments is unclear. What is the purpose of FH2 and DNase-I? Please explain.

Reply 1.2a: We thank the reviewer for mentioning this. The first version of our manuscript did indeed not explain the rationale behind our approach for generating short filaments. We originally decided to keep this first part of the Results section (on how the short filaments were generated) very short to keep the manuscript concise and prevent it from becoming too technical. However, we agree that it is important to give a detailed description of the experimental design for the many scientists within the actin community. We have now expanded the first paragraph of the Results section, as well as a part of the Methods section, to clarify this.

The text added to the Results section on page 4/5:

“This required the generation of short filaments (<150 nm) so that we could pick enough filament ends per micrograph to allow for the high-resolution structure determination of the barbed end. In vitro, actin polymerization generally results in long filaments (>500 nm), because filament growth is kinetically favored over nucleation. We therefore optimized a workflow for generating short filaments which featured DNase I, a G-actin binding protein³⁴ that depolymerizes actin filaments under physiological conditions. However, in the presence of the toxin phalloidin, DNase I does not disassemble F-actin and effectively acts as a pointed-end capper³⁵ that prevents filament reannealing. Accordingly, to generate short filaments, we mixed DNase I – G-actin complex with free G-actin and the FH2 domain of formin mDia1 (mDia1_{FH2}, which acts as actin-nucleator) in low-ionic-strength buffer, and induced polymerization by the addition of KCl. Shortly afterwards, we stabilized the formed filaments with phalloidin and then separated F-actin from unpolymerized actin and mDia1_{FH2} using size-exclusion chromatography (SEC).”

In the Methods section on page 24, we now refer to previous studies that reported the formation of short filaments, and we specify why we could not use the same protocols:

“The in vitro polymerization of actin filaments from purified G-actin generally results in long filaments (>500 nm). Cryo-EM imaging of such filaments at high magnification typically yields 0 – 2 actin ends per micrograph, which does not allow for the averaging of enough end-particles to obtain a high-resolution structure. Hence, previous studies relied on capping protein (CP) as potent actin nucleator and barbed end capper to obtain short filaments^{60,68}. However, CP binds the F-actin barbed end with high affinity and we aimed to reconstruct an undecorated barbed end. We therefore designed a new protocol where we used two other ABPs to create short filaments: DNase I and mDia1_{FH2}, which act as pointed end capper and actin nucleator, respectively.”

[1.2b] Also, isn't it odd that no ABP is observed at the barbed end? Why do you think that is? Could the presence of FH2 on a fraction of the observed barbed ends be missed and alter the conclusions? Please comment.

Reply 1.2b: We do not believe that it is odd that no ABP is observed at the barbed end. The ABPs that are present during our sample preparation are DNase I and mDia1_{FH2}. Firstly, DNase I interacts with the D-loop of actin, and the D-loops of all actin subunits at the barbed end are buried and not available for interactions. Hence, DNase I only binds to the pointed end of the filament, which we also observe at a significant fraction of our short filaments (structural analysis will be published separately). Secondly, mDia1_{FH2} is a formin that nucleates actin filaments and stays processively bound to the barbed end during filament growth. However, the processivity of mDia1 is rather low compared to other formins (see e.g., Cao *et al.* PMID: 29799413), meaning that it is prone to disengage from the filament over long time scales. In our sample preparation, we performed a size-exclusion chromatography step after formation of the short filaments, which is a relatively harsh purification step. We therefore expect that essentially all mDia1_{FH2} is dissociated from the filament, which is in line with our cryo-EM data; in the micrographs, 2D-class averages and 3D reconstruction, we did not observe any evidence for bound mDia1_{FH2}, defining our structure as an undecorated barbed end. The main reason that we added mDia1_{FH2} was to ensure fast actin nucleation in order to obtain short filaments.

[1.2c] Finally, one could expect the terminal subunit's conformation at the barbed to fluctuate considerably more than for the rest of the subunits, in the filament. Do the cryo-EM data provide any insights into this?

Reply 1.2c: We do indeed observe some evidence for a more flexible terminal subunit. For instance, as mentioned in the manuscript, the last 13 C-terminal residues (363 – 375) are not visible in the cryo-EM density, suggesting that they are disordered. In addition, the local resolution of the terminal subunit A₀ is slightly lower than the local resolutions of other subunits in the map (see Extended Data Fig. 1f), indicating

it is slightly more flexible. We used the actin-stabilizing toxin phalloidin for our sample preparation, but we do not believe that this majorly affects the conformation of the ultimate subunit – especially because the ultimate subunit is barely in contact with phalloidin.

[1.3a] About the single filament experiments with microfluidics: some filaments seem to pause, or stop fully and keep a constant length, during their depolymerization (supp movie 4). How are these pauses taken into account when analyzing the data? Isn't there a risk that short pauses could be missed and affect the measured depolymerization rate?

Reply 1.3a: We agree that we should be more explicit in explaining how we dealt with pauses in our data analysis, which we now describe in more detail in the Methods section of the manuscript. In brief, we manually inspected depolymerization time-courses at the level of individual filaments and included only depolymerization data up to onset of pause events. We do not believe that potential short pauses significantly contribute to the measured depolymerization rates because of the following reasons: 1. Pauses appeared to be very long-lived because none of the few filaments analyzed that exhibited pauses resumed growth during the imaging time course (~10 min since depolymerization onset). 2. The parameters we derived from fits to our depolymerization data for wild-type actin very closely match previously measured values using a similar setup (Jegou *et al.* PMID: 21980262). This is quite remarkable, given that the two sets of experiments differ in the labeling method (see below) and actin isoform used, which illustrates the robustness and reproducibility of the specific method and data analysis.

[1.3b] The paused filaments do not appear stuck on the surface. Could these pausing events be similar to the previously reported photo-induced pauses, observed in similar experiments (Niedermayer *et al.* PNAS 2012) though with a different fluorescent labeling method?

Reply 1.3b: We indeed believe that the pauses observed in our single filament experiments are caused by a mechanism equivalent to the one established by Niedermayer *et al.* in their 2012 paper. Similar to their results, we find that the probability distribution of pause onset as function of time is sigmoidal and can be fitted to the analytical model they developed for photo-induced dimerization (Extended Data Fig. 5g). We also observe that filaments subjected to multiple polymerization-depolymerization cycles frequently pause at the same length (not shown). While we do not place the label on actin itself, our method still relies on a fluorescently-tagged short peptide (Lifeact) that places the fluorophore in immediate proximity of the interface between two neighboring actin protomers (Belyy *et al.* PMID: 33216759). Given that fluorophore excitation will produce reactive molecular species, we consider infrequent, stochastic dimerization of adjacent actin protomers very likely. For the reasons outlined in our previous comment (Reply 1.3a), however, we do not believe that these long-lived and infrequent pauses significantly affect our depolymerization data.

[1.3c] Also, isn't there a chance that the presence of LifeAct might impact the depolymerization rate? (could LifeAct cause the pauses observed during depolymerization?). Please provide data, or a reference, indicating that this impact is negligible in these experiments.

Reply 1.3c: In our experiments, we use a very low (50 nM) concentration of Alexa488-LifeAct as fluorescent trace label. According to previous measurements of the Lifeact affinity for filamentous actin (reported K_{DS} of 2 μM and 10 μM by, respectively, Riedl *et al.* PMID: 18536722 and Courtemanche *et al.* PMID: 27159499) we can estimate that only one in 50-250 actin protomers is decorated with Lifeact on average. Such a sparse labeling is in line with the slightly speckled appearance of the Lifeact signal in our TIRF data we observe at high time resolution. In addition, previous experiments demonstrated that Lifeact does not affect actin depolymerization in bulk assays even at micromolar concentrations (Riedl *et al.* PMID: 18536722), which we confirmed under the conditions of our experiments (Reviewer Fig. 1). Finally, the fit parameters we derive from our depolymerization assays for wild-type actin are virtually identical to those obtained by Jegou *et al.* (PMID: 21980262), despite the absence of Lifeact in their assays. Hence, we conclude that the impact of trace amounts of Lifeact is negligible in our experiments.

Reviewer Figure 1:

Depolymerization of 2 μM actin filaments (10% pyrene-labeled) after addition of 5 μM Latrunculin B in either presence (red) or absence (black) of 50nM Alexa488-

[1.4] About the analysis of the single filament experiments: the x-axis of Fig 5c indicates that the time since depolymerization is taken into account, yet this seems incorrect. It would be more appropriate to take into account the time since each portion of filament was polymerized, as this marks the onset of the Pi release process (if one neglects the brief delay of hydrolyzing ATP into ADP+Pi). This should be corrected (i.e. recalculating the correct x for each data point) and I expect that it would shift and stretch the data set to the right, and that the fit would then result in a lower rate of Pi release. This could in fact bring it closer to the value estimated from solution assays (Fig 3b).

Reply 1.4: We are very grateful to the reviewer for identifying and pointing out this error in our original analysis, for which we sincerely apologize. We have corrected this mistake by recalculating the correct lifetime of actin protomers within the filament as detailed in our revised Method section. As correctly anticipated by the reviewer, this recalculation indeed stretches out the data to longer times, which results in P_i release rates that are more closely matched to our bulk experiments and previous data. These newly calculated values are now reported in Fig. 5d.

[1.5] Also, in fig 5c, why do the data points for wild type start so late? Even with a sliding window of size 20 to determine the instantaneous depolymerization velocity (as explained line 652), it seems that there should be data points at shorter time scales. These missing data points would make the parameters estimated from the fit more accurate. Please explain. If possible, please use these data points at short time scale.

Reply 1.5: Our initial analysis for wild-type actin was done with a sliding window of 20 data points flanking the central one (total window size 41). For imaging at 5s intervals, this results in the 100s delay after depolymerization onset in our velocity data. Motivated by this comment, however, we have now systematically explored the effect of the sliding window size on our analysis. We find that the combined prediction error for the three fit parameters of the kinetic model displays a broad minimum as a function of window size (Extended Data Fig. 5e). Altering the window size within this minimum (from 17-29) yields correspondingly smaller or larger delay times in the velocity data, however, with marginal effects on the obtained fit parameters (Extended Data Fig. 5f). Instead of fixing our analysis to an arbitrary window size, we chose a prediction error cutoff of 10% for the combined percentage standard deviation error for all three fit parameters, which includes data from window sizes 17-29. Hence, we now report values that were determined from fits to the arithmetic mean of the combined data. All of these changes are described in detail in the Methods section of our revised manuscript.

More minor points:

[1.6] Fig 1b, it is not easy to distinguish the orange-red from orange (at least on my screen).

Reply 1.6: We agree with this assessment and have changed the color of the penultimate subunit to 'salmon', making it more distinguishable from the other colors. This has been updated in Fig. 1, Extended Data Figs. 2, 3 and Supplementary Video 1.

[1.7] Fig 2b, is the F-actin structure fluctuating during that time? The image gives the impression that the Pi is travelling through a very static structure, so perhaps this should be clarified in the legend.

Reply 1.7: The F-actin structure is indeed fluctuating during the simulation. We have now added the following sentence to the legend: "The F-actin structure fluctuated during the simulations but is shown in a single representative conformation for clarity." This has also been added to the figure legend of Extended Data Fig. 4.

[1.8] Fig 3a, in the sketch, the MDCC Pi-sensor seems to emerge from the filament. The sketch should be drawn to show Pi being released from the filament and then interacting with the MDCC sensor.

Reply 1.8: Thank you for noticing this. We have updated Fig. 3a so that the MDCC is no longer emerging from the filament. An additional P_i-cartoon is shown to emphasize that P_i is released from the filament.

[1.9] Ref 51 is cited for the purification and biotinylation of spectrin-actin seeds. However, this reference does not say anything about biotinylating the seeds. Please describe (even briefly) how the seeds were biotinylated. Also, the original protocol for the purification of spectrin-actin seeds is in Casella, Maack & Lin, JBC 1986, which should be cited along with ref 51.

Reply 1.9: We are grateful to reviewer #1 for this suggestion. We now also cite the original JBC-1986 study by Casella, Maack & Lin when referring to the preparation of spectrin-actin seeds and describe how the seeds were biotinylated in the Method section.

Reviewer #2

[2.1] What is the evidence that the mini-filaments used to determine the actin barbed end structure (Fig. 1) capture a depolymerization state, especially since they are formed in the presence of an actin stabilizing drug? It is crucial that the logic here is clear, since the findings concerning the phosphate release backdoor from this structure frame the rationale of the rest of the study.

Reply 2.1: Thank you for pointing this out. We fully agree that it is important to provide a reasoning for why our barbed end structure is captured in a depolymerizing state. During filament growth, the incoming actin subunit at the barbed end will only adopt the ATP state before becoming an internal subunit. This was previously mentioned in the discussion, but is now added to the introduction to make the reader familiar with this concept and to emphasize why P_i release from the barbed end is only relevant under depolymerizing conditions. In our experiment to reconstitute short filaments for cryo-EM, we rapidly polymerized actin into short filaments, stabilized the filaments with phalloidin and then separated them from all monomeric components by size-exclusion chromatography. Therefore, the cryo-EM sample was prepared in conditions that do not allow for filament growth, because there was essentially no monomer pool left. Accordingly, the terminal subunit in our structure is bound to ADP, which is defined as the

nucleotide state after P_i release, making our structure consistent with a depolymerization state (stabilized by phalloidin). This has now been further specified in the results section.

The updated Introduction section on page 3 states:

“During filament growth, P_i release occurs solely from filament core subunits because the barbed end growth velocity ($\sim 10 - 500$ monomers s^{-1}) (ref.²⁶) is much faster than the ATP hydrolysis rate of actin ($0.3 s^{-1}$), indicating that barbed end subunits effectively only adopt the ATP state before becoming internal subunits. However, after the transition from filament growth to depolymerization, actin subunits that have hydrolyzed ATP and adopt the ADP- P_i state, can be exposed at the shortening barbed end. Interestingly, these barbed end subunits release P_i more than 300-fold faster ($\sim 2 s^{-1}$) than those within the filament core, ...”

The updated Results section on page 5 states:

“All actin subunits in our reconstruction adopt the aged ADP nucleotide state (Extended Data Fig. 2a, b), indicating that we captured the barbed end structure in a state that resembles a depolymerizing filament. This is in line with our experimental setup, where the filaments were separated from actin monomers after polymerization.”

[2.2] Relatedly, although the literature context for studying actin depolymerization conditions with respect to phosphate released is provided (paragraph from line 78), the relevance to conditions of ATP-driven actin polymerization and in-filament ATPase are less obvious to the more general reader, and could be more clearly explained.

Reply 2.2: See Reply 2.1. By specifying the fast barbed-end growth rates of actin in the introduction, we believe that it is now easier to understand why ATP is hydrolyzed in actin subunits that reside in the filament core, rather than at the barbed end.

[2.3] Reynolds et al (PMID: 36289330) recently described how bound phosphate profoundly affects the mechanical properties of F-actin. What is the effect of filament bending on the behaviour of the phosphate release backdoor and its dynamics?

Reply 2.3: This is an interesting point. The work by Reynolds *et al.* (ref. 32) indeed elegantly showed that the presence of P_i at the nucleotide binding site affects shearing forces between actin subunits during

filament bending. However, we do not have evidence that bending directly affects the behavior of the P_i release backdoor.

This is now addressed in the Discussion section of the manuscript on page 13:

“It was recently shown that the presence of P_i at the nucleotide binding site affects the bending structural landscape of F-actin³², which raises the question if the opening of the backdoor could also be affected by filament bending. While backdoor opening involves a confined region of SD1 (Pro-rich loop and sensor loop) and SD3 (R177-strand) in actin, the subdomains that display the largest displacements during filament bending are SD2 and SD4 (ref.³²). Accordingly, the backdoor is closed in all subunits of bent F-actin structures in the ADP (pdb 8D15) and ADP- P_i (pdb 8D16) states, suggesting that the backdoor state is not strongly affected by filament bending. In addition, if filament bending would dramatically affect the P_i release kinetics of actin, one would expect P_i release to be cooperative, i.e., actin subunits in the ADP state would bend more and thereby stimulate P_i release from neighboring subunits that adopt the ADP- P_i state. However, P_i release from the F-actin core is stochastic²¹. Therefore, we do not anticipate that filament bending majorly affects the dynamic opening and closing of the backdoor.”

[2.4] The coloring of plausible/improbable release pathways needs to be explicit in the legend of figure 2b.

Reply 2.4: Thank you. We have now changed the figure legend of Fig. 2b to specifically mention the used colors for plausible and improbable pathways:

“The two plausible egress paths that were analyzed further are the putative R183 backdoor (gold) and R177-N111 backdoor (blue). Improbable P_i release pathways are colored purple and magenta and are further shown in Extended Data Fig. 4.”

Reviewer #3

[3.1] The discussion remains superficial for understanding the implication and control of P_i release in the cellular context. Given the group’s extended knowledge of actin and the abundance of structural and functional data on actin binding proteins, the authors should discuss how the rate of P_i release could be controlled in vivo. For instance, as the nucleotide state of the actin subunits controls the selective binding of accessory proteins, how could these proteins inhibit/stimulate the phosphate release? The authors could discuss the matter for at least the known protein families present at the cellular leading edge. On the other

hand, could accessory proteins known to have a higher affinity for ADP-actin, e.g. ADF/cofilin, tropomyosin, myosin, etc., induce a structural reorganisation that accelerates the release?

Reply 3.1: We fully agree that it is important to place our data in a cellular context. We have now updated the Introduction and Discussion sections to emphasize and discuss the importance of P_i release for actin-mediated cellular processes.

On page 3, the introduction now states:

“For instance, ADF/cofilin family proteins efficiently bind and sever the ADP-bound state of the filament, but bind only weakly to ADP- P_i -F-actin¹⁵⁻¹⁷. Severing by ADF/cofilin promotes actin turnover following network assembly¹⁸, making P_i release from the F-actin interior an essential mechanism for polarized, directed cell migration.”

The following paragraph was added to the Discussion on pages 13 and 14:

“How are the kinetics of P_i release regulated in a complex cellular environment? Interestingly, the rates of P_i release from the filament interior observed *in vitro* appear to be slower than the turnover of some cellular actin structures such as lamellipodial networks or endocytic patches⁴⁵. This strongly suggests that P_i release *in vivo* is accelerated by ABPs through poorly defined mechanisms, which can be discussed in the context of our results. The P_i release backdoor opens towards the inner side of the filament, adjacent to where the two strands of the actin helix interact (Fig. 1d). This site is targeted by small-molecule toxins phalloidin and jasplakinolide, which block the opening of the backdoor and, hence, sterically inhibit P_i release³⁷⁻³⁹. Importantly, however, the backdoor and its immediate surrounding are not known to be directly engaged by many ABPs, presumably because molecules larger than phalloidin and jasplakinolide cannot easily enter the narrow cavity between the two actin strands. Accordingly, factors implicated in actin aging and turnover such as ADF/cofilin and associated regulators like coronin^{46,47} all bind at the filament periphery⁴⁸⁻⁵⁰, suggesting that these ABPs can only affect P_i release allosterically. Such a mechanism has indeed been postulated for ADF/cofilin proteins, which only efficiently bind and sever F-actin following P_i release, but also accelerate the release of P_i from the filament^{15,24}. Structurally, cofilin binding changes the tilt of the F-actin^{48,49}, resulting in rearrangements at the nucleotide-binding site incompatible with the presence of P_i , which renders actin tilting and P_i binding mutually exclusive^{8,49}. Hence, initial cofilin binding to the filament likely results in an actin conformation that releases P_i more rapidly. However, how such ABP-induced conformational changes promote backdoor opening and how this relates to actin disassembly in cells, remains to be elucidated.”

The reviewer also mentions tropomyosin and myosin, and wonders how these ABPs could accelerate P_i release kinetics. Previous work has revealed that myosin-V and VI walk at different speeds on actin dependent on the nucleotide state of the filament (Zimmermann *et al.* PMID: 26190073). However, to the best of our knowledge, there are no reports that myosin and tropomyosin directly alter the P_i release kinetics of F-actin. We have therefore not named these specific ABPs in the Discussion.

[3.2] The authors investigated the effect of the mutations R183W, R183G and N111S, all of which are associated with severe pathological diseases. Although the effects of R183 mutations were limited *in vitro*, a broader discussion of how the R183W mutant could explain such defects in actin organisation and/or dynamics *in vivo* is lacking. For instance, the authors could discuss how the subsequent structural reorganisation of actin subunits may affect the recruitment of key actin binding proteins.

Reply 3.2: We thank reviewer #3 for pointing this out. The reason that we did not discuss the effects of the R183-mutations in detail is that our manuscript focuses mainly on P_i release from actin, and not on other defects regarding actin organization/dynamics. The R183-mutations do not majorly affect P_i release kinetics *in vitro* and, therefore, the diseases linked to these variants are not expected to be caused by alterations in the P_i release kinetics of actin. However, we acknowledge that it is important to also mention how the R183W mutation could result in disease. To this end, it was previously shown by the Manstein and Müller groups that introducing the R183W mutation in actin resulted in perturbed nucleotide release from actin monomers (Hundt *et al.* PMID: 25255767). In addition, they showed that the interaction between R183W-F-actin and non-muscle myosin-2A is impaired. Of note, the work by the Manstein and Müller groups was already cited in the previous version of our manuscript.

We have now added the following sentences to the manuscript on page 10:

“Interestingly, it was previously shown that the R183W mutation results in perturbed nucleotide release from actin monomers and impaired binding of non-muscle myosin-2A to the actin filament⁴¹. It is therefore expected that these altered molecular properties of R183W-actin are the major cause of defects in actin organization that lead to disease, rather than the slightly altered P_i release kinetics of this actin variant.”

[3.3] A limitation of this study may be it has not investigated actin dynamics *in vivo*. A potential experiment to strengthen the authors’ statement could be a direct observation of actin dynamics inside living N111S mutant yeast to follow actin dynamics using high resolution fluorescence microscopy.

Reply 3.3: We agree with the reviewer that it would be interesting to further investigate the effects of N111S-actin on dynamics using fluorescence microscopy. We would however like to emphasize that the primary aim of our work was to understand P_i release from F-actin at the atomic/molecular level.

In our original manuscript, we performed an assay where we investigated the effect of the N111S mutation on yeast growth (Fig. 5e). While no growth difference was observed for yeast expressing N111S-actin compared to the strain expressing wild-type actin, we identified a major growth defect when the N111S-actin expressing yeast was exposed to the toxin Latrunculin A during the two-day growth period. To address the comment by the reviewer, we have performed an additional experiment. Specifically, we have transformed the yeast strains expressing either wild-type- or N111S-actin with a Lifeact-mCherry construct to fluorescently label the actin population. We then performed live-cell fluorescence imaging to quantify the lifetime of endocytic actin patches in the yeast strains (now in: Extended Data Fig. 5i, j). Under normal growth conditions, we observed no large changes in the lifetime of actin patches (Extended Data Fig. 5i, j). This is consistent with our growth phenotype assays, where no growth defects were observed for the N111S-actin strain under normal conditions (Fig. 5e). Unfortunately, exposing the two strains to different concentrations of Latrunculin A will make any quantitative comparisons between live yeast cells far-from-trivial, because Latrunculin A sequesters G-actin, thereby reducing actin dynamics and also resulting in a strong overall decrease of fluorescent signal for F-actin. Therefore, any further live-cell fluorescence imaging experiments were not within the scope of our study, nor would they be in line with the ‘molecular focus’ of our manuscript.

On page 11: the following sentence was now added to the Results section: “Accordingly, live-cell fluorescence imaging of the two yeast variants did not reveal large differences in the lifetime of endocytic actin patches (Extended Data Fig. 5i, j).” A detailed description of the live-cell imaging is now also added to the methods section.

Decision Letter, first revision:

Message: Our ref: NSMB-A47569A

13th Jul 2023

Dear Dr. Raunser,

Thank you for submitting your revised manuscript "Molecular mechanisms of inorganic-phosphate release from the core and barbed end of actin filaments" (NSMB-A47569A). It has now been seen by the original referees and their comments are below. The reviewers find that the paper has improved in revision, and therefore we'll be happy in principle to publish it in Nature Structural & Molecular Biology, pending minor revisions to satisfy the

referees' final requests and to comply with our editorial and formatting guidelines.

Sincerely,
Kat

Katarzyna Ciazynska
(she/her)
Associate Editor
Nature Structural & Molecular Biology
<https://orcid.org/0000-0002-9899-2428>

Reviewer #1 (Remarks to the Author):

I am satisfied with the authors' responses, which address all my concerns. I find the revised manuscript improved and I congratulate the authors on their beautiful work.

I have one final suggestion. A recent paper from the lab of Roberto Dominguez (Carman et al. PMID: 37228182), as well as a recent preprint by Chou and Pollard (on bioRxiv), also report cryo-EM structures for the barbed ends of actin filaments. They do not take anything away from this manuscript, and will certainly not reduce its impact. For the benefit of the reader, the authors could consider commenting briefly, in the discussion, how they position their results with respect to these two recent reports.

Reviewer #2 (Remarks to the Author):

I am satisfied with the edits that the authors have made. Congratulations to them on a beautiful study.

Reviewer #3 (Remarks to the Author):

The revised manuscript is a substantial improvement over the original submission. Several clarifications have been added in multiple sections of the manuscript.

The authors have discussed the importance of Pi release and how it can be controlled in the cellular context, and have clarified the effects of the R183W mutation in vivo. They also convincingly addressed the question of actin patch dynamics in the N111S mutant strain.

As requested, the points raised in my previous review have been satisfactorily addressed ; I consider this work is now acceptable for publication.

Author Rebuttal, first revision:**Point-to-point response to the reviewers' comments – round 2.**

We are extremely grateful to all three reviewers for their positive feedback. Their comments aided us to improve the quality and readability of our manuscript, both for experts and non-experts in the actin field. We are happy that all three reviewers now find the manuscript acceptable for publication. Below is a response to a final suggestion from reviewer #1.

Reviewer #1

[1.1] I have one final suggestion. A recent paper from the lab of Roberto Dominguez (Carman et al. PMID: 37228182), as well as a recent preprint by Chou and Pollard (on bioRxiv), also report cryo-EM structures for the barbed ends of actin filaments. They do not take anything away from this manuscript, and will certainly not reduce its impact. For the benefit of the reader, the authors could consider commenting briefly, in the discussion, how they position their results with respect to these two recent reports.

Reply 1.1: We are aware of the two studies that also reported structures of the undecorated barbed end of actin. However, a comparison between the structures in the discussion is not feasible because: 1) the structure by Chou and Pollard is not released yet, 2) we do not have space for an additional paragraph as our manuscript is already on the long side, and, most importantly, 3) a comparison between the structures would not be in line with the focus of our manuscript. Instead, to guide the reader, we have added the following sentence to the discussion:

“In general, we believe that investigating the structure of actin ends, either undecorated or bound by diverse classes of ABPs, will remain a prevalent theme of future research, guided by recent advances in actin-end structure determination by our lab and others^{45–48}.”

Here, we cite both the work by Carman *et al.* (ref 47) and Chou and Pollard (ref 48).

Final Decision Letter:

Message 18th Aug 2023

:

Dear Dr. Raunser,

We are now happy to accept your revised paper "Molecular mechanisms of inorganic-phosphate release from the core and barbed end of actin filaments" for publication as an Article in Nature Structural & Molecular Biology.

As soon as your article is published, you can generate your shareable link by entering the DOI of your article here: http://authors.springernature.com/share. Corresponding authors will also receive an automated email with the shareable link

Your paper will be published online soon after we receive proof corrections and will appear in print in the next available issue. You can find out your date of online publication by contacting the production team shortly after sending your proof corrections. Content is published online weekly on Mondays and Thursdays, and the embargo is set at 16:00 London time (GMT)/11:00 am US Eastern time (EST) on the day of publication. Now is the

time to inform your Public Relations or Press Office about your paper, as they might be interested in promoting its publication. This will allow them time to prepare an accurate and satisfactory press release. Include your manuscript tracking number (NSMB-A47569B) and our journal name, which they will need when they contact our press office.

About one week before your paper is published online, we shall be distributing a press release to news organizations worldwide, which may very well include details of your work. We are happy for your institution or funding agency to prepare its own press release, but it must mention the embargo date and Nature Structural & Molecular Biology. If you or your Press Office have any enquiries in the meantime, please contact press@nature.com.

Please note that *Nature Structural & Molecular Biology* is a Transformative Journal (TJ). Authors may publish their research with us through the traditional subscription access route or make their paper immediately open access through payment of an article-processing charge (APC). Authors will not be required to make a final decision about access to their article until it has been accepted. [Find out more about Transformative Journals](https://www.springernature.com/gp/open-research/transformative-journals)

Authors may need to take specific actions to achieve [compliance](https://www.springernature.com/gp/open-research/funding/policy-compliance-faqs) with funder and institutional open access mandates. If your research is supported by a funder that requires immediate open access (e.g. according to [Plan S principles](https://www.springernature.com/gp/open-research/plan-s-compliance)) then you should select the gold OA route, and we will direct you to the compliant route where possible. For authors selecting the subscription publication route, the journal's standard licensing terms will need to be accepted, including [self-archiving policies](https://www.springernature.com/gp/open-research/policies/journal-policies). Those licensing terms will supersede any other terms

that the author or any third party may assert apply to any version of the manuscript.

Sincerely,
Kat

Katarzyna Ciazynska
(she/her)
Associate Editor
Nature Structural & Molecular Biology
<https://orcid.org/0000-0002-9899-2428>

Click here if you would like to recommend Nature Structural & Molecular Biology to your librarian:

<http://www.nature.com/subscriptions/recommend.html#forms>